# Temporal and context-dependent requirements for the transcription factor Foxp3 expression in regulatory T cells

Wei Hu [1,2,10] ✉, Gabriel A. Dolsten[3,9], Eric Y. Wang [1,4,5,9], Giorgi Beroshvili[1,4,9], Zhong-Min Wang[1,6,8], Aazam P. Ghelani[1,4], Lion F. K. Uhl[1], Regina Bou-Puerto [1,4], Xiao Huang[1], Anthony J. Michaels[1,4], Beatrice E. Hoyos[1], Wenjie Jin [2], Yuri Pritykin [3,7,10] ✉ & Alexander Y. Rudensky [1,4,6,10] ✉

Regulatory T ($T_{reg}$) cells, expressing the transcription factor Foxp3, are obligatory gatekeepers of immune responsiveness, yet the mechanisms by which Foxp3 governs the $T_{reg}$ transcriptional network remain incompletely understood. Using a novel chemogenetic system of inducible Foxp3 protein degradation in vivo, we found that while Foxp3 was indispensable for the establishment of transcriptional and functional programs of newly generated $T_{reg}$ cells, Foxp3 loss in mature $T_{reg}$ cells resulted in minimal functional and transcriptional changes under steady state. This resilience of the Foxp3-dependent program in mature $T_{reg}$ cells was acquired over an unexpectedly long timescale; however, in settings of severe inflammation, Foxp3 loss led to a pronounced perturbation of $T_{reg}$ cell transcriptome and fitness. Furthermore, tumoral $T_{reg}$ cells were uniquely sensitive to Foxp3 degradation, which led to impairment in their suppressive function and tumor shrinkage in the absence of pronounced adverse effects. These studies demonstrate a context-dependent differential requirement for Foxp3 for $T_{reg}$ transcriptional and functional programs.

Regulatory T ($T_{reg}$) cells are requisite watchmen of the immune system, whose identity is distinguished by the expression of their X-chromosome-encoded lineage-specifying transcription factor Foxp3 (refs. 1–3). Foxp3 plays a critical role in $T_{reg}$ cell differentiation, conferring both suppressor function and fitness, largely by exploiting the pre-established epigenetic landscape in precursor cells before Foxp3 expression[4–6]. In mice on a non-autoimmune prone genetic background showed that Foxp3 expression is stable in differentiated $T_{reg}$ cells, whereas recently generated $T_{reg}$ cells can lose Foxp3 expression[7–9]. In inflammatory settings, $T_{reg}$ cells upregulate Foxp3 expression and increase their suppressor function, yet both can become compromised in severe infections or autoimmunity, in particular in conjunction with genetic predispositions or interleukin (IL)-2 deprivation[7,10,11].

The indispensable role of Foxp3 in establishing $T_{reg}$ cell functionality during their differentiation has been demonstrated by comparisons of mice expressing a $Foxp3^{GFPKO}$ reporter-null versus a functional $Foxp3^{GFP}$ allele[1,4,12]. The currently prevailing notion of a requirement for continuous Foxp3 expression was suggested by Cre recombinase-induced ablation of a $Foxp3$ conditional allele in vitro in rapidly dividing $T_{reg}$ cells followed by their adoptive transfers into lymphopenic hosts[13]; however, the confounders of these early studies left unresolved the question of whether the Foxp3-dependent $T_{reg}$ functional program is intrinsically resilient or vulnerable. This uncertainty is particularly intriguing in light of the recent studies suggesting a model where Foxp3 is acting largely indirectly by inducing relatively modest changes in the expression of a few, yet to be defined, direct target genes, which in turn can act to establish the genome-wide transcriptional and functional

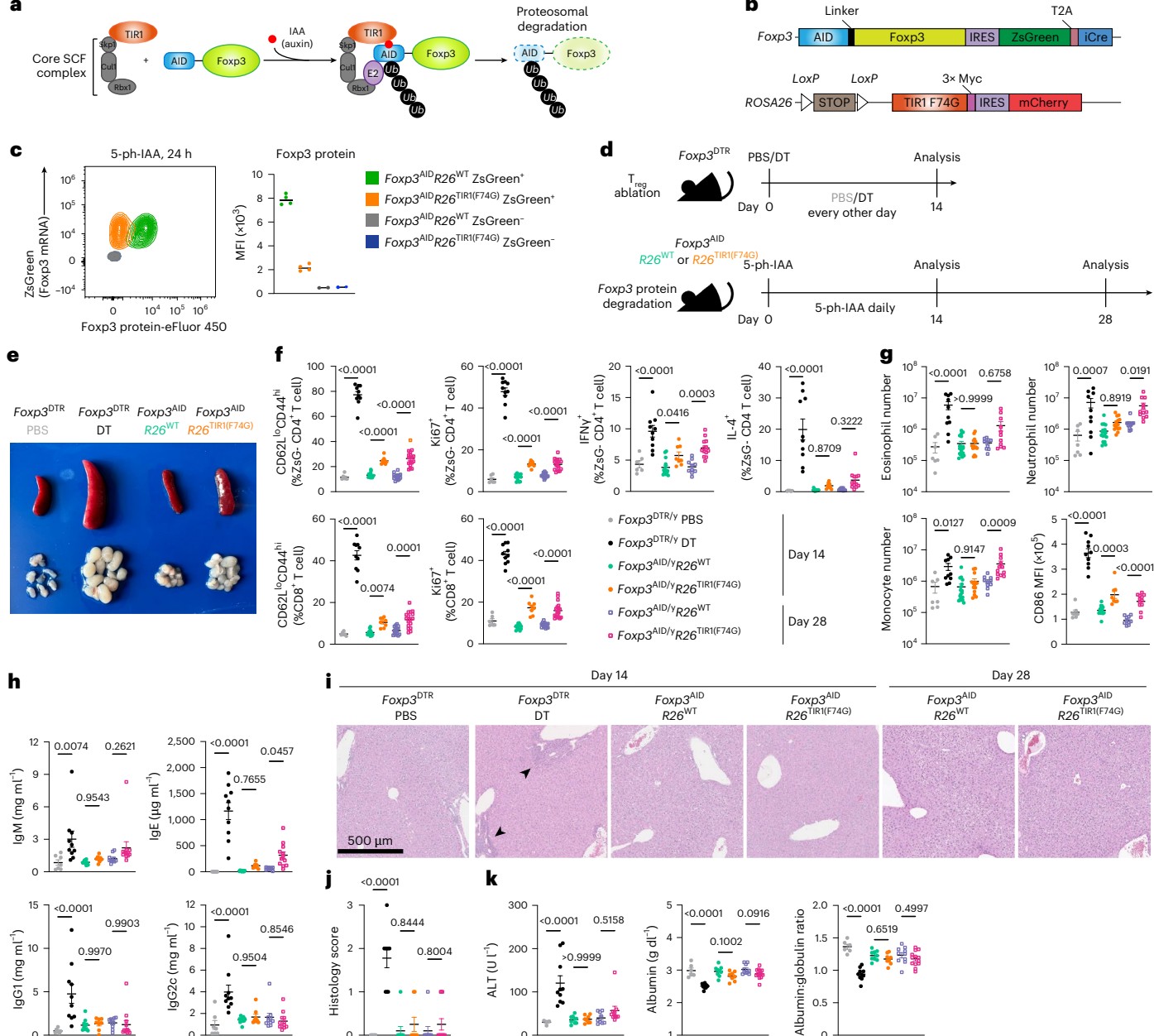

**Fig. 1 | Foxp3 degradation causes minimal immune activation in adult lymphoreplete mice. a**, Schematic of the inducible Foxp3 protein degradation model. SCF, Skp-1-Cullin-F-box complex. **b**, Schematic of the *Foxp3*[AID] and *R26*[TIR] alleles. **c**, Flow cytometry plot showing 5-ph-IAA-induced Foxp3 protein degradation after 24 h (left). Scatter-plot of Foxp3 protein expression median fluorescence intensity (MFI) as assessed by flow cytometric analysis in Foxp3[AID] mice after 7 days of daily 5-ph-IAA injection (right). **d**, Experimental design. **e**, Size of spleen and lymph nodes after 14 days of T$_{reg}$ cell ablation or Foxp3 degradation. **f**, Activation, proliferation and cytokine production of CD4$^+$ (top) and CD8$^+$ (bottom) T cells following T$_{reg}$ cell ablation or Foxp3 degradation.

**g**, Number of eosinophils, neutrophils and monocytes and CD86 levels on dendritic cells following T$_{reg}$ cell ablation or Foxp3 degradation. **h**, Serum antibody levels following T$_{reg}$ cell ablation or Foxp3 degradation. **i,j**, Representative hematoxylin and eosin (H&E) stain (**i**) and histology scores (**j**) of the liver following T$_{reg}$ cell ablation or Foxp3 degradation. **k**, Liver damage measured by serum alanine aminotransferase, albumin and albumin:globulin ratio. Scatter-plots represent mean ± s.e.m. Each point represents a unique mouse. Data are pooled from two independent experiments. Statistical analysis was conducted by one-way analysis of variance (ANOVA).

program of T$_{reg}$ cells[12]. Thus, the role for Foxp3 in the maintenance of transcriptional and functional features of differentiated T$_{reg}$ cells remains unknown despite major previous efforts.

Here we investigated the role of Foxp3 during T$_{reg}$ cell differentiation, maintenance and turnover using a novel chemogenetic model, which enabled punctual inducible degradation of the Foxp3 protein in vivo. By analyzing the transcriptional and functional features of T$_{reg}$ cells following short-term Foxp3 protein degradation, we found that Foxp3 was essential for the establishment of the gene expression

program and T$_{reg}$ cell function during thymic differentiation and in recently differentiated cells. In contrast with the complete loss of the T$_{reg}$ cell-mediated suppression observed in developmental Foxp3 deficiency or T$_{reg}$ cell ablation, Foxp3-degraded mature T$_{reg}$ cells largely maintained their suppressive capacity, both in vivo and ex vivo. Accordingly, Foxp3 degradation led to minimal gene expression changes limited to a small group of genes that are likely enriched for direct Foxp3 targets; however, under severe inflammatory conditions, Foxp3 degradation led to exaggerated transcriptional changes. Finally,

induced Foxp3 protein degradation preferentially destabilized intratumoral $T_{reg}$ cells, leading to a loss of function and tumor rejection. These results highlight the varying roles of Foxp3 in $T_{reg}$ cell phenotypic and functional features in different contexts, shedding light on its distinct mode of action as a transcription factor.

## Results

### Inducible Foxp3 protein degradation in vivo

Despite being central to $T_{reg}$ cell biology, the role of Foxp3 protein expression in developing versus mature $T_{reg}$ cells, particularly the mechanisms underlying the vulnerabilities and resilience of the Foxp3-dependent gene regulatory network in early-life versus adulthood, as well as in health versus disease, remains unknown. A major obstacle to gaining this insight has been the limitations of *Foxp3* gene ablation strategies for dissecting gene regulatory programs, as the prolonged turnover of Foxp3 RNA and protein following gene deletion confounds the distinction between direct and indirect effects.

Therefore, we generated a new chemogenetic mouse model that enables rapid drug-inducible Foxp3 protein degradation in vivo (Fig. 1a,b). In this model, based on the auxin-sensing pathway in plants[14], the endogenous *Foxp3* allele encoding an auxin-inducible degron (AID)–Foxp3 fusion protein alongside a ZsGreen transcriptional reporter and Cre recombinase (Extended Data Fig. 1a,b) was combined with the *ROSA26* (*R26*) allele harboring a plant-derived E3 ligase TIR1 and mCherry reporter preceded by a loxP-flanked STOP cassette (Extended Data Fig. 2a–c). The resulting *Foxp3*[AID] mice exhibited the expected Foxp3 expression pattern (Extended Data Fig. 1c) limited to ZsGreen+ $T_{reg}$ cells, whose suppressive capacity was similar to that of *Foxp3*[GFP] $T_{reg}$ cells[15] (Extended Data Fig. 1d). Upon the addition of indole acetic acid (IAA) to TIR1-expressing *Foxp3*[AID] $T_{reg}$ cells in vitro, AID-fused Foxp3 underwent poly-ubiquitination and proteasomal degradation in a TIR1-dependent manner[16] (Extended Data Figs. 1e and 2d–f). As we found the in vivo performance of the original AID-Foxp3-TIR1 protein degradation system to be suboptimal (Extended Data Fig. 2j,k), we mutated TIR1 phenylalanine 74 to a glycine in *R26*[TIR1] mice using CRISPR-mediated gene editing (Extended Data Fig. 2g–k). Instead of unmodified IAA, the mutant TIR1(F74G) protein recognizes 5-phenyl-IAA (5-ph-IAA)[16]. This improved degradation in *Foxp3*[AID]*R26*[TIR1(F74G)] mice enabled rapid and near-complete in vivo degradation of the Foxp3 protein within 6 h upon 5-ph-IAA administration (Extended Data Fig. 2j,k). The effect persisted for at least 24 h after drug administration, ensuring continuous Foxp3 degradation upon once daily 5-ph-IAA administration (Fig. 1c).

### Foxp3 expression is largely dispensable for preventing autoimmune disease in adulthood

Using continuous in vivo Foxp3 degradation, we investigated its role in $T_{reg}$ cell maintenance and function in adult mice using a side-by-side analysis of *Foxp3*[AID]*R26*[TIR1(F74G)] treated with 5-ph-IAA and *Foxp3*[DTR] mice subjected to $T_{reg}$ cell ablation upon administration of diphtheria toxin (DT) (Fig. 1d). While continuous $T_{reg}$ cell depletion led to flagrant splenomegaly and lymphadenopathy, these manifestations were unexpectedly mild following continuous Foxp3 degradation in *Foxp3*[AID]*R26*[TIR1(F74G)] mice treated with 5-ph-IAA for the same duration (Fig. 1e). Accordingly, DT-mediated $T_{reg}$ cell ablation induced pronounced T, B and myeloid cell activation, whereas Foxp3 degradation only had modest effects (Fig. 1f–h). Most $T_{reg}$ cell-depleted *Foxp3*[DTR] mice succumb to the resulting autoimmune syndrome within 2–3 weeks[17]. In contrast, continuous Foxp3 degradation for 4 weeks did not result in any noticeable clinical manifestations of autoimmune disease with only mildly increased state of immune cell activation observed (Fig. 1f–h and Extended Data Fig. 3a,b). In this regard, hepatitis and liver damage, associated with a marked immune infiltration, elevated serum alanine aminotransferase and diminished albumin in $T_{reg}$ cell-depleted *Foxp3*[DTR] mice, were undetectable in *Foxp3*[AID]*R26*[TIR1(F74G)] mice after 4

weeks of continuous Foxp3 degradation (Fig. 1i–k). Notably, immune cell activation following Foxp3 degradation progressed at a slow pace as minimal changes were observed between 2 and 4 weeks of 5-ph-IAA treatment (Fig. 1f–k and Extended Data Fig. 3b–d).

While our Foxp3 degradation system enabled near-complete Foxp3 degradation, one potential caveat was that minute residual Foxp3 amounts remaining upon 5-ph-IAA treatment (Fig. 1c). To ensure that the latter does not account for the preserved $T_{reg}$ cell functionality after Foxp3 degradation, we generated *Cd4*[creERT2]*Foxp3*[fl/fl] mice in which tamoxifen administration induced deletion of a conditional *Foxp3*[fl] gene. Given the rapid turnover of Foxp3 protein, this approach resulted in a complete loss of its expression shortly after CreER-induced recombination, yet $T_{reg}$ cells retained their identity and the function consistent with the Foxp3 degradation results (Extended Data Fig. 4). Thus, contrary to an absolute requirement of Foxp3± $T_{reg}$ cells for the restraint of fatal autoimmunity, Foxp3 protein in differentiated $T_{reg}$ cells is largely dispensable for their suppressor function.

### Foxp3 loss induces minimal transcriptional and functional changes in mature $T_{reg}$ cells

To gain insights into the mechanisms of the observed retention of mature $T_{reg}$ cell function upon Foxp3 loss, we analyzed its effect on $T_{reg}$ gene expression using single-cell RNA sequencing (scRNA-seq) of fluorescence-activated cell sorting (FACS)-sorted ZsGreen+ cells from the secondary lymphoid organs (SLOs) of *Foxp3*[AID/WT]*R26*[TIR1(F74G)] mice and *Foxp3*[AID/WT]*R26*[WT] controls on days 3 and 7 of continuous 5-ph-IAA treatment (Fig. 2a). To avoid potential effects of mildly increased inflammation in hemizygous *Foxp3*[AID] males following Foxp3 degradation, we performed the experiment in *Foxp3*[AID/WT] heterozygous females, which harbor a mixed population of $T_{reg}$ cells expressing either the *Foxp3*[WT] or *Foxp3*[AID] allele. Upon 5-ph-IAA treatment, the *Foxp3*[WT]-expressing $T_{reg}$ cells remain unaffected and maintain comparable immune tone in experimental and control mice. Our analysis of global ZsGreen+ cell transcriptomes, visualized by Uniform Manifold Approximation and Projection (UMAP), suggested minute differential gene expression on both day 3 and day 7 of induced Foxp3 degradation (Fig. 2b). Similarly, overlaying the UMAP plots of *Foxp3*[AID]*R26*[TIR1(F74G)] $T_{reg}$ cells on days 0, 3 and 7 of 5-ph-IAA treatment revealed minor changes (Extended Data Fig. 5a). To perform a more systematic, quantitative comparison and account for potential transcriptional changes within rare $T_{reg}$ cell subpopulations, we performed Leiden clustering yielding 18 cell clusters (Fig. 2c and Extended Data Fig. 5b). *Foxp3*[AID]*R26*[WT] and *Foxp3*[AID]*R26*[TIR1(F74G)] $T_{reg}$ cells from all three time points were similarly represented in most of these clusters indicative of minimal transcriptional changes across the entire $T_{reg}$ cell population (Fig. 2d). Consistent with these findings, 5-ph-IAA-induced Foxp3 degradation in sorted *Foxp3*[AID]*R26*[TIR1(F74G)] $T_{reg}$ cells did not impact their ability to suppress CD4 T cell proliferation in vitro when compared to similarly treated *Foxp3*[AID]*R26*[WT] $T_{reg}$ cells (Fig. 2e).

Because of its known limitation in capture efficiency, we complemented scRNA-seq analysis with bulk gene expression and chromatin accessibility analyses of resting and activated ZsGreen± cells sorted from the SLOs of *Foxp3*[AID/WT]*R26*[TIR1(F74G)] and *Foxp3*[AID/WT]*R26*[WT] mice on day 7 of 5-ph-IAA treatment (Fig. 2f,g). As a comparison, we examined differential gene expression in cells expressing a *Foxp3*[GFPKO] reporter-null allele[4], which have never expressed Foxp3 protein, versus Foxp3-sufficient *Foxp3*[GFP] $T_{reg}$ cells[12] (Fig. 2h). Consistent with the scRNA-seq analysis, bulk RNA-seq revealed a dramatically reduced number of differentially expressed genes (DEGs) in both resting and activated $T_{reg}$ cells upon Foxp3 protein degradation compared to those in *Foxp3*[GFPKO] cells (Fig. 2i and Supplementary Table 1). Likewise, ATAC-seq analysis of Foxp3-degraded resting and activated *Foxp3*[AID] $T_{reg}$ cells revealed minimal changes in chromatin accessibility, unlike the substantial differences observed between resting and activated *Foxp3*[GFPKO] $T_{reg}$ cells and Foxp3-sufficient *Foxp3*[GFP] $T_{reg}$ cells[12] (Extended Data Fig. 5c).

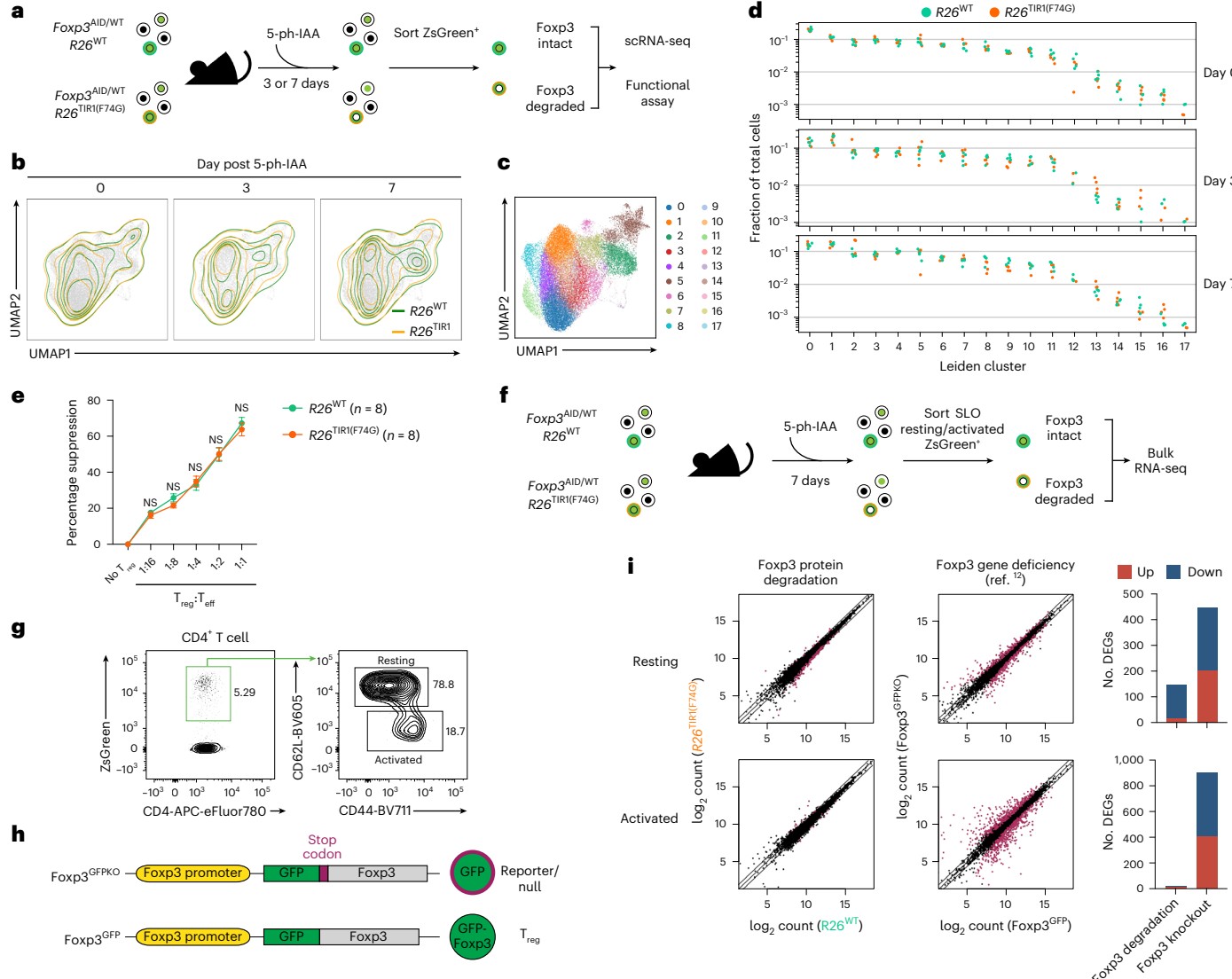

**Fig. 2 | Foxp3 degradation induces minimal gene expression and functional changes in mature T$_{reg}$ cells. a**, Experimental design of scRNA-seq and functional assays. Each genotype and time point consisted of four independent biological replicates. **b**, UMAP visualization of scRNA-seq data from *Foxp3*$^{AID}$*R26*$^{WT}$ and *Foxp3*$^{AID}$*R26*$^{TIR1(F74G)}$ T$_{reg}$ cells before and 3 or 7 days after 5-ph-IAA-induced Foxp3 degradation. **c**, UMAP visualization of the same scRNA-seq data, colored by identified clusters. **d**, Fraction of each cluster within the total pool of *Foxp3*$^{AID}$*R26*$^{WT}$ or *Foxp3*$^{AID}$*R26*$^{TIR1(F74G)}$ T$_{reg}$ cells separated by time point. Each point represents a unique mouse. **e**, In vitro suppression assay of T$_{reg}$ cells sorted from *Foxp3*$^{AID}$*R26*$^{WT}$ and *Foxp3*$^{AID}$*R26*$^{TIR1(F74G)}$ mice after 7 days of in vivo 5-ph-IAA

treatment. 5-ph-IAA was included in culture to sustain Foxp3 degradation. Line graph represents mean ± s.e.m. Data are pooled from two independent experiments and analyzed using two-sided multiple *t*-tests. NS, not significant. **f**, Experimental design of the bulk RNA-seq analysis. Each genotype and time point consisted of three independent biological replicates. **g**, Gating strategy for sorting resting and activated T$_{reg}$ cells. **h**, Schematic comparison of the *Foxp3*$^{GFPKO}$ reporter-null allele and the functional *Foxp3*$^{GFP}$ allele. **i**, Scatter-plots and bar graphs showing the number of DEGs in resting or activated T$_{reg}$ cells caused by Foxp3 protein degradation or genetic Foxp3 deficiency.

Next, we performed additional analyses on the small group of genes whose expression was affected by Foxp3 degradation in our scRNA-seq dataset with activation states defined using the gene scores based on previously identified T$_{reg}$ cell transcriptional signatures[12] (Fig. 3a and Extended Data Fig. 6a,b). We then performed pseudo-bulk differential gene expression analyses for the resting and activated T$_{reg}$ cells. Gene expression changes on day 3 and day 7 post Foxp3 degradation were well correlated, with a larger fold change observed on day 7 indicating time-dependent augmentation of gene expression changes (Fig. 3b). We classified the latter into four groups: genes upregulated ('resting TIR1-up' and 'activated TIR1-up') and downregulated ('resting TIR1-down' and 'activated TIR1-down') upon Foxp3 degradation in resting and activated T$_{reg}$ cells, respectively. A closer examination of Foxp3-regulated transcripts in resting T$_{reg}$ cells showed that the

'resting TIR1-up' genes repressed by Foxp3 were expressed more highly in activated versus resting T$_{reg}$ cells, whereas 'resting TIR1-down' genes induced by Foxp3 showed the opposite pattern (Fig. 3c). This result suggests that in resting T$_{reg}$ cells, Foxp3 enables a 'goldilocks' state of expression of T cell activation-associated genes.

The gene set with statistically significant differential expression caused by Foxp3 degradation in either resting or activated T$_{reg}$ cells included 32 TIR1-up and 38 TIR1-down genes repressed and activated by Foxp3, respectively (Fig. 3d). The latter group included the *Foxp3* gene itself. Indeed, ZsGreen expression, reporting *Foxp3* mRNA levels, showed a slight but statistically significant reduction in resting T$_{reg}$ cells at day 7, consistent with the bulk RNA-seq analysis (Fig. 3e and Extended Data Fig. 6c). Flow cytometric analyses also showed a Foxp3 degradation-induced reduction in CD25(*Il2ra*) expression in both

resting and activated $T_{reg}$ cells (Fig. 3f and Extended Data Fig. 6d,e), whereas a reduction in CD122 (*Il2rb*), OX40 (*Tnfrsf4*), GITR (*Tnfrsf18*) and FR4 (*Izumo1r*) levels was limited to resting $T_{reg}$ cells (Fig. 3f and Extended Data Fig. 6d,e). On the flip side, CD127 (*Il7r*) and TCF1 (*Tcf7*) protein expression was increased in resting $T_{reg}$ cells following Foxp3 degradation, consistent with the observed changes in their transcript levels (Fig. 3g and Extended Data Fig. 6d,f).

### Foxp3 degradation-sensitive genes are enriched for Foxp3 binding

Previous studies of resting and activated *Foxp3*[GFP] $T_{reg}$ and *Foxp3*[GFPKO] $T_{reg}$ 'wannabe' cells identified the overall Foxp3-dependent gene set without distinguishing between potential direct and indirect Foxp3 targets[4,5]. Given the short duration of Foxp3 degradation and the small number of genes impacted by it, we reasoned that 'degradation-sensitive' genes are likely enriched for direct Foxp3 transcriptional targets. To examine this, we grouped genes that were degradation sensitive (*Foxp3*[AID]*R26*[TIR1(F74G)] versus *Foxp3*[AID]*R26*[WT]) in both resting and activated $T_{reg}$ cells based on their *P* values and examined the number of Foxp3 binding sites[18] near the stratified genes in each group (Fig. 4a). As Foxp3 is known to bind predominantly to open chromatin regions and its global genome occupancy is not associated with Foxp3-dependent gene expression or chromatin accessibility changes, we normalized the number of Foxp3 peaks to the number of open chromatin regions surrounding each gene, using previously published Foxp3 CUT&RUN and $T_{reg}$ ATAC-seq datasets[12]. Notably, the top 'TIR1-down' degradation-sensitive genes in resting $T_{reg}$ cells contained significantly more Foxp3 binding sites per gene (Fig. 4a). This observation suggests that genes in this group, such as *Il2ra*, *Lrrc32* and *Il2rb*, known for their role in $T_{reg}$ functionality[19,20], are extensively bound and likely directly regulated by Foxp3 (Extended Data Fig. 7a). Likewise, the top 'TIR1-up' degradation-sensitive genes in resting $T_{reg}$ cells also contained more Foxp3 binding sites per gene (Fig. 4a). Genes in this group, including *Tcf7*, *Id2* and *Sox4*, have been implicated in $T_{reg}$ gene expression and function[21-25] (Extended Data Fig. 7b). We refer to these gene groups as 'Foxp3-activated' and 'Foxp3-repressed' genes, respectively. Of note, when the overall Foxp3-dependent (*Foxp3*[GFPKO] versus *Foxp3*[GFP]) gene set was stratified and analyzed in the same fashion no enrichment for Foxp3 binding was observed, likely because a larger number of indirect Foxp3 target genes obscured small number of direct ones (Fig. 4b). In contrast, similar analyses of Foxp3 degradation-sensitive genes in their activated counterparts did not reveal significant enrichment of Foxp3 binding in any gene groups, suggesting that upon $T_{reg}$ activation direct gene regulation by Foxp3 can be compensated by other T cell activation-dependent transcription factors (Fig. 4a,b). This observation was also consistent with the smaller number of DEGs identified by both bulk and scRNA-seq analyses of Foxp3 degradation in activated $T_{reg}$ cells (Figs. 2i and 3d).

By leveraging our published ChIP-seq data[18], we explored additional features of the 'Foxp3-activated' and 'Foxp3-repressed' genes. Compared to genes not enriched for Foxp3 binding, 'Foxp3-activated' genes showed the highest level of the activating-H3K27Ac and the lowest level of the repressive-H3K27me3 histone modifications, whereas 'Foxp3-repressed' genes showed the opposite pattern (Fig. 4c).

In addition, motif enrichment analysis of Foxp3-bound ATAC-seq peaks near 'Foxp3-activated' genes revealed a pronounced enrichment for STAT-binding motifs (Fig. 4d) consistent with the inclusion of both *Il2ra* and *Il2rb* within this gene set.

### Foxp3 degradation effects are generally conserved across tissues

To assess Foxp3 degradation effects on the mature $T_{reg}$ cell transcriptional program across both nonlymphoid versus lymphoid tissues under steady state, we performed scRNA-seq analysis of ZsGreen[+] cells isolated from the SLOs, lung, liver and large intestine lamina propria (LILP) of *Foxp3*[AID/WT]*R26*[TIR1(F74G)] and *Foxp3*[AID/WT]*R26*[WT] mice on day 7 of continuous 5-ph-IAA treatment (Extended Data Fig. 8a). For all tissues, *Foxp3*[AID]*R26*[TIR1(F74G)] and *Foxp3*[AID]*R26*[WT] $T_{reg}$ transcriptome distributions were highly overlapping when visualized using a UMAP embedding (Extended Data Fig. 8b). To account for varying activation states in different tissues, we further classified cells as resting or activated as in Fig. 3a (Extended Data Fig. 8c). Next, we identified significant DEGs in *Foxp3*[AID]*R26*[TIR1(F74G)] versus *Foxp3*[AID]*R26*[WT] $T_{reg}$ cells across all tissues and correlated log fold changes of these DEGs between tissues with higher correlation corresponded to a higher degree of similarity of Foxp3 degradation-induced transcriptional effects. Consistent with our previous findings, resting $T_{reg}$ cells had greater numbers of DEGs compared to activated ones in each tissue (Extended Data Fig. 8d); accordingly, hierarchical clustering of log fold change correlations revealed that resting and activated cells clustered separately irrespective of tissue origin (Extended Data Fig. 8e). Among resting cells, the effects of Foxp3 degradation were largely consistent between tissues as evidenced by highly correlated transcriptional changes (Pearson $r = 0.83-0.91$). Of note, DEGs in nonlymphoid tissue-activated $T_{reg}$ cells were less correlated with those in SLOs (Pearson $r = 0.53-0.69$). *K*-means clustering of the DEGs from all tissues, based on log fold changes, revealed gene clusters with distinct patterns of regulation across tissues and activation states (Extended Data Fig. 8f,g). A few clusters exhibited tissue-specific alterations, most prominently in LILP-activated cells (clusters 5 and 7). Thus, Foxp3 degradation-induced transcriptional effects, while highly conserved across tissues in resting $T_{reg}$ cells, were more variable across tissues in activated ones, particularly in the LILP, a tissue enriched for highly activated and devoid of resting $T_{reg}$ cells. The lack of enrichment of Foxp3 binding sites near degradation-sensitive genes in activated cells (Fig. 4a) was consistent with the supposition that tissue-specific factors may contribute to distinct transcriptional effects of Foxp3 degradation in activated cells.

The LILP $T_{reg}$ cell population encompasses peripherally (p$T_{reg}$) and thymically generated (t$T_{reg}$) $T_{reg}$ cells[26]. Subclustering of LILP cells identified a small actively proliferating cluster, a *Rorc* expressing 'p$T_{reg}$' cluster alongside *Ikzf2*[intermediate] and *Ikzf2*[high] 't$T_{reg}$' clusters enriched for lymphoid tissue-associated genes and *Gata3* expression, respectively (Extended Data Fig. 8h,i). We refer to these populations as p$T_{reg}$, lymphoid tissue t$T_{reg}$ (LT-t$T_{reg}$) and nonlymphoid tissue t$T_{reg}$ (NLT-t$T_{reg}$), respectively, consistent with published LILP $T_{reg}$ transcriptomes[27]. Comparing the DEGs induced by Foxp3 degradation within each cluster, the p$T_{reg}$ cluster had

---

**Fig. 3 | Foxp3 degradation in mature $T_{reg}$ cells induces expression changes in a small set of genes. a**, $T_{reg}$ cells from the scRNA-seq dataset were classified as resting or activated based on exceeding the threshold for resting or activated gene signature scores and were subsequently analyzed. **b**, Scatter-plot showing the correlation of gene expression changes induced by Foxp3 degradation at day 3 and day 7 in resting and activated $T_{reg}$ cells. FC, fold change. **c**, UMAP visualization of resting and activated $T_{reg}$ cells colored by gene signature scores for the 'TIR1-up' and 'TIR1-down' gene sets, up- and downregulated upon Foxp3 degradation, respectively. **d**, Dot plot summarizing statistically significant DEGs in resting or activated $T_{reg}$ cells following 3 or 7 days of 5-ph-IAA-induced Foxp3 degradation. The color represents the log$_2$ fold change of *R26*[TIR1(F74G)] versus *R26*[WT] and the size represents the Benjamini–Hochberg adjusted *P* value of the differential expression test. **e**, Flow cytometry analysis of Foxp3 protein and

mRNA levels (reported by ZsGreen) in *Foxp3*[AID] $T_{reg}$ cells from heterozygous *Foxp3*[AID/WT]*R26*[WT] and *Foxp3*[AID/WT]*R26*[TIR1(F74G)] females after 7 days of Foxp3 degradation. Scatter-plots represent mean ± s.e.m. Data are pooled from two independent experiments and analyzed using a two-way ANOVA. pLN, peripheral lymph nodes. **f**, Flow cytometry analysis of CD25, CD122, OX40, GITR and FR4 protein levels in *Foxp3*[AID] $T_{reg}$ cells from heterozygous *Foxp3*[AID/WT]*R26*[WT] and *Foxp3*[AID/WT]*R26*[TIR1(F74G)] females after 7 days of Foxp3 degradation. **g**, Flow cytometry analysis of CD127 and TCF1 protein levels in *Foxp3*[AID] $T_{reg}$ cells from heterozygous *Foxp3*[AID/WT]*R26*[WT] and *Foxp3*[AID/WT]*R26*[TIR1(F74G)] females after 7 days of Foxp3 degradation. Scatter-plots represent mean ± s.e.m. (**f,g**). Each point represents a unique mouse. Data are pooled from two independent experiments and analyzed with two-sided multiple *t*-tests.

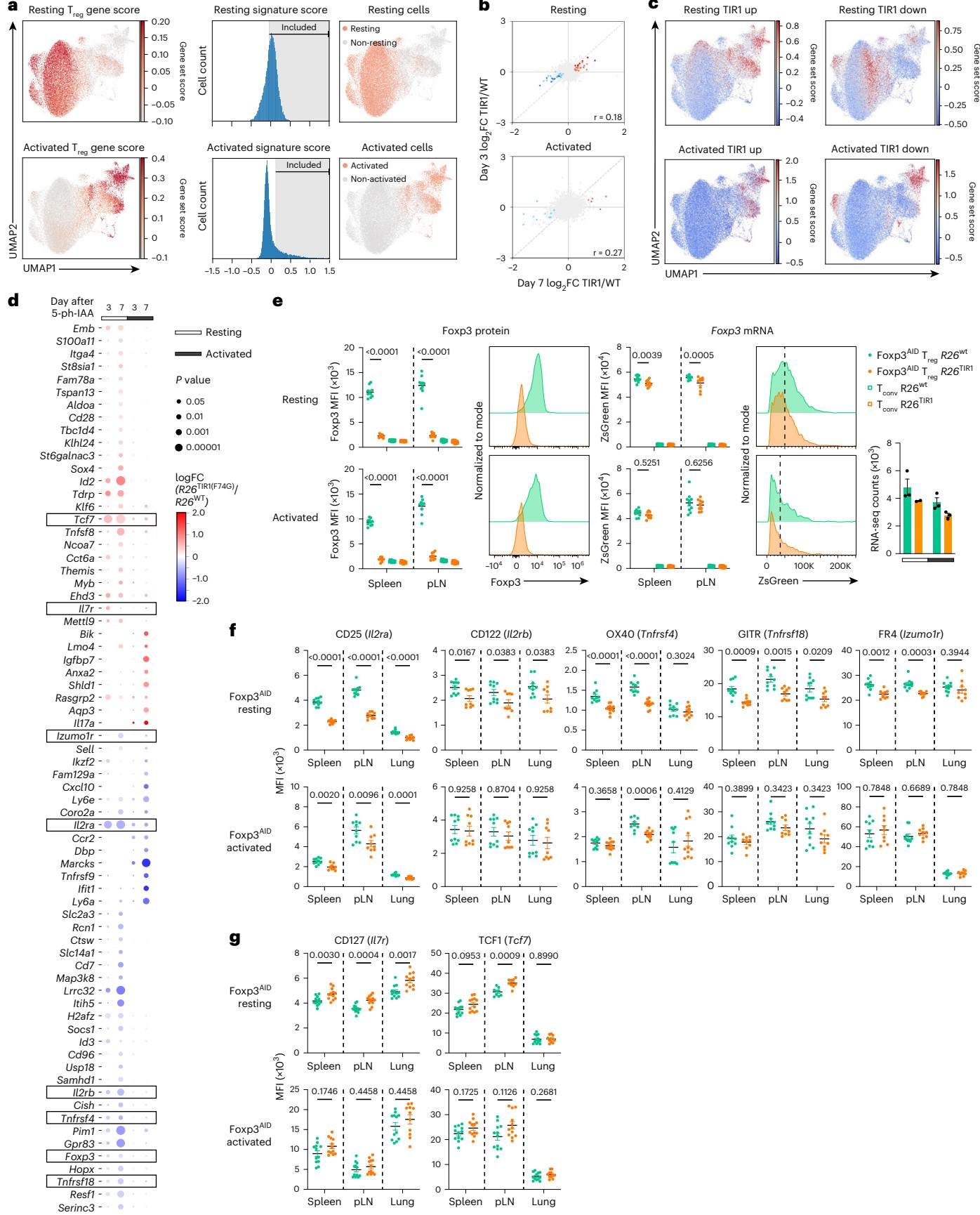

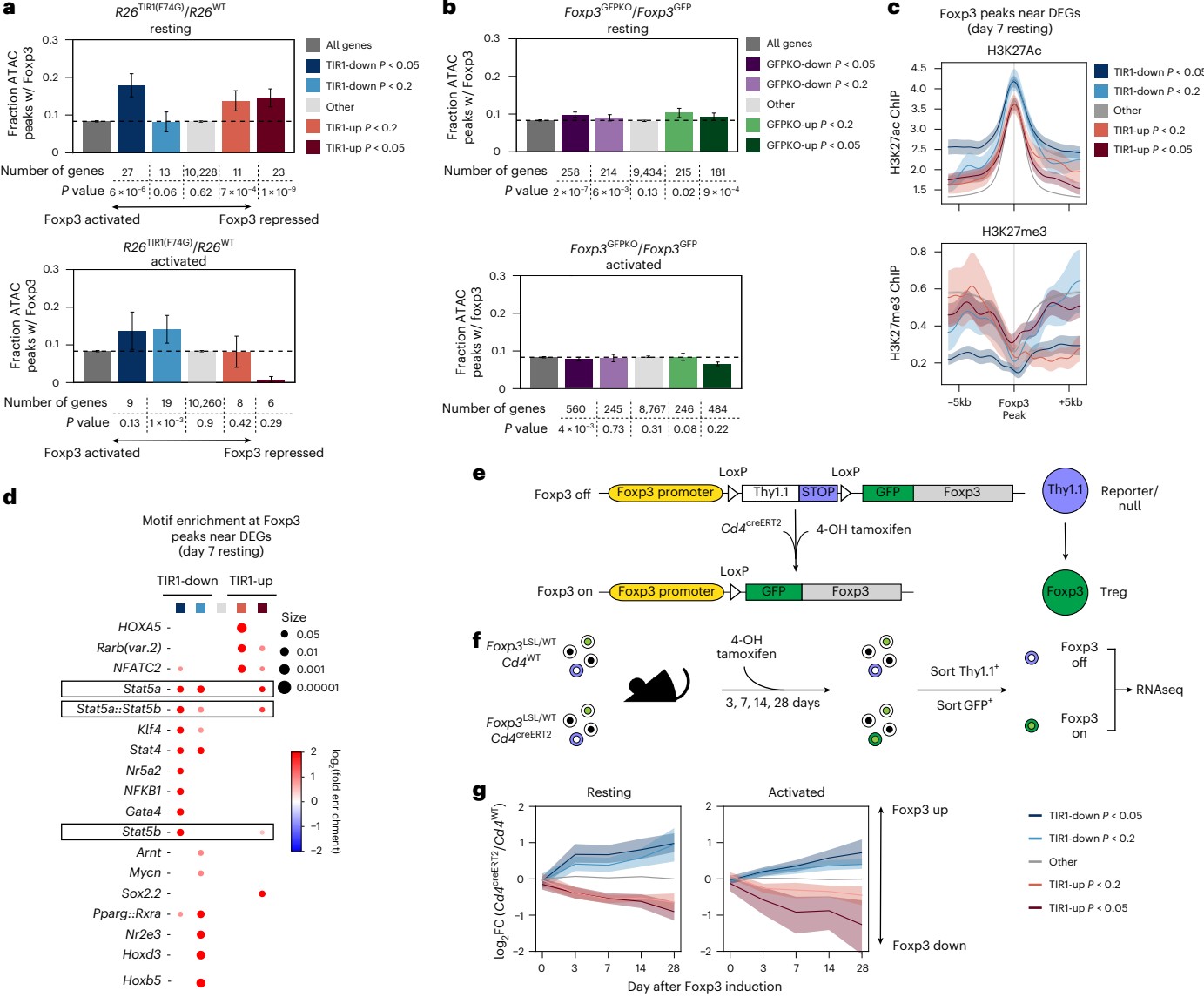

**Fig. 4 | Foxp3 degradation-sensitive genes in mature T_reg cells are enriched for Foxp3 binding. a**, Bar graphs showing the proportion of ATAC-seq peaks near Foxp3 degradation-induced DEGs bound by Foxp3 (ref. 19). Genes are stratified by statistical significance (*P* values) in resting and activated T_reg cells. Bars represent mean ± s.e.m. Data were analyzed using a Mann–Whitney *U*-test. **b**, Bar graphs showing the proportion of ATAC-seq peaks near Foxp3-dependent DEGs bound by Foxp3. Genes are stratified by *P* values in resting and activated T_reg cells. Bars represent mean ± s.e.m. Data were analyzed using a Mann–Whitney *U*-test. **c**, H3K27Ac and H3K27me3 ChIP-seq signals[19] at Foxp3-bound ATAC-seq peaks near Foxp3 degradation-induced DEGs. Line graph represents mean ± s.e.m. Data were

analyzed using a Mann–Whitney *U*-test. **d**, Dot plot showing transcription factor motif enrichment within Foxp3-bound regions near Foxp3 degradation-induced DEGs. **e**, Schematic diagram illustrating the 'on' and 'off' states of the reversible reporter-null *Foxp3*^LSL allele. **f**, Experimental design of the gain-of-function experiment to induce Foxp3 expression in T_reg 'wannabe' cells. Each genotype and time point consisted of two independent biological replicates. **g**, Line graph depicting gene expression changes of TIR1-up or TIR1-down genes below a specific *P* value cutoff across different time points following Foxp3 induction in T_reg 'wannabe' cells. The line and shading represent the mean ± s.e.m. Data were analyzed using a Mann–Whitney *U*-test.

---

the highest number of DEGs (Extended Data Fig. 8j); however, compared to the pT_reg cluster, the NLT-tT_reg cluster had a significantly larger magnitude of differential expression of degradation-sensitive Foxp3-repressed and activated genes (Extended Data Fig. 8k). Furthermore, the Foxp3 degradation-induced DEGs poorly correlated (Pearson *r* = 0.38) between the NLT-tT_reg and pT_reg cells (Extended Data Fig. 8l). These results are consistent with findings that colonic pT_reg cells maintain their fitness and exert some of their suppressive functions independently of Foxp3.

**Long timescale of establishment of Foxp3-dependent gene program**

To complement Foxp3-degradation-based 'loss-of-function' studies, we employed a 'gain-of-function' approach using a reversible

*Foxp3*^loxP-Thy1.1-STOP-loxP-GFP reporter-null allele (*Foxp3*^LSL)[7]. The *Foxp3*^LSL allele harbors a loxP site-flanked Thy1.1 reporter followed by a STOP cassette and a *Foxp3*^GFP reporter. In *Foxp3*^LSL mice, Thy1.1 reporter marks T_reg 'wannabe' cells with the transcriptionally active *Foxp3* locus yet lacking Foxp3 expression similar to the GFP+ cells in *Foxp3*^GFPKO mice. 4-hydroxytamoxifen (4-OHT) treatment of *Cd4*^creERT2*Foxp3*^LSL mice led to the excision of the STOP cassette and punctual induction of the Foxp3 protein, converting Foxp3−Thy1.1+ T_reg 'wannabe' cells into fully functional Foxp3-expressing GFP+ T_reg cells (Fig. 4e). *Foxp3*^LSL/WT female heterozygous mice are healthy as they harbor both functional Foxp3-sufficient T_reg cells and T_reg 'wannabe' cells expressing *Foxp3*^WT and *Foxp3*^LSL allele, respectively. Thus, we sought to investigate the temporal dynamics of the emerging Foxp3-dependent transcriptional

features upon acquisition of Foxp3 expression by $T_{reg}$ 'wannabe' cells in 4-OHT-treated female heterozygous $Cd4^{creERT2}Foxp3^{LSL/WT}$ mice. Resting and activated Foxp3+GFP+ cells and control Foxp3−GFP−Thy1.1+ cells with a matching activation state were sorted from the $Cd4^{creERT-2}Foxp3^{LSL/WT}$ and $Cd4^{WT}Foxp3^{LSL/WT}$ littermates, respectively, on days 3, 7, 14 and 28 after single 4-OHT administration and subjected to RNA-seq analysis (Fig. 4f and Extended Data Fig. 7c). The top Foxp3 degradation-sensitive 'TIR1-down' and 'TIR1-up' genes showed time-dependent increases and decreases in their expression in both resting and activated Foxp3+GFP+Thy1.1− cells in comparison to time- and activation-state-matched Foxp3−GFP−Thy1.1+ 'wannabe' controls, respectively, consistent with a likely direct role for Foxp3 in regulating their expression (Fig. 4g). Further analysis of the pace at which Foxp3 installation drives transcriptional changes in $T_{reg}$ 'wannabe' cells toward a bona fide $T_{reg}$ transcriptome revealed an unexpectedly prolonged timeline of approximately 2 weeks. While it remains formally plausible that Foxp3-dependent transcriptional programs are established more rapidly in normally differentiating $T_{reg}$ cells, these observations suggest that the establishment of the $T_{reg}$ cell transcriptional program and functionality is critically dependent on Foxp3 during $T_{reg}$ cell maturation. In contrast, fully differentiated $T_{reg}$ cells in healthy adult mice may not rely on Foxp3 to the same extent. Moreover, the slow kinetics of the Foxp3-dependent gene program acquisition highlights the necessity for direct Foxp3 targets to act in *trans* to regulate downstream, indirect Foxp3-dependent genes.

## A requirement for Foxp3 during $T_{reg}$ cell maturation

To test whether Foxp3 is essential for the acquisition of the $T_{reg}$ cell-specific transcriptional program and function during thymic $T_{reg}$ cell differentiation and peripheral maturation, we employed two complementary approaches to assess the effects of induced Foxp3 protein degradation in developing $T_{reg}$ cells. First, we investigated Foxp3 degradation-induced gene expression changes in developing $T_{reg}$ cells in the thymus. Following 7 days of 5-ph-IAA-mediated Foxp3 degradation in $Foxp3^{AID/WT}R26^{TIR1(F74G)}$ mice, we performed bulk RNA-seq analysis of developing ZsGreen+ CD73low $T_{reg}$ cells isolated from the thymus (Fig. 5a,b). Differential CD73 expression was used to discern recently generated nascent ZsGreen+CD73low CD62Lhigh $T_{reg}$ cells from recirculating CD73high $T_{reg}$ cells entering the thymus from the periphery[28]. We then compared the expression of Foxp3 degradation-sensitive and Foxp3-dependent ($Foxp3^{GFPKO}$ versus $Foxp3^{GFP}$) genes across $T_{reg}$ developmental stages. The number of DEGs resulting from Foxp3 degradation was the highest in developing thymic $T_{reg}$ cells followed by resting $T_{reg}$ cells, whereas activated $T_{reg}$ cells exhibited the lowest number (Fig. 5c). These data suggest that the transcriptional program of developing $T_{reg}$ cells is markedly more vulnerable to Foxp3 loss in comparison to mature $T_{reg}$ cells, with the latter becoming even less dependent on continuous Foxp3 expression as they become activated. This observation

was consistent with the absence of significant enrichment for Foxp3 binding in Foxp3 degradation-sensitive gene loci in activated $T_{reg}$ cells. Notably, Foxp3 degradation-sensitive genes and Foxp3-dependent genes exhibited the strongest correlation in developing thymic $T_{reg}$ cells, with this correlation progressively decreasing as $T_{reg}$ cells mature, reaching its lowest values in activated $T_{reg}$ cells (Fig. 5d,e). This trend persisted even among Foxp3-bound genes (Extended Data Fig. 9a). The declining correlation suggests that while Foxp3 deficiency closely mirrors the effects of Foxp3 degradation in recently generated $T_{reg}$ cells, sustained Foxp3 loss in mature $T_{reg}$ cells, particularly those with a history of activation, may lead to secondary transcriptional effects beyond the primary Foxp3-regulated program.

Meta-analysis of previously published scRNA-seq of thymic Foxp3+ $T_{reg}$ cells and their progenitors[29] and our datasets showed that the 'TIR1-up' (Foxp3-repressed) and 'TIR1-down' (Foxp3-activated) degradation-sensitive gene signatures negatively and positively correlate with Foxp3 expression, respectively, in a dose-dependent manner (Fig. 5f). Notably, similar metacell analysis of Foxp3-degradation-induced transcriptional changes in resting and activated $T_{reg}$ cells revealed a correlation between Foxp3 dosage and the 'TIR1-down' signature only (Extended Data Fig. 9b). In contrast, the 'TIR1-up' gene set showed no correlation with Foxp3 expression level. These findings further support a shift in the role of Foxp3 in gene regulation as $T_{reg}$ cells mature, particularly in its function as a transcriptional repressor.

In adult mice, thymic output contributes minimally to the peripheral pool of differentiated $T_{reg}$ cells, which is maintained by their self-renewal and whose transcriptional program and functionality we found resilient to the Foxp3 loss[7,9,30] (Figs. 1 and 2). In neonatal mice, Foxp3+ cells first appear among CD4SP thymocytes on days 2–3 after birth; thymic $T_{reg}$ cell output continues to steadily increase until day 21 with recently generated $T_{reg}$ cells accounting for the bulk of the $T_{reg}$ peripheral pool[15]. Thus, we tested the requirement for Foxp3 expression in the suppressor function of early-life $T_{reg}$ cells by treating neonatal $Foxp3^{AID}R26^{TIR1(F74G)}$ and $Foxp3^{AID}R26^{WT}$ control mice with 5-ph-IAA daily for 2 weeks, starting from day 1 after birth (Fig. 5g). Contrary to adults, Foxp3 degradation in neonates led to severe autoimmune disease featuring pronounced T cell activation (Fig. 5h and Extended Data Fig. 9c), myeloproliferation (Fig. 5i and Extended Data Fig. 9d), and tissue inflammation (Fig. 5j,k and Extended Data Fig. 9e) similar to those in Foxp3-deficient $Foxp3^{GFPKO}$ mice indicative of a loss of $T_{reg}$ function. The latter was confirmed by the lack of in vitro suppressor capacity of ZsGreen+ cells isolated from 5-ph-IAA-treated $Foxp3^{AID}R26^{TIR1(F74G)}$ neonates (Fig. 5l). Consistently, RNA-seq analysis of neonatal $Foxp3^{AID}R26^{TIR1(F74G)}$ and $Foxp3^{AID}R26^{WT}$ $T_{reg}$ cells subjected to 7 days of in vivo Foxp3 degradation revealed hundreds of up- and downregulated genes far exceeding the number of DEGs resulting from Foxp3 degradation induced in

**Fig. 5 | Foxp3 is preferentially required for regulation of gene expression during early $T_{reg}$ cell differentiation. a**, Experimental design for transcriptional profiling of developing thymic $T_{reg}$ cells. Each genotype consisted of three independent biological replicates. **b**, Gating strategy used to sort CD73− nascent thymic $T_{reg}$ cells. **c**, Bar graph comparing the number of Foxp3 degradation-induced DEGs in thymic, resting and activated $T_{reg}$ cells from $Foxp3^{AID/WT}$ mice. **d**, Pearson correlation between Foxp3 degradation-induced and Foxp3-dependent DEGs in thymic, resting and activated $T_{reg}$ cells. **e**, Scatter-plot and cumulative distribution function (CDF) plots comparing Foxp3 degradation-induced and Foxp3-dependent DEGs across the three $T_{reg}$ populations. Data were analyzed using a Mann–Whitney *U*-test. **f**, Metacell analysis of thymocyte scRNA-seq data[30] correlating UMI-normalized Foxp3 expression levels in each metacell with the expression of TIR1-up and TIR1-down gene signatures identified in **a–c**. UMAP plots are colored by scaled expression levels of TIR1-up, TIR1-down and UMI-normalized counts of Foxp3. Dashed red line depicts line of best fit. Correlations and corresponding *P* values were calculated with Pearson correlation over all genes. **g**, Experimental design for in vivo Foxp3 degradation

in 1-day-old neonatal $Foxp3^{AID}$ mice and adult $Foxp3^{AID}$ mice. **h**, CD4+ and CD8+ T cell activation in adult and neonatal $Foxp3^{AID}$ mice following Foxp3 degradation. **i**, Expansion of eosinophils and neutrophils in adult and neonatal $Foxp3^{AID}$ mice after Foxp3 degradation. **j,k**, Representative H&E staining (**j**) and histology scores of liver inflammation (**k**) in neonatal $Foxp3^{AID}$ mice following Foxp3 degradation. **l**, In vitro suppression assay of $T_{reg}$ cells sorted from $Foxp3^{AID}R26^{WT}$ and $Foxp3^{AID}R26^{TIR1(F74G)}$ neonatal mice after 7 days of in vivo 5-ph-IAA administration. 5-ph-IAA was also included in culture to maintain Foxp3 degradation. Each point represents a unique mouse (**h–l**). Data are pooled from two independent experiments. Scatter-plots represent mean ± s.e.m. Data were analyzed using a one-way ANOVA. **m**, Bar graphs summarizing the number of Foxp3 degradation-induced DEGs in $T_{reg}$ cells from neonatal and adult $Foxp3^{AID/y}$ mice after 14 days of Foxp3 degradation. **n**, Scatter-plot correlating gene expression changes induced by Foxp3 degradation and Foxp3 gene deficiency[7] in neonatal mice. **o**, Scatter-plots comparing gene expression changes induced by 7 days of Foxp3 degradation in neonatal $T_{reg}$ cells to those in adult thymic, resting and activated $T_{reg}$ cells from $Foxp3^{AID/WT}$ mice.

mature T$_{reg}$ cells in adults (Figs. 3 and 5m, Extended Data Fig. 9f and Supplementary Table 1). Foxp3 degradation-induced DEGs in early-life T$_{reg}$ cells showed strong correlation with DEGs observed in Foxp3⁻ T$_{reg}$ 'wannabes' from *Foxp3*$^{LSL}$ neonates versus Foxp3⁺ T$_{reg}$ cells from

*Foxp3*$^{DTR}$ controls, confirming that the loss of Foxp3 in recently generated T$_{reg}$ cells phenocopies *Foxp3* genetic deficiency (Fig. 5n). Among all maturation stages of adult T$_{reg}$ cells, Foxp3 degradation-induced DEGs in adult thymic T$_{reg}$ cells showed the highest concordance with those in

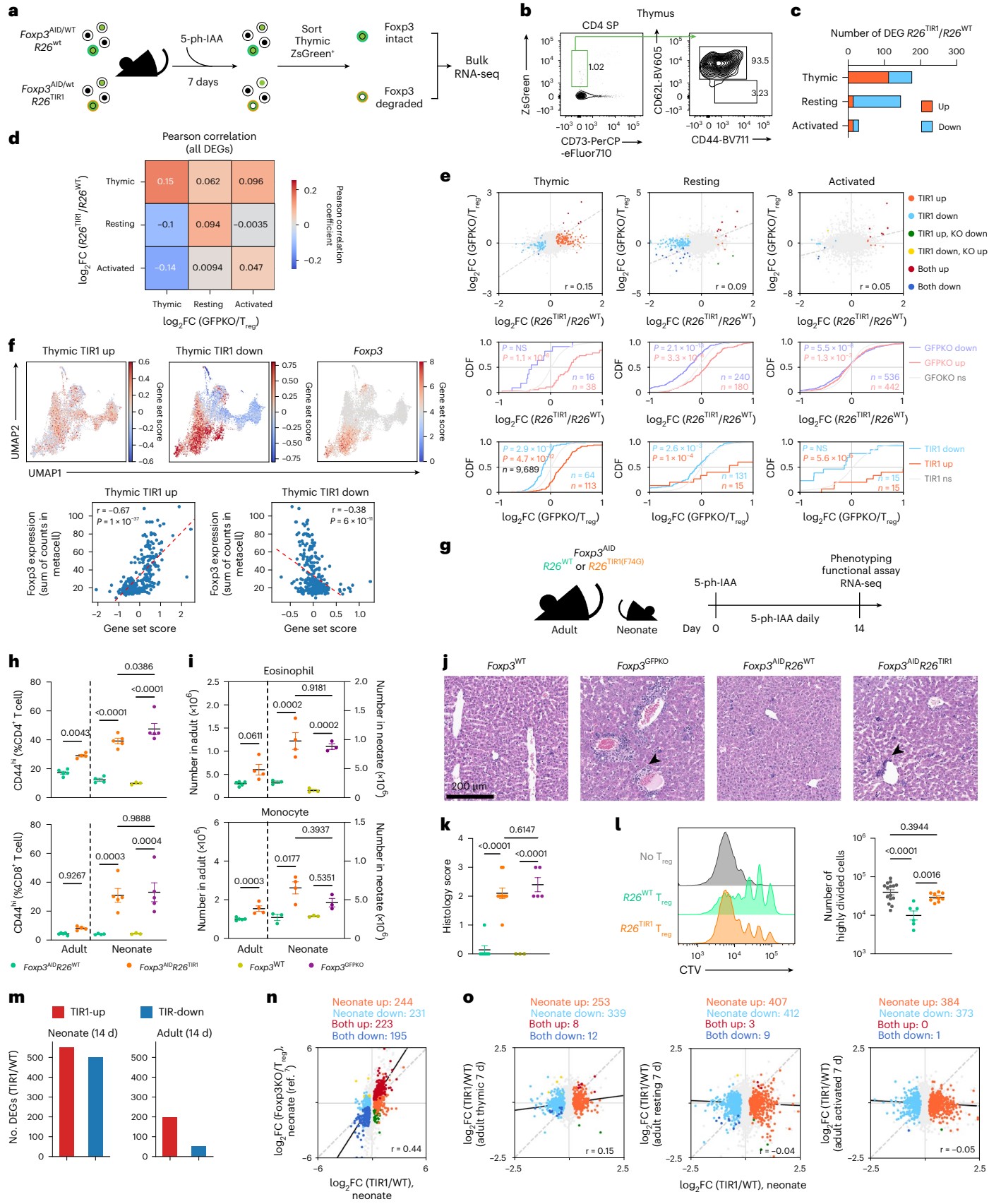

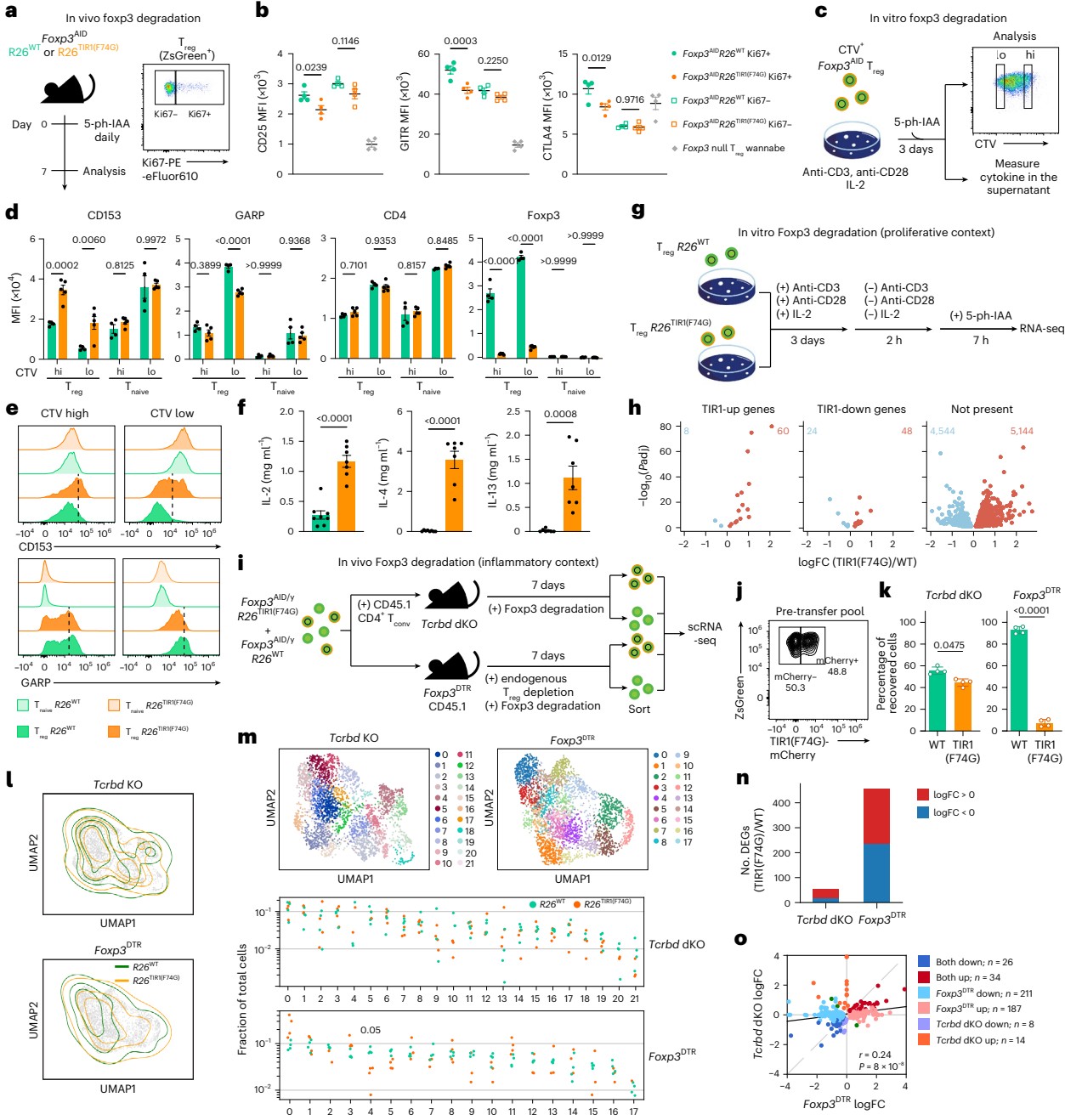

**Fig. 6 | Proliferation and inflammation increase T_reg cell sensitivity to Foxp3 degradation. a**, Experimental design of proliferating T_reg analysis in vivo. **b**, Flow cytometry analysis of CD25, GITR and CTLA4 protein levels in dividing versus nondividing T_reg cells following 7 days of in vivo Foxp3 degradation, in comparison to Foxp3-deficient T_reg 'wannabe' cells. Scatter-plots represent mean ± s.e.m. Each point represents a unique mouse. Data are representative of two independent experiments and were analyzed using a one-way ANOVA. **c**, Experimental design of proliferating T_reg cell analysis in vitro. **d,e**, Combined data (**d**) and representative plots (**e**) showing CD153, GARP, CD4 and Foxp3 protein levels in lowly and highly dividing T_reg cells. Similarly treated naive CD4 T cells serve as Foxp3⁻ controls. Bar graphs represent mean ± s.e.m. Each point represents cells from a unique mouse. Data are representative of two independent experiments and were analyzed using a two-way ANOVA. **f**, IL-2, IL-4 and IL-13 concentrations in the supernatant of in vitro proliferating T_reg assay. Bar graphs represent mean ± s.e.m. Each point represents cells from a unique mouse. Data are pooled from two independent experiments and were analyzed using a two-tailed t-test. **g**, Experimental design of proliferating T_reg cell analysis in vitro. Experiment consisted of five technical replicates per genotype. **h**, Volcano plots of DEGs from **g** separated by their presence among all up- or

downregulated genes in identified in Fig. 3a. Numbers of up- or downregulated genes are labeled in each plot. **i**, Experimental design of inflammatory T_reg cell analysis in vivo. Experiment consisted of four biological replicates per condition (*Tcrbd* double knockout (dKO) or *Foxp3*^DTR). **j**, Flow cytometric analysis of the transferred T_reg cell mixture in **i**. **k**, Recovery of cells in each condition stratified by genotype (mCherry⁺ for *Foxp3*^AID*R26*^TIR1(F74G) or mCherry⁻ for *Foxp3*^AID*R26*^WT) as identified in the scRNA-seq data. Each point represents a unique mouse. Bar graphs represent mean ± s.d. Data were analyzed using a paired two-tailed t-test. **l**, Density contour plots of *R26*^WT and *R26*^TIR1(F74G) overlaid on UMAP embeddings of the scRNA-seq data. **m**, Leiden clustering of gene expression data visualized on UMAP embedding for T_reg cells from each condition. Clustering was performed independently for each condition using the same resolution value. Fraction of each cluster within the total pool of *R26*^WT and *R26*^TIR1(F74G) T_reg cells separated by condition. Within each genotype, each point represents a unique mouse. Data were analyzed using a paired two-tailed t-test. **n**, Number of DEGs in each condition, colored by up- or downregulation. **o**, Scatter-plot of log₂ fold changes of DEGs between *Tcrbd* KO and *Foxp3*^DTR conditions. Points are colored by their direction and populations in which they are altered.

neonates, suggesting a similarity in terms of Foxp3 dependence of their transcriptional programs. In contrast, Foxp3 degradation-induced DEGs in adult resting and activated $T_{reg}$ cells showed no such similarity to the neonatal Foxp3-dependent gene expression features (Fig. 5o). These results suggest that in early life, persistent Foxp3 expression is required for the establishment of a stable gene regulatory network and functionality in recently generated $T_{reg}$ cells, likely by acting on its few direct targets and through continuous enforcement of initially unstable feed-forward regulation of indirect targets via intermediates acting in *trans*[12].

## Proliferation and severe inflammation sensitize $T_{reg}$ cell transcriptome to Foxp3 loss

The observed dispensability of Foxp3 for the function of differentiated $T_{reg}$ cells stood in a sharp contrast with our early finding of a loss of $T_{reg}$ cell function upon ablation of a conditional Foxp3 allele in differentiated $T_{reg}$ cells via Cre[13]. While $T_{reg}$ cells residing in lymphoreplete healthy mice undergo a slow turnover, they undergo pronounced proliferation and activation during in vitro retroviral transduction and transfer into lymphopenic settings employed in these studies. Therefore, Foxp3 expression may be needed to maintain the $T_{reg}$-specific transcriptional program in robustly proliferative cells. To test this supposition, we first performed flow cytometric analysis of $Foxp3^{AID}$ $T_{reg}$ cells following 7 days of in vivo Foxp3 degradation, after parsing them into proliferative and nonproliferative cells on the basis of Ki67 expression (Fig. 6a). While the overall phenotypic shift in Foxp3-degraded versus -replete $T_{reg}$ cells was modest, the proliferating Ki67+ subset accounted for most changes in CD25, GITR and CTLA4 protein levels, whereas the Ki67− $T_{reg}$ subset underwent little change (Fig. 6b), consistent with the above idea. Next, we performed phenotypic analysis of cell trace violet (CTV)-labeled $Foxp3^{AID}$ $T_{reg}$ cells stimulated to proliferate in vitro with anti-CD3 and anti-CD28 antibodies, alongside 5-ph-IAA treatment to induce Foxp3 degradation (Fig. 6c). The highly divided (CTV$^{low}$) cells showed a greater difference in $T_{reg}$ cell markers encoded by Foxp3 degradation-sensitive genes such as CD153 (*Tnfsf8*) and GARP (*Lrrc32*) compared to their lowly divided (CTV$^{high}$) counterparts (Fig. 6d,e and Extended Data Fig. 6d). Expression of CD4, serving as a control, was unaffected by Foxp3 degradation regardless of cell division (Fig. 6d). Moreover, $T_{reg}$ cells subjected to Foxp3 degradation for 72 h secreted more Foxp3-repressed proinflammatory cytokines, including IL-2, IL-4 and IL-13 (Fig. 6f and Extended Data Fig. 6d). To isolate the transcriptional effects of Foxp3 degradation in a highly proliferative context, we activated and expanded $Foxp3^{AID}R26^{TIR1(F74G)}$ and $Foxp3^{AID}R26^{WT}$ $T_{reg}$ cells in vitro for 3 days, rested them in the absence of stimulation for 2 h, treated them with 5-ph-IAA for 7 h and performed bulk RNA-seq (Fig. 6g). Using this approach, we assessed the effects of Foxp3 loss in robustly dividing cells minimizing the effects of T cell receptor (TCR) stimulation and inflammatory cues, including Foxp3 degradation-induced inflammatory cytokine production. While some of the most highly significant DEGs included those identified in mature $T_{reg}$ cells upon Foxp3 degradation in vivo, many were not identified in other experimental settings (Figs. 3a and 6h). Of note, as activated $T_{reg}$ cells in other contexts consistently had fewer DEGs than resting cells (Figs. 2i and 3b and Extended Data Fig. 8d), the observed changes are unlikely to be driven solely by previous TCR-induced activation. Overall, these data are consistent with the notion of a heightened dependency of $T_{reg}$ cells for the maintenance of their transcriptional features on Foxp3 during cell division. As neonatal $T_{reg}$ cells were markedly more proliferative compared to adult cells (Extended Data Fig. 9g,h), their enhanced proliferation together with other potential factors could contribute to their heightened sensitivity to Foxp3 degradation.

Given the upregulation of Foxp3 expression by $T_{reg}$ cells and their proliferation in inflammatory settings[7], we surmised that severe inflammation coupled to cell division may make mature $T_{reg}$ cells vulnerable to the loss of Foxp3. Thus, we adoptively transferred $Foxp3^{AID}R26^{TIR1(F74G)}$

and $Foxp3^{AID}R26^{WT}$ $T_{reg}$ cells mixed at a 1:1 ratio into $Foxp3^{DTR}$ recipients subjected to DT-mediated host $T_{reg}$ cell depletion, which creates severe polytypic inflammation. As a control, we assessed the transcriptional effects of Foxp3 degradation upon short-term transfers of the same $T_{reg}$ cell mixture into T cell-deficient mice, which afford lymphopenia-induced $T_{reg}$ cell proliferation in a minimal inflammatory setting. After 7 days of continuous Foxp3 degradation, sorted mCherry+ $Foxp3^{AID}R26^{TIR1(F74G)}$ and mCherry− $Foxp3^{AID}R26^{WT}$ $T_{reg}$ cells were subjected to scRNA-seq analysis (Fig. 6i,j). We observed severely compromised fitness of Foxp3-deprived $Foxp3^{AID}R26^{TIR1(F74G)}$ versus control Foxp3-sufficient $Foxp3^{AID}R26^{WT}$ $T_{reg}$ cells in inflammatory settings reflected in their strongly biased ratios in the $Foxp3^{DTR}$ recipients; however, in T cell-deficient recipients $Foxp3^{AID}R26^{TIR1(F74G)}$ $T_{reg}$ cells were only mildly outcompeted (Fig. 6k). At the gene expression level, UMAP visualization showed that $Foxp3^{AID}R26^{TIR1(F74G)}$ and $Foxp3^{AID}R26^{WT}$ $T_{reg}$ transcriptomes were largely overlapping (Fig. 6l). Fine-grained Leiden clustering showed similar representation of cells of each genotype from T cell-deficient recipients in all the clusters; however, the frequencies of $Foxp3^{AID}R26^{TIR1(F74G)}$ $T_{reg}$ cells from $Foxp3^{DTR}$ recipients were significantly reduced in cluster 4, whereas they were elevated in cluster 11 (Fig. 6m). Accordingly, many more DEGs in $Foxp3^{AID}R26^{TIR1(F74G)}$ versus $Foxp3^{AID}R26^{WT}$ cells were observed in severe inflammatory versus lymphopenic settings of $Foxp3^{DTR}$ and $Tcrb^{−/−}Tcrd^{−/−}$ recipients, respectively (Fig. 6n,o and Supplementary Table 1). These results suggest that the Foxp3-dependent gene regulatory network resilient to Foxp3 loss in mature $T_{reg}$ cells loses its stability under severe inflammatory conditions.

## Foxp3 degradation-induced tumor shrinkage with minimal adverse effects

Next, we asked whether the resilient state of $T_{reg}$ cell transcriptional and functional program can be lost in a disease state associated with high $T_{reg}$ turnover rates. Solid organ tumors are highly enriched for activated $T_{reg}$ cells[31,32]. Using both Ki67 staining and 5-ethynyl-2′-deoxyuridine (EdU) incorporation assay, we confirmed that tumoral $T_{reg}$ cells were markedly more proliferative in comparison to their counterparts residing in tumor-draining lymph nodes (dLNs) (Extended Data Fig. 10a,b). To test whether Foxp3 degradation would compromise the tumoral $T_{reg}$ function, we implanted B16-OVA melanoma cells in the flank of $Foxp3^{AID}R26^{TIR1(F74G)}$ and $Foxp3^{AID}R26^{WT}$ mice. On day 5 after tumor implantation, tumor-bearing mice were treated daily with 5-ph-IAA to induce Foxp3 degradation (Fig. 7a). While the tumors grow unabatedly in $Foxp3^{AID}R26^{WT}$ mice, tumors in $Foxp3^{AID}R26^{TIR1(F74G)}$ ceased to grow and underwent rapid shrinkage (Fig. 7b,c). CD8+ T cells and natural killer (NK) cells within the tumor exhibited heightened effector function, as evidenced by increased interferon (IFN)γ production (Fig. 7d and Extended Data Fig. 10c). Of note, ZsGreen− effector CD4 T cells, instead of upregulating IFNγ, showed increased IL-4 expression (Fig. 7e and Extended Data Fig. 10d), which has been recently implicated in antitumor immunity[33,34]. Notably, severe adverse effects, typically seen upon pan-$T_{reg}$ cell ablation in cancer-bearing mice, were completely lacking, as there were no signs of body weight loss, hunched posture, skin lesions or tissue inflammation based on clinical or histological evaluations (Fig. 7f,g). Although Foxp3 degradation had no effect on the abundance of ZsGreen+ $T_{reg}$ cells in the tumor, dLNs and nondraining lymph nodes (ndLNs), Foxp3 degradation-induced phenotypic changes were markedly more pronounced in tumoral $T_{reg}$ cells. While increased TCF1 expression was observed in both nontumoral and tumoral $T_{reg}$ cells, reduced CTLA4, GITR and CD39 expression was only observed in the latter (Fig. 7h). Thus, Foxp3 degradation boosts antitumor immunity with minimal immune related adverse effects due to a selective loss of intratumoral $T_{reg}$ cell function.

scRNA-seq profiling of tumor-infiltrating $T_{reg}$ cells isolated from $Foxp3^{AID}R26^{WT}$ and $Foxp3^{AID}R26^{TIR1(F74G)}$ mice following 15 days of continuous Foxp3 degradation showed profound and widespread

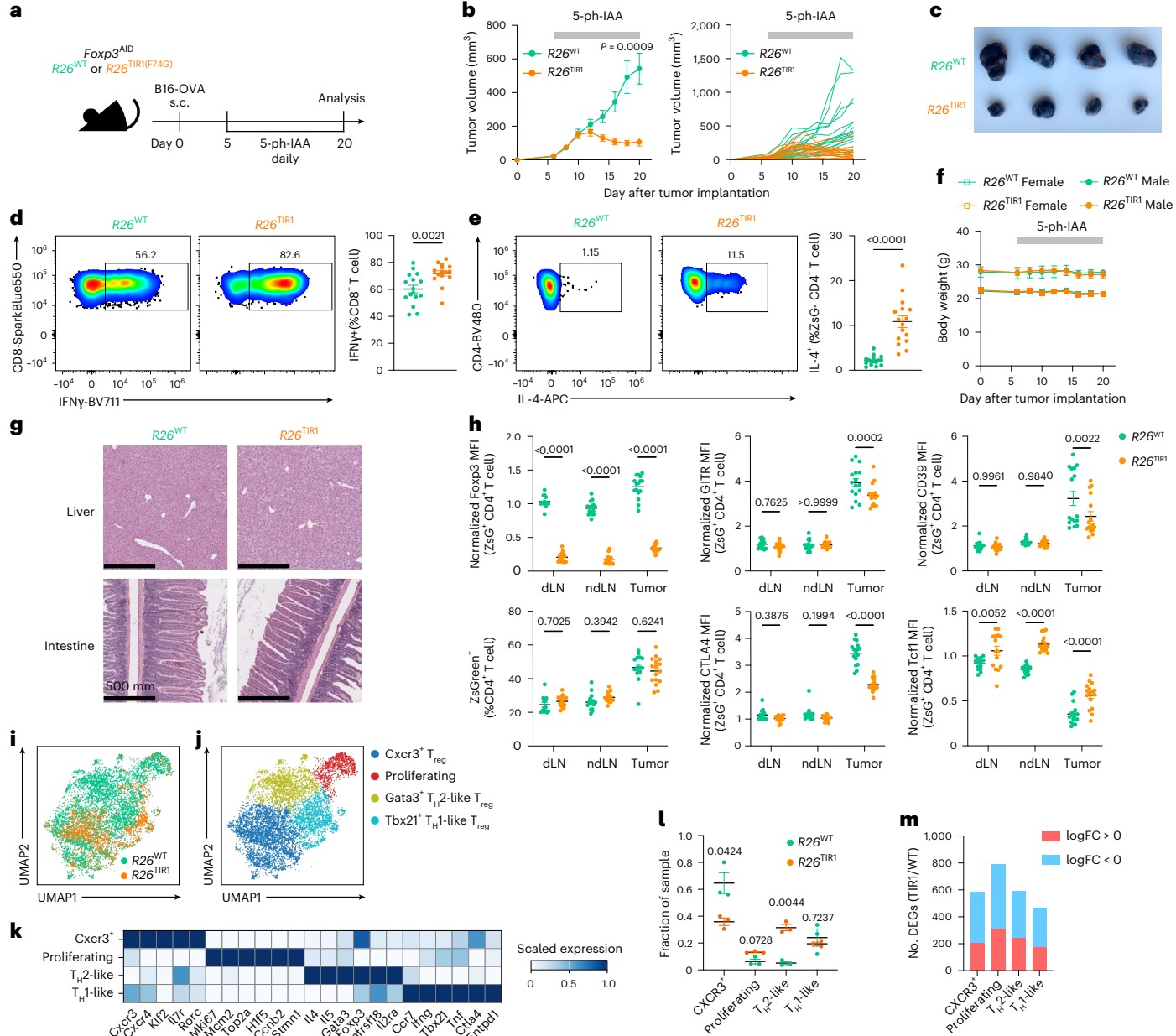

**Fig. 7 | Foxp3 degradation leads to tumor shrinkage with minimal adverse effects. a**, Schematic of the tumor experiment design. s.c., subcutaneous. **b**, Tumor burden over time, shown as average (left) and individual (right) tumor growth curves. Line graph represents mean ± s.e.m. (left). Each line represents a unique mouse (right). Data are pooled from two independent experiments and were analyzed using a two-way ANOVA (mixed-effects model) with Geisser-Greenhouse correction. **c**, Representative tumor images on day 20. **d**, Representative flow cytometry plots (left) and combined data (right) of IFNγ production by tumor-infiltrating CD8⁺ T cells. Each point represents a unique mouse. Scatter-plot shows mean ± s.e.m. Data are pooled from two independent experiments and were analyzed using a two-tailed *t*-test. **e**, Representative flow cytometry plots (left) and quantification (right) of IL-4 production by tumor-infiltrating ZsGreen⁻ CD4⁺ T cells. Each point represents a unique mouse. Scatter-plot shows mean ± s.e.m. Data are pooled from two independent experiments and were analyzed using a two-tailed *t*-test. **f**, Body weight monitoring

throughout the experiment. Line graph shows mean ± s.e.m. Data are pooled from two independent experiments. **g**, H&E staining of liver and intestine on day 20. Images are representative of two independent experiments. **h**, Expression levels of Foxp3, GITR, CD39, ZsGreen, CTLA4 and TCF1 in ZsGreen⁺ CD4 T cells from the dLN, ndLN and tumor on day 20. Each point represents a unique mouse. Scatter-plots show mean ± s.e.m. Data are pooled from two independent experiments and were analyzed using multiple *t*-tests. **i,j**, UMAP visualization of scRNA-seq analysis of tumor T$_{reg}$ cells from *Foxp3*^AID*R26*^WT and *Foxp3*^AID*R26*^TIR1(F74G) mice on day 14 after tumor implantation, colored by genotype (**i**) or cluster (**j**). **k**, Heatmap showing scaled mean UMI-normalized expression values for each cluster in **j**. **l**, Proportional distribution of *Foxp3*^AID*R26*^WT and *Foxp3*^AID*R26*^TIR1(F74G) T$_{reg}$ cells within each cluster in **j**. Each point represents a unique mouse. Scatter-plot represents mean ± s.e.m. Data were analyzed using multiple log-normal *t*-tests. **m**, Number of DEGs between *Foxp3*^AID*R26*^WT and *Foxp3*^AID*R26*^TIR1(F74G) T$_{reg}$ cells in each cluster shown in **j**.

transcriptional alterations within tumor T$_{reg}$ cells visualized by UMAP (Fig. 7i). Characterization of the heterogeneity of Foxp3-degraded and control tumoral T$_{reg}$ cells using coarse clustering identified four clusters: (1) *Gata3*^hi, T$_H$2-like cluster; (2) *Cxcr3*^hi cluster; (3) *Tbx21*^hi,

T$_H$1-like cluster; and (4) *Mki67*^hi, proliferating cluster (Fig. 7j,k). Unlike steady-state T$_{reg}$ cells, where both Foxp3-sufficient and Foxp3-degraded cells remained similarly distributed across clusters even under refined clustering conditions, intratumoral T$_{reg}$ cells displayed a markedly

uneven distribution depending on Foxp3 status, even at this level. Specifically, the $T_H2$-like ($Gata3^{hi}$) cluster was overwhelmingly populated by Foxp3-degraded $T_{reg}$ cells from $Foxp3^{AID}R26^{TIR1(F74G)}$ mice, suggesting that Foxp3 loss preferentially skewed tumor $T_{reg}$ cells toward a $T_H2$-like state. Conversely, the $Cxcr3^{hi}$ cluster was predominantly composed of Foxp3-sufficient $T_{reg}$ cells. The proliferating ($Mki67^{hi}$) cluster contained a slightly higher proportion of Foxp3-degraded $T_{reg}$ cells, whereas the $T_H1$-like ($Tbx21^{hi}$) cluster maintained a comparable representation of $T_{reg}$ cells of both genotypes (Fig. 7l). Given developmental stage-dependent requirements for Foxp3 in maintaining $T_{reg}$ functionality observed in healthy mice, we sought to determine whether Foxp3 exerts cell state context-specific roles across distinct tumoral $T_{reg}$ subsets. Thus, we performed pseudo-bulk differential gene expression analyses between $Foxp3^{AID}R26^{WT}$ and $Foxp3^{AID}R26^{TIR1(F74G)}$ tumor $T_{reg}$ cells within each cluster. Notably, the proliferating cluster revealed the highest number of DEGs (Fig. 7m and Extended Data Fig. 10e). These results confirm a heightened dependence on Foxp3 for the maintenance of tumoral $T_{reg}$ cells.

## Discussion

Foxp3 binding, while sequence specific, is associated with transcriptional changes in only a few Foxp3-bound genes[12]. The latter has been proposed to propagate Foxp3-dependent genome-wide features in a 'relay-like' manner[12]. Such transcriptional 'staging' by Foxp3 would be expected to result in a readily reversible cell state especially when Foxp3 expression is decreased or lost. Indeed, it has been suggested that in inflammatory or hypoxic settings $T_{reg}$ cells can lose Foxp3 expression and become proinflammatory cells, which may contribute to the disease[35–37]. On the other hand, it has been proposed that Foxp3 instability is limited to newly generated $T_{reg}$ cells and that after its transient loss, re-expression of Foxp3 restores $T_{reg}$ suppressor function[8,9].

Our studies demonstrated that, contrary to a complete halt of functional $T_{reg}$ development resulting from genetic $Foxp3$ deficiency, the function and fitness of differentiated $T_{reg}$ cells in adult mice was remarkably durable upon the loss of Foxp3 at steady state. This conclusion is supported by our findings that Foxp3 degradation induced upon 5-ph-IAA treatment of adult $Foxp3^{AID}R26^{TIR1(F74G)}$ mice for up to 4 weeks did not result in clinical manifestations of autoimmune disease or wasting, whereas $Foxp3^{DTR}$ mice of similar age subjected to DT-induced $T_{reg}$ cell ablation succumbed to the disease within 2–3 weeks. The mild increase in the immune tone in the absence of any notable clinical manifestations observed upon continuous Foxp3 degradation for up to 4 weeks was in stark contrast to the rapid progression of fatal inflammatory disease commencing upon $T_{reg}$ cell ablation[17]. Accordingly, instead of impacting the full set of direct and indirect Foxp3-dependent transcriptional features, Foxp3 degradation affected the expression of a much smaller gene set, likely enriched for Foxp3 direct target genes. These results suggest that after Foxp3 establishes the identity, fitness and functionality of $T_{reg}$ cells during their differentiation, $T_{reg}$ transcriptional and functional programs acquire resilience to its loss under physiological conditions.

Notably, our studies revealed that contrary to adult mice, Foxp3-dependent transcriptional and functional $T_{reg}$ programs are absolutely dependent on Foxp3 in neonates, with its loss resulting in severe autoimmune inflammatory disease indistinguishable from that seen in mice with a congenital $Foxp3$ deficiency. It seems unlikely that the observed disease can be accounted for solely by the disruption of Foxp3 expression during thymic $T_{reg}$ cell generation, because the expression of TIR1 is controlled by a Foxp3-driven Cre. In addition, small numbers of functional $T_{reg}$ cells present in the periphery would be able to rescue the disease if they were resilient to Foxp3 loss. In this regard, adoptive transfer of small numbers of $T_{reg}$ cells into neonatal $Foxp3^{null}$ mice or temporally induced installation of Foxp3 protein expression in a small cohort $Foxp3^{LSL}$ expressing $T_{reg}$ 'wannabe' cells affords protection from the disease[1,7]. Furthermore, Foxp3 protein

degradation in both adult Foxp3-expressing thymocytes and early-life peripheral $T_{reg}$ cells caused markedly more extensive transcriptional changes than those observed following Foxp3 degradation in fully differentiated adult peripheral $T_{reg}$ cells. These findings suggest that stabilization of the Foxp3-dependent transcriptional program occurs at a relatively slow tempo. In support of this notion, it took approximately 2–3 weeks for the transcriptional program initiated by newly installed Foxp3 protein in $T_{reg}$ 'wannabe' cells to approach that of bona fide $T_{reg}$ cells. Together, these results support a model whereby the initially vulnerable Foxp3-dependent gene regulatory network in $Foxp3^+$ thymocytes and early-life peripheral $T_{reg}$ cells gradually matures into a stable, largely Foxp3 independent state. This transition seems to unfold over an unexpectedly long timescale, possibly reflecting the 'relay-like' propagation of Foxp3-mediated gene expression changes, whereby a small number of direct Foxp3 targets act in $trans$ to modulate broader networks of gene expression[12]. Additionally, a relatively modest scale of modulation of a number of genes by Foxp3 can contribute to the slow tempo of the acquisition of Foxp3-dependent network resilience. Finally, early-life peripheral T cells, almost exclusively made of recent thymic emigrants, produce limited amounts of IL-2, which can further compromise maintenance of $T_{reg}$ cell transcriptional program upon removal of Foxp3 (ref. 38). In line with this, we found that this program encompasses genes, whose $cis$-regulatory elements are enriched for STAT-binding motifs.

Developmentally established $T_{reg}$ cell transcriptional and functional features can be further tuned by context-dependent interactions between Foxp3 and its protein partners activated in response extracellular cues such as TCR stimulation among others[39–44]. Accordingly, transcriptional changes in $T_{reg}$ cells observed across different tissues upon Foxp3 degradation were more variable in activated in comparison to resting $T_{reg}$ cells, where they were highly correlated. Notably, Foxp3 degradation affected genes in resting $T_{reg}$ cells were enriched for Foxp3 binding, whereas those in activated cells were not. These observations suggest that the indirect components of Foxp3-dependent transcriptional $T_{reg}$ cell program can be influenced by environmental cues. Cell division presents a major challenge for inheritance of differentiated cell states which necessitates a wide range of epigenetic and genetic enforcement mechanisms. Fittingly, robust $T_{reg}$ cell division in vitro and in vivo, especially in a severe inflammatory environment, compromised the resilience of the Foxp3-dependent transcriptional program with a wide range of Foxp3 degradation-induced transcriptional changes beyond the core 'Foxp3-repressed' and 'Foxp3-activated' genes identified under homeostatic conditions.

While the specific mechanisms behind the observed durability of the Foxp3-dependent transcriptional network and its vulnerability remain to be elucidated, their context dependence presents a unique therapeutic opportunity for selective targeting of $T_{reg}$ cell functionality. In this regard, our observation of enhanced antitumor immunity and tumor growth control upon Foxp3 degradation in the absence of severe adverse effects typically observed upon wholesale $T_{reg}$ cell ablation in experimental models of cancer[45–55] offers a novel strategy for immunotherapy of tumors featuring highly activated and proliferative $T_{reg}$ cells.

In conclusion, our studies suggest that the initially vulnerable Foxp3-dependent gene regulatory network and associated functionality of $T_{reg}$ cells progress over time to a resilient state. Once this state is established, $T_{reg}$ cell function, and much of the Foxp3-driven transcriptional program, except for a few genes likely enriched for Foxp3 direct targets, are maintained even after the loss of Foxp3 expression. Yet, the durability of Foxp3-dependent gene regulatory network in mature $T_{reg}$ cells can be compromised in diseased tissue contexts such as the tumor microenvironment; this vulnerability can be leveraged for therapeutic disruption of intratumoral $T_{reg}$ cell function. These findings have noteworthy implications for understanding $T_{reg}$ cell function in health and disease.

## Online content

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

[1]Howard Hughes Medical Institute and Immunology Program, Sloan Kettering Institute, Memorial Sloan Kettering Cancer Center, New York, NY, USA. [2]Department of Immunobiology and Institute of Biomolecular Design and Discovery, Yale University, West Haven, CT, USA. [3]Lewis-Sigler Institute for Integrative Genomics, Princeton University, Princeton, NJ, USA. [4]Immunology and Microbial Pathogenesis Program, Weill Cornell Graduate School of Medical Sciences, New York, NY, USA. [5]Tri-Institutional MD–PhD Program, Weill Cornell Medicine, The Rockefeller University and Memorial Sloan Kettering Cancer Center, New York, NY, USA. [6]Gerstner Sloan Kettering Graduate School of Biomedical Sciences, Memorial Sloan Kettering Cancer Center, New York, NY, USA. [7]Department of Computer Science, Princeton University, Princeton, NJ, USA. [8]Present address: Immunology Discovery, Genentech Inc., South San Francisco, CA, USA. [9]These authors contributed equally: Wei Hu, Gabriel A. Dolsten, Eric Y. Wang, Giorgi Beroshvili. [10]These authors jointly supervised this work: Wei Hu, Yuri Pritykin, Alexander Y. Rudensky. ✉e-mail: wei.hu.wh447@yale.edu; pritykin@princeton.edu; rudenska@mskcc.org

## Methods

### Mice

All animal experiments in this study were approved by the Sloan Kettering Institute (SKI) Institutional Animal Care and Use Committee under protocol no. 08-10-023 or Yale University Institutional Animal Care and Use Committee under protocol no. 2023-20503. Mice were housed at the SKI or Yale University animal facility under specific-pathogen-free conditions on a 12-h light–dark cycle with free access to water and regular chow diet. The average ambient temperature is 21.5 °C and the average humidity is 48%. $Foxp3^{DTR}$, $Foxp3^{fl}$ and $Cd4^{creERT2}$ mice used in this study have been previously described[1,17,56]. $Foxp3^{AID}$, $ROSA26^{TIR1}$ and $ROSA26^{TIR1(F74G)}$ mice were generated in this study. All control and experimental animals were age-matched and littermates were used as controls unless otherwise indicated.

### Generation of $Foxp3^{AID}$, $ROSA26^{TIR1}$ and $ROSA26^{TIR1(F74G)}$ mice

Gene targeting was carried out in 129/B6 F1 hybrid embryonic stem (ES) cells using a targeting vector spanning the Foxp3 locus. At the N terminus, the AID sequence was fused to Foxp3 via a seven-amino-acid flexible linker (GSHGGSG). An IRES-ZsGreen-T2A-iCre-Frt-neo-Frt cassette was inserted into the 3′ untranslated region (UTR) immediately downstream of the stop codon. ES cell clones with successful targeting were validated and injected into CD-1 tetraploid blastocysts to generate knock-in founders. These founders were then crossed with a Flp-deleter line to excise the neo-cassette. The resulting F1 progeny were backcrossed to the C57BL/6 background for at least three generations before in vivo experiments.

To generate the $ROSA26^{TIR1}$ strain, a targeting construct was assembled by cloning a TIR1 (wild-type; WT)-3xMyc-P2A-mCherry fragment into the FseI-linearized Ai32-targeting vector (Addgene, #34880), positioned between a loxP-Stop-loxP cassette and a WPRE (woodchuck hepatitis virus post-transcriptional regulatory element). The complete targeting vector included the $ROSA26$ left homology arm, the CAG promoter, a loxP-Stop-loxP cassette, WPRE, a bovine growth hormone (bGH) polyadenylation signal, an AttB-neo-AttP drug selection cassette and the $ROSA26$ right homology arm. The linearized construct was transfected into albino C57BL/6 ES cells. Neomycin-resistant clones were screened by Southern blot, followed by PCR to confirm correct targeting. Karyotypically normal ES cell clones were used to generate chimeras, which were subsequently bred with albino C57BL/6 mice. Germline-transmitted founders were crossed with a PhiC31-deleter strain to remove the neomycin resistance cassette.

To generate the $ROSA26^{TIR1(F74G)}$ strain, CRISPR-mediated gene editing was used to introduce a point mutation in ROSA26$^{TIR1/+}$ zygotes, converting phenylalanine (F; **TTC**) to glycine (G; **GGC**) at position 74.

gRNA spacer sequence (synthesized by IDT):
GAAGCGGCTGAAGTTGGAGC

Homology-directed repair template sequence (synthesized by IDT as single-stranded DNA):

GCGGCGTCTTCGTGGGCAACTGCTACGCCGTGCGCGCCGGC-
CGCGTCGCCGCGCGGTTCCCCAACGTGCGGGCGCTCACGGTGAAGGG-
GAAGCCACAC**GGC**GCCGACTTCAACCTCGTGCCCCCCGACTGGGGCG-
GCTACGCGGGGCCGTGGATCGAGGCGGCCGCGAGGGGATGCCACG-
GCCTGGAGGAGCTCAGGATG

In addition to the desired point mutation, a silent mutation (CCC to CCA) was included in the homology-directed repair template to prevent re-cutting by Cas9. Offspring were screened by PCR amplification followed by KasI restriction digest. Confirmed mutants were validated by Sanger sequencing and bred to C57BL/6 mice to establish germline-transmitting founders.

### Treatment of mice with 5-ph-IAA and diphtheria toxin

DT (List Biological Laboratories, 150) was dissolved in phosphate-buffered saline (PBS) and administered intraperitoneally (i.p.) at a dose of 20 µg kg$^{-1}$ on day 0. This was followed by six subsequent injections of 5 µg kg$^{-1}$ every other day. Mice were killed for analysis on day 14. Then, 5-ph-IAA (MedChemExpress, HY-134653) was dissolved with 0.2 M NaOH, diluted in PBS and administered daily via i.p. injection at a dose of 10 mg kg$^{-1}$ per mouse.

### Reagents and antibodies

The following antibodies and reagents were used in this study for flow cytometry, with clones, venders, catalog numbers and dilutions as indicated: anti-Siglec-F (E50-2440, BD, 562681, 1:400), anti-I-A/I-E (M5/114.15.2, Biosciences, 566086, 1:1,200), anti-NK1.1 (PK136, Thermo Fisher, 47-5941-82, 1:400), anti-CD45 (30-F11, BioLegend, 103136, 1:600), anti-CD11b (M1/70, BioLegend, 101257, 1:800), anti-CD11b (M1/70, BD Biosciences, 363-0112-82, 1:400), anti-CD3ε (17A2, BioLegend, 100237, 1:500), anti-γδTCR (GL3, BD Biosciences, 750410, 1:300), anti-Cd278 (C398.4A, BD Biosciences, 567918, 200), anti-TCR (H57-597, BD Biosciences, 748405, 1:300), anti-TCR β (H57-597, Thermo Fisher, 47-5961-82, 1:300), anti-TCR (H57-597, BioLegend, 109227, 1:200), anti-TCR β (H57-597, Thermo Fisher, 12-5961-83, 1:400), anti-CD153 (RM153, BD Bioscience, 741575, 1:400), anti-CD24 (M1/69, Thermo Fisher, 46-0242-82, 1:800), anti-CD304 (3E12, BioLegend, 145209, 1:300), anti-CD3 (IM7, BioLegend, 103049, 1:400), anti-CD44 (IM7, BD Biosciences, 563971, 1:400), anti-CD44 (IM7, BioLegend, 103026, 1:100), anti-ZsGreen (polyclonal, Frontier Institute Co., LTD, MSFR106470, 1:800), anti-KLRG1 (2F1, Thermo Fisher, 35-5893-82), anti-CD39 (24DMS1, Thermo Fisher, 25-0391-82, 1:400), anti-TCF1 (C63D9, Cell Signaling, 6709, 1:200), anti-IL-10 (JES5-16E3, BioLegend, 505021, 1:200), anti-CD4 (RM4-5, BD Biosciences, 414-0042-82, 1:400), anti-CD4 (RM4-5, BioLegend, 100536, 1:400), anti-CD4 (RM4-5, Thermo Fisher, 47-0042-82, 1:400), anti-CD4 (RM4-5, BioLegend, 100548, 1:400), anti-TNF (MP6-XT22, BioLegend, 506329, 1:400), anti-IFNg (XMG1.2, BioLegend, 505836, 1:200), anti-IL-22 (1H8PWSR, Thermo Fisher, 46-7221-80, 1:400), anti-IL-13 (eBio13A, Thermo Fisher, 12-7133-82, 1:400), anti-IL-4 (11B11, Thermo Fisher, 17-7041-82, 1:300), anti-CD11c (N418, Thermo Fisher, 48-0114-82, 1:200), anti-Ly6c (HK1.4, BioLegend, 128037, 1:1,200), anti-Ly6C (HK1.4, BioLegend, 128041, 1:1,000), anti-CD122 (TM-β1, BD Bioscience, 564763, 1:200), anti-GARP (YGIC86, Thermo Fisher, 25-9891, 1:200), anti-CD86 (GL1, Thermo Fisher, 12-0862-85, 1:400), anti-Ly6G (1A8, BioLegend, 127618, 1:500), anti-CD64 (X54-5/7.1, BioLegend, 139306, 1:200), anti-CD127 (A7R34, Tonbo Bioscience, 20-1271-U100, 1:200), anti-CD122 (5H4, Thermo Fisher, 13-1221-82, 1:200), anti-Guinea Pig (polyclonal, Thermo Fisher, SA5-10094, 1:1,000), anti-FR4 (12A5, BD Biosciences, 744121, 1:200), anti-FR4 (12A5, BD Biosciences, 560318, 1:200), anti-OX40 (OX-86, Thermo Fisher, 46-1341-82, 1:300), anti-CD120b (TR75-89, BD Bioscience, 564088, 1:200), anti-CD103 (M290, BD Biosciences, 566118, 1:300), anti-Ly-6C (HK1.4, BioLegend, 128037, 1:1,000), anti-CD90.2 (30-H12, BioLegend, 105320, 1:800), anti-CD90.2 (53-2.1, BD Biosciences, 564365, 1:1,500), anti-Foxp3 (FJK-16s, Thermo Fisher, 48-5773-82, 1:200), anti-Foxp3 (FJK-16s, Thermo Fisher, 17-5773-82, 1:200), anti-CD19 (6D5, BioLegend, 115510, 1:600), anti-F4|80 (BM8, BioLegend, 123133, 1:200), anti-CD4 (RM4-5, EBioscience, 564667, 1:400), anti-CD4 (RM4-5, BioLegend, 100553, 1:400), anti-CD8α (53-6.7, BioLegend, 100780, 1:600), anti-CD8α (53-6.7, BioLegend, 564297, 1:400), anti-CD8α (53-6.7, BioLegend, 100752, 1:500), anti-GITR (DTA-1, Thermo Fisher, 48-5874-82, 1:500), anti-CD73 (eBioTY/11.8, Thermo Fisher, 46-0731-82, 1:400), anti-CD73 (TY/11.8, BioLegend, 127208, 1:400), anti-CD62L (MEL-14, BioLegend, 104441, 1:100), anti-CD62L (MEL-14, BD Biosciences, 565213, 1:600), anti-CD62L (MEL-14, BD Biosciences, 741230, 1:800), anti-CD62L (MEL-14, BioLegend, 104441, 1:400), anti-CD62L (MEL-14, BioLegend, 104438, 1:1,600), anti-CTLA4 (UC10-4B9, BioLegend, 106323, 1:200), anti-CTLA4 (UC10-4B9, Thermo Fisher, 12-1522-82, 1:400), anti-Helios (22F6, BioLegend, 137216, 1:400), anti-Helios (22F6, BioLegend, 137236, 1:400), anti-Eos (ESB7C2, Thermo Fisher, 12-5758-82, 1:400), anti-Ki-67 (SolA15, Thermo Fisher, 61-5698, 1:2,000), anti-Ki67 (B56, BD Biosciences, 563757, 1:1,000), anti-Ki67

(SolA15, Fisher Scientific, 15-5698-82, 1:8,000), anti-CD25 (PC61, BD Biosciences, 564022, 1:300; Thermo Fisher, 17-0251-82, 1:400), anti-PD-1 (29 F.1A12, BioLegend, 135225, 1:400), anti-CD45 (30-F11, BioLegend, 103157, 1:1,000), anti-IL-2 (JES6-5H4, BioLegend, 503818, 1:400), streptavidin (Thermo Fisher, 46-4317-82, 1:1,000), Picolyl-Azide (Jena Bioscience, CLK-1288-5), CTV (Thermo Fisher, C34557), Zombie NIR dye (BioLegend, 423105, 1:1,000), Sytox Blue (Thermo Fisher, S34857) and anti-mouse CD16/32 (2.4G2, Tonbo, 70-0161-M001, 1:500).

The following capturing antibodies were used for ELISA: anti-mouse IL-13 (14-7133-68, Invitrogen, 88-7137-88), anti-mouse IL-4 (14-7041-68 A, Invitrogen, 88-7044-88), anti-mouse IL-2 (eBioscience, 14-7022-68), anti-mouse IgE (R35-72, BD Pharmingen, 553413), goat anti-mouse IgG1 (2794408, Southern Biotech, 1070-01), goat anti-mouse IgG3 (2794567, Southern Biotech, 1100-01), goat anti-mouse IgG2a (2794475, Southern Biotech, 1080-01), goat anti-mouse IgG2b (2794517, Southern Biotech, 1090-01), goat anti-mouse IgG2c (2794464, Southern Biotech, 1079-01), goat anti-mouse IgA (2314669, Southern Biotech, 1040-01) and goat anti-mouse IgM (2794197, Southern Biotech, 1020-01). The following detection antibodies were used for ELISA: biotin anti-mouse IL-13 (13-7135-68A, Invitrogen, 88-7137-88), anti-mouse IL-4 (13-7042-68C, Invitrogen, 88-7044-88), anti-mouse IL-2 (eBioscience, 33-7021-68), goat anti-mouse Ig (2728714, Southern Biotech, 1010-05) and biotin rat anti-mouse IgE (R35-118, BD Pharmingen, 553419).

The following reagents were used to generate ELISA standard curves: mouse IL-4 lyophilized standard (39-8041-60, Invitrogen, 88-7044-88), mouse IL-13 lyophilized standard (39-7137/2EB-60, Invitrogen, 88-7137-88), mouse IL-2 (Thermo Fisher 212-12-5UG), mouse IgG1, κ, isotype control (15H6, Southern Biotech, 0102-01), mouse IgG2a, κ, isotype control (UPC-10, Sigma, M5409), IgG2b isotype control (MOPC-141, Sigma, M5534), Mouse IgG2c (6.3, AB_2794064, Southern Biotech, 0122-01), purified mouse IgG3, κ, isotype control (A112-3, BD Pharmingen, 553486), Purified Mouse IgA, κ, isotype control (M18-254, BD Pharmingen, 553476), IgM isotype control from murine myeloma (MOPC 104E, Sigma, M5909), purified mouse IgE, κ, isotype control (C38-2, BD Pharmingen, 557079).

## Enzyme-linked immunosorbent assay

ELISA experiments for IL-2, IL-4 and IL-13 were performed in the following way. Cells were isolated from pooled SLOs (peripheral lymph nodes (cervical, axillary, brachial and inguinal) and spleen) and cultured on a 96-well U-bottom plate with 5% CO$_2$ in 200 µl complete cell culture medium (RPMI 1640 medium supplemented with 10% fetal bovine serum (FBS), 100 U ml$^{-1}$ penicillin–streptomycin, 2 mM L-glutamine, 10 mM HEPES and 50 µM β-mercaptoethanol) and recombinant human IL-2 (0.5 U µl$^{-1}$, Roche, C168121-01). Cells were treated with 5-ph-IAA (5 mM daily) in the presence of anti-CD3/CD28 activation beads (Thermo Fisher, 11452D). The culture was terminated after 3 days and the supernatant was used for the detection of the aforementioned cytokines. In brief, a 96-well flat-bottom plate was coated with capture antibody in Coating Buffer (00-0000-53, Invitrogen, 88-7044-88) and incubated overnight at 4 °C. The next day, the plate was washed and blocked with ELISA/ELISPOT diluent (00-4202-55, Invitrogen, 88-7044-88) at room temperature for 1 h. Then, the plate was washed and serial dilutions of standards were performed using the ELISA/ELISPOT diluent. Next, samples were added to the plate and incubated overnight at 4 °C. The next day, the plate was washed and the detection antibody was added and incubated for 1 h at room temperature. Next, the plate was washed and incubated with streptavidin–HRP for IL-2 and IL-4 (00-50050-68, Invitrogen, 88-7044-88) or avidin–HRP for IL-13 (00-4100-94, Invitrogen, 88-7137-88) for 30 min at room temperature. The plate was then washed and incubated with TMB solution (00-4201-56, Invitrogen, 88-7044-88) at room temperature for 5–15 min. Finally, 1 M H$_3$PO$_4$ (Sigma-Aldrich, P5811) was added to the plate to stop the colorimetric reaction.

Antibody ELISAs were conducted as previously described. In brief, mouse peripheral blood was collected via cardiac puncture immediately after killing into BD SST microcontainer tubes (02-675-185) and sera were collected after centrifugation. Flat-bottom 96-well plates were coated with capturing antibodies in 50 µl 0.1 M NaHCO$_3$ solution at pH 9.5 overnight at 4 °C. The plates were then emptied, blocked with 200 µl 1% bovine serum albumin (VWR, 97061-422) in PBS and washed three times with PBS containing 0.05% Tween-20 (Sigma-Aldrich, P1379). Then, 50 µl serum at appropriate dilutions was added and incubated overnight at 4 °C. The plate was then incubated with 50 µl biotinylated detection antibodies at 37 °C for 2–3 h, followed by 50 µl avidin–HRP (Thermo Fisher, 18-4100-51) at 37 °C for 30 min and 100 µl TMB solution (Thermo Fisher, 00-4201-56) at room temperature, with 3–4 washes with PBS–Tween in between each incubation step. The colorimetric reaction was stopped with 100 µl 1 M H$_3$PO$_4$ after 5–10 min.

Absorbance at 450 nm was measured with a Synergy HTX plate reader (BioTek). Concentrations of antigens were determined using standard curves constructed with purified recombinant proteins and calculated with Gen5 3.02.2 (BioTek).

## Isolation of cells from lymphoid organs, lungs and tumors

For flow cytometry analyses, animals were killed and perfused with 20 ml PBS. Cells were isolated from lymphoid organs by meshing with syringe plunger through a 100-mm cell strainer (Corning, 07-201-432). Lungs and tumors were digested in RPMI 1640 with 2% FBS, 10 mM HEPES buffer, 100 U ml$^{-1}$ penicillin–streptomycin, 2 mM L-glutamate, 0.2 U ml$^{-1}$ collagenase A (Sigma, 11088793001) and 1 U ml$^{-1}$ DNase I (Sigma-Aldrich, 10104159001) for 45 min at 37 °C with vigorous shaking at 250 rpm. Then, 6.35-mm ceramic beads (MP Biomedicals, 116540034) were included to help with tissue dissociation. The digested lungs were filtered through 70-mm separation filters (Miltenyi Biotec, 130-095-823), washed and centrifuged in PBS-adjusted 40% Percoll (Sigma-Aldrich, 17-0891-01) to enrich for lymphocytes. Erythrocytes from spleen, lung and liver were lysed using ACK lysis buffer (150 mM NH$_4$Cl (Sigma-Aldrich, A9434), 10 mM KHCO$_3$ (Sigma-Aldrich, P7682) and 0.1 mM Na$_2$EDTA at pH 7.4).

For flow cytometry analysis, cells were stained with Zombie NIR dye in PBS for 10 min at 4 °C to identify the dead cells followed by staining with anti-mouse CD16/32 in staining buffer (PBS with 0.2% bovine serum albumin (BSA), 10 mM HEPES buffer and 2 mM EDTA) for 10 min at 4 °C to block the Fc receptors. Next, cells were stained with fluorescently conjugated antibodies detecting cell surface antigens for 30 min at 4 °C. To access the intracellular antigens, cells were fixed and permeabilized with eBioscience transcription factor staining buffer set (00-5523-00) according to the manufacturer's instructions. Samples were recorded on Aurora cytometer (Cytek) by using of SpectroFlo software v.3.1.2 and analyzed in FlowJo v.10.10.0.

For cell sorting, cells isolated from pooled peripheral lymph nodes (cervical, axillary, brachial and inguinal) and spleen were enriched for CD4$^+$ T cells using a mouse CD4$^+$ T Cell Isolation kit (Miltenyi, 130-104-454) according to the manufacturer's instructions. Next, samples were stained with antibodies, washed and resuspended in a Sytox Blue-containing (1:8,000) staining buffer to exclude dead cells. T$_{reg}$ cells (CD4$^+$TCRβ$^+$ZsGreen$^+$ from Foxp3$^{AID}$ROSA26$^{WT}$ mice and CD4$^+$TCRβ$^+$ZsGreen$^+$mCherry$^+$ from Foxp3$^{AID}$ROSA26$^{TIR1(F74G)}$ mice) and naive CD4$^+$ T cells (CD4$^+$TCRβ$^+$ZsGreen$^-$CD44$^{lo}$CD62L$^{hi}$) were sorted into cell culture medium.

## Flow cytometric analysis of cytokine production

To measure cytokine production following ex vivo stimulation, a single-cell suspension was incubated with 5% CO$_2$ at 37 °C for 4 h in cell culture medium (200 ml per well) supplied with 50 ng ml$^{-1}$ phorbol-12-myristate-13-acetate (Sigma-Aldrich, P8139), 500 ng ml$^{-1}$ ionomycin (Sigma-Aldrich, I0634), 2 µM monensin (Sigma-Aldrich, M5273) and 1 µg ml$^{-1}$ brefeldin A (Sigma-Aldrich, B6542). Cells were stained for flow cytometry as described above except for the fixation/permeabilization step and cytokine staining in which case BD Cytofix/

Cytoperm Kit (BD Biosciences, 554715) was used according to the manufacturer's instructions.

## Ex vivo CTV labeling

Sorted $T_{reg}$ cells were labeled with CTV (5 mM) and cultured with 5% $CO_2$ on a 48-well flat-bottom plate pre-coated with anti-CD3/CD28 antibody with concentration of 5 mg ml$^{-1}$ each of anti-CD3 (145-2C11, BioXcell, BE0001-1) and anti-CD28 (37.51, BioXcell, BE0015-1). The culture was maintained in cell culture medium (600 ml per well) supplied with a recombinant human IL-2 (0.5 U μl$^{-1}$). Cells were treated with 5-ph-IAA daily (5 mM). The first part of the culture was terminated after 16 h to capture undivided cells and the cells that divided once. Another part of the culture was terminated after 72 h to capture the cells dividing twice and more. Cells were stained for analysis as described above.

## In vitro suppression assay

Cells were isolated from pooled SLOs and enriched for CD4$^+$ T cells as described above. Resting $T_{reg}$ cells (CD4$^+$TCRβ$^+$ZsGreen$^+$CD62L$^{hi}$CD44$^{lo}$) were FACS sorted and 40,000 cells per well were plated on a 96-well U-bottom plate. Then, 40,000 sorted naive CD4$^+$ T cells from CD45.1 *Foxp3*$^{DTR}$ mice (CD4$^+$TCRβ$^+$GFP$^-$CD62L$^{hi}$CD44$^{lo}$) and 100,000 red blood cell-lysed splenocytes from *Tcrb*$^{-/-}$*Tcrd*$^{-/-}$ mice were added to the culture. Next, anti-CD3 (145-2C11, BioXcell, BE0001-1) antibody was added to each well to a final concentration of 1 mg ml$^{-1}$. Cells were treated with 5-ph-IAA daily (5 mM). Cells were incubated in cell culture medium (200 ml per well) for 72 h with 5% $CO_2$ and then prepared for flow cytometry as described above. CTV-labeled Naive CD4$^+$ T cells that were divided more than four times were used to calculate $T_{reg}$ cell-mediated suppression with the following formula: %Suppression (sample X) = (%divided (no $T_{reg}$) − %divided (sample X))/(%divided (no $T_{reg}$)), where sample X refers to any given sample.

## Histology

Tissues were fixed in 10% neutral buffered formalin, transferred into 70% alcohol and sent to the HistoWiz for the downstream services. In brief, tissues were embedded in paraffin and sectioned into 5-mm slices followed by H&E staining. Lymphocytic infiltration was blindly scored with the criteria, 0, normal; 1, mild increase; 2, moderate increase; and 3, severe increase.

## 5-ethynyl-2′-deoxyuridine labeling

Mice were injected i.p. with EdU (5 mg per mouse, MedChemExpress, HY-118411) for three consecutive days. Cells were stained with surface and intracellular antibodies as described above. To stain for EdU, a click reaction was performed after staining with intracellular antibodies. In brief, cells were incubated in the Click Reaction Buffer (CuSO$_4$ 16.67 mM (Fisher Science Education, 7758-98-7), BTTAA(2-(4-((bis((1-(tert-butyl)-1H-1,2,3-triazol-4-yl)methyl)amino)methyl)-1H-1,2,3-triazol-1-yl)acetic acid) 41.67 mM (Jena Bioscience, CLK-067-100), picolyl-azide Alexa Fluor 555 1.25 μM (Jena Bioscience, CLK-1288-5), sodium ascorbate 100 mM (Spectrum, 134-03-2), H$_2$O and PBS, pH 7.4) for 1 h at room temperature followed by extensive washing before recording data on the cytometer.

## B16 melanoma model

B16-OVA melanoma cells[57] were cultured in RPMI medium supplemented with 10% FBS (Gibco), 100 U ml$^{-1}$ penicillin–streptomycin at 37 °C in a humidified incubator with 5% $CO_2$. Cells were maintained in logarithmic growth phase and collected at ~70–80% confluency using 0.05% trypsin-EDTA. Cells were washed twice with sterile PBS. For tumor implantation, 1.5 × 10$^6$ cells in 100 μl sterile PBS were injected subcutaneously into the right flank of mice using a 27G needle. Starting from day 5 after tumor implantation, mice were injected daily with 5-ph-IAA. Mice were monitored daily for general health and tumor growth. Tumor dimensions were measured using calipers and the volume was calculated using the formula (length × width$^2$)/2.

## Bulk RNA-seq and data analysis

For bulk RNA-seq from adult female *Foxp3*$^{AID/+}$*ROSA26*$^{WT}$ and *Foxp3*$^{AID/+}$*ROSA26*$^{TIR1(F74G)}$ mice, single cell suspensions from pooled SLOs or thymus were enriched for CD4$^+$ T cells as described above. CD4$^+$TCRβ$^+$ZsGreen$^+$CD62L$^{hi}$CD44$^{lo}$ resting $T_{reg}$ cells and CD4$^+$TCRβ$^+$ZsGreen$^+$CD62L$^{lo}$CD44$^{hi}$ activated $T_{reg}$ cells from the SLOs, as well as CD4$^+$TCRβ$^+$ZsGreen$^+$CD73$^-$ nascent $T_{reg}$ cells from the thymus, were double sorted into TRIzol Reagent (Thermo Fisher, 15596-018). RNA extraction was then performed according to the manufacturer's instructions. In brief, phase separation in cells lysed in 1 ml TRIzol Reagent was induced with 200 μl chloroform. RNA was extracted from 350 μl of the aqueous phase using the miRNeasy Micro kit (QIAGEN cat. no. 217084) on the QIAcube Connect (QIAGEN) according to the manufacturer's protocol. Samples were eluted in 15–18 μl RNase-free water.

After RiboGreen quantification and quality control by Agilent BioAnalyzer, 1–2 ng total RNA with RNA integrity numbers ranging from 5.5 to 9.3 underwent amplification using the SMART-Seq v.4 Ultra Low Input RNA kit (Clonetech cat. no. 63488), with 12 cycles of amplification. Subsequently, 4.5–8 ng of amplified cDNA was used to prepare libraries with the KAPA Hyper Prep Kit (Roche 07962363001) using eight cycles of PCR. Samples were barcoded and run on a NovaSeq 6000 in a PE100 run, using the NovaSeq 6000 S4 Reagent kit (200 cycles) (Illumina). An average of 58 million paired reads were generated per sample and the percent of mRNA bases per sample ranged from 79% to 87% and ribosomal reads averaged 0.35%.

Bulk RNA-seq was aligned to the mm39 reference using the STAR aligner v.2.7.10b run with default settings. Reads were counted using Rsubread FeatureCounts v.2.8.2 to create gene count matrices with the following settings: –primary -C -Q 30 –countReadPairs -B -p.

For bulk RNA sequencing of in vitro activated and Foxp3-degraded cells (Fig. 6g), cells from *Foxp3*$^{AID/y}$*ROSA26*$^{WT}$ and *Foxp3*$^{AID/y}$*ROSA26*$^{TIR1(F74G)}$ were isolated from pooled SLOs and enriched for CD4$^+$ T cells as described above. $T_{reg}$ cells (CD4$^+$TCRβ$^+$ZsGreen$^+$, mCherry$^-$ for *ROSA26*$^{WT}$ and mCherry$^+$ for *ROSA26*$^{TIR1(F74G)}$) were FACS sorted and cultured at a density of 1 × 10$^6$ cells per ml in complete RPMI (RPMI 1640 medium supplemented with 10% FBS, 100 U m$^{-1}$ penicillin–streptomycin, 2 mM L-glutamine, 10 mM HEPES and 50 μM β-mercaptoethanol), recombinant human IL-2 (2,000 U ml$^{-1}$) and Dynabeads Mouse T-Activator CD3/28 (2:1 bead-to-cell ratio, Gibco). $T_{reg}$ cells were cultured at 37 °C in a humidified incubator with 5% $CO_2$ for 3 days. $T_{reg}$ cells were washed with complete RPMI, resuspended at a density of 1 × 10$^6$ cells per ml in complete RPMI, and rested in the incubator for 2 h, after which 5-ph-IAA (5 mM) was added to the medium. After 7 h, cells were lysed in RNA lysis buffer and RNA was extracted using the Quick-RNA Microprep kit (Zymo Research). Five technical replicates were performed for each genotype. RNA was quantified by NanoDrop. RNA-seq libraries were generated using the BRB-seq kit (Alithea Genomics) as per the manufacturer's instructions. Then, 200 ng RNA per sample was used as input. Final sequencing libraries were run on an AVITI instrument (Element Biosciences) with the following read structure: index 1, eight cycles; index 2, eight cycles; read 1, 28 cycles; and read 2, 90 cycles. An average of 5.7 million reads and 3.9 million unique molecular identifiers (UMIs) were acquired per sample. FASTQ files were aligned to the GRCm39 reference genome and a counts matrix was generated using STARsolo[58] following commands provided in the Alithea Genomics BRB-seq user guide (v.0.8.1. rev A August 2024).

## Single-cell RNA analysis of SLO $T_{reg}$ cells

*Foxp3*$^{AID/+}$*ROSA26*$^{WT}$ and *Foxp3*$^{AID/+}$*ROSA26*$^{TIR1(F74G)}$ mice were treated with 5-ph-IAA for 0, 3 and 7 days. Enriched CD4$^+$ T cells from the spleen and lymph nodes were barcoded using hashtag oligonucleotides (BioLegend). FACS-sorted *Foxp3*$^{AID}$*R26*$^{WT}$ and *Foxp3*$^{AID}$*R26*$^{TIR1(F74G)}$ $T_{reg}$ cells (four biological replicates of each genotype) within each time point were multiplexed together, washed once with PBS containing 0.04% BSA and resuspended in PBS containing 0.04% BSA to a final

concentration of 700–1,200 cells per µl. For each time point, RNA sequencing was performed in one lane on a Chromium instrument (10x Genomics) following the user guide manual for 3′ v.3.1. The viability of cells was above 80%, as confirmed with 0.2% (w/v) Trypan blue staining (Countess II). Approximately 30,000 cells were targeted. Following reverse transcription and cell barcoding in droplets, emulsions were broken and cDNA was purified using Dynabeads MyOne SILANE followed by PCR amplification as per the manual instructions. Final libraries were sequenced on Illumina NovaSeq 6000-S4 platform (R1, 28 cycles; i7, eight cycles; and R2, 90 cycles). Single-cell RNA-seq count matrices were generated using Cell Ranger v.6.1.2. Replicates from the same day of treatment (Tir1 or WT) were demultiplexed using hashtag oligonucleotides. This resulted in a filtered UMI matrix containing 45,314 cells and 32,285 genes, which was analyzed using Scanpy v.1.9.3. Cells were filtered based on mitochondrial counts and gene expression levels. Lowly expressed genes, nonprotein-coding genes, pseudogenes and genes corresponding to TCR/BCR were removed from the data. Gene expression was normalized using Pearson residual normalization with theta = 1. Principal-component analysis (PCA) was run with 100 principal components (PCs) and a $k$-nearest neighbors ($k$-NN) graph was established with $k = 30$. Leiden clustering was run with a resolution of 1. UMAP was run with default parameters. Clusters were scored using gene sets from ImmGen. Two clusters (342 cells) scoring high in pDC and NK gene signatures were filtered out. Together, this resulted in a filtered gene expression matrix of 39,256 cells and 9,155 genes. To remove the effect from genes that were differential due only to genotype (WT/Tir) and not the degradation of Foxp3, we identified differential genes between D0 Tir1 and D0 WT using DESeq2 and filtered these genes out from the data matrix, resulting in a matrix of 39,256 cells and 9,119 genes. Cells were then re-clustered according to the protocol (PCA, $k$-NN, Leiden and UMAP) described above.

### Single-cell RNA-seq analysis of intratumoral T_reg cells

Single-cell preparation was performed using GEM-X Flex (10x Genomics). $Foxp3^{AID/y}ROSA26^{WT}$ and $Foxp3^{AID/y}ROSA26^{TIR1(F74G)}$ mice (three per group) were injected subcutaneously with $1.5 \times 10^6$ B16-OVA cells in 100 µl sterile PBS. Starting on day 5 post-implantation, 200 µg 5-ph-IAA in 200 µl was injected i.p. daily. On day 14 post-implantation, tumors were processed and prepared for sequencing using the GEM-X Flex assay (10x Genomics). Following dissociation into single-cell suspensions, cells were stained with fluorescent and TotalSeq-C anti-mouse hashtag antibodies (BioLegend) and washed extensively. Cells from each mouse were stained with a unique hashtag antibody. Cells were pooled and fixed in Fixation Buffer B (10x Genomics) at 4 °C for 18 h. After quenching and washing cells as per the manufacturer's instructions (10x Genomics, demonstrated protocol CG000782 Rev A), cells were resuspended in 500 µl Quenching Buffer B. Reporter-positive live/dead stain negative T_reg cells (CD45$^+$CD3$^+$TCRβ$^+$CD8α$^-$CD4$^+$ZsGreen$^+$mCherry$^+$ for $R26^{TIR1(F74G)}$ mice or mCherry$^-$ for $R26^{WT}$ mice) were isolated from fixed single-cell suspensions using FACS and sorted into PBS with 1% nuclease-free BSA Fraction V (Millipore Sigma, 126609) and 0.2 U µl$^{-1}$ RNase-inhibitor (Millipore Sigma, PN-3335399001). Following FACS isolation, cells were spun down, resuspended in 200 µl of Quenching Buffer B and pooled to normalize cell numbers between different mice. The remaining steps were performed as per the manufacturer's instructions (10x Genomics, user guide CG000788 Rev A). In brief, cells were hybridized for 18 h, washed and encapsulated in one lane targeting 60,000 cells. Indexing PCR steps were performed using nine amplification cycles for both the gene expression and cell surface protein libraries. Final sequencing libraries were sequenced on an Illumina NovaSeq X Plus System (index 1, ten cycles; index 2, ten cycles; read 1, 28 cycles; and read 2, 88 cycles).

FASTQ files were processed using Cell Ranger v.9.0 and reads were aligned to the Chromium Mouse Transcriptome Probe Set v.1.1.0 GRCm39-2024-A (10x Genomics). Cells containing fewer than 1,000

UMI counts or 1,000 unique genes were filtered out. Cells containing more than 5% mitochondria-derived transcripts were filtered out. Genes that were expressed in >5 cells were retained for further analysis. Hashtag antibody data were demultiplexed using HashSolo with the following prior probabilities (negative, 0.01; singlet, 0.81; and doublet, 0.18). Cells called doublet or negative were filtered out. The resulting count matrix consisted of 35,367 cells × 19,088 genes and was normalized to median UMI counts. The normalized data were then ln(1 + counts)-transformed for downstream analysis.

### Single-cell RNA-seq analysis of tissue T_reg cells

Single-cell preparation was performed using GEM-X Flex (10x Genomics). $Foxp3^{AID/+}ROSA26^{WT}$ and $Foxp3^{AID/+}ROSA26^{TIR1(F74G)}$ mice were treated with 5-ph-IAA for 7 days (four per group) as described above. Animals were killed and perfused with 20 ml PBS. SLOs, lung and liver were collected and digested as described above. LILP was processed as follows. Dissected large intestines (from cecum to anus) were placed in 3 ml wash medium (RPMI 1640, 2% FBS, 10 mM HEPES buffer, 1% penicillin–streptomycin and 2 mM L-glutamine) on ice until they were ready to be processed. Samples were centrifuged at 700$g$ for 4 min at 4 °C unless mentioned otherwise. Each large intestine was placed in a 50-ml screw-cap tube with 25 ml wash medium supplemented with 5 mM EDTA and 1 mM dithiothreitol and shaken horizontally at 250 rpm for 15–20 min at 37 °C to remove the epithelial layer. After a brief vortex, the suspension was filtered through a tea strainer. Intact tissue from the filter was quickly rinsed in PBS and placed into new 50-ml tubes and digested by shaking in 25 ml wash medium supplemented with 0.2 U ml$^{-1}$ collagenase A and 1 U ml$^{-1}$ DNase I, along with four ceramic beads, at 250 rpm for 45 min at 37 °C. The suspension was then passed through a 100-µm strainer, centrifuged and the pellets were washed by centrifugation in 40% Percoll in wash medium. Percoll-washed cell pellets were washed once with staining buffer (PBS with 2% FBS and 2 mM EDTA) and used for downstream analysis.

Following dissociation into single-cell suspensions, cells were stained with fluorescent and TotalSeq-C anti-mouse hashtag antibodies (BioLegend) and washed extensively. For each genotype, cells from individual mice and tissues were stained with a unique hashtag antibody. Cells from each genotype were pooled and fixed in Fixation Buffer B (10x Genomics) at 4 °C for 24 h. After quenching and washing of cells per manufacturer instructions (10x Genomics, demonstrated protocol CG000782 Rev A), cells were resuspended in 500 µl Quenching Buffer B. Reporter-positive live/dead stain negative T_reg cells (CD45$^+$CD3$^+$TCRβ$^+$CD8α$^-$CD4$^+$ZsGreen$^+$mCherry$^+$ for $R26^{TIR1(F74G)}$ mice or mCherry$^-$ for $R26^{WT}$ mice) were isolated from fixed single-cell suspensions using FACS and sorted into PBS with 1% nuclease-free BSA Fraction V (Millipore Sigma, 126609) and 0.2 U µl$^{-1}$ RNase-inhibitor (Millipore Sigma, PN-3335399001). Following FACS isolation, cells were spun down, resuspended in 500 µl Quenching Buffer B, 50 µl Enhancer and 137.5 µl 50% glycerol, and stored at −80 °C until further processing. On the day of processing, cells were thawed and washed as per the manufacturer's instructions. Probe hybridization was performed as per the manufacturer's instructions. Cells from $ROSA26^{TIR1(F74G)}$ and $ROSA26^{WT}$ genotypes were hybridized with separately barcoded probes. The remaining steps were performed as per the manufacturer's instructions (10x Genomics, user guide CG000788 Rev A). In brief, cells were hybridized for 18 h and washed. Cells from this experiment and the $Foxp3^{DTR}$ versus $Tcrbd$ KO experiment, which were hybridized with unique barcoded probes, were pooled and encapsulated in one lane targeting 80,000 cells. Indexing PCR steps were performed using nine amplification cycles for both the gene expression and cell-surface protein libraries. Final sequencing libraries were sequenced on an Element Biosciences AVITI24 system (index 1, ten cycles; index 2, ten cycles; read 1, 28 cycles; and read 2, 90 cycles).

FASTQ files were processed using Cell Ranger v.9.0 and reads were aligned to the Chromium Mouse Transcriptome Probe Set v.1.1.0 GRCm39-2024-A (10x Genomics). Cells containing fewer than 800 UMI counts or 800 unique genes were filtered out. Cells containing more than 5% mitochondrial derived transcripts were filtered out. Genes that were expressed in >10 cells were retained for further analysis. Hashtag antibody data were demultiplexed using HashSolo with the following prior probabilities (negative, 0.01; singlet, 0.91; and doublet, 0.08). Cells called doublet or negative were filtered out. The resulting count matrix consisted of 32,709 cells × 19,087 genes and was normalized to median UMI counts. The normalized data were then ln(1 + counts)-transformed for downstream analysis.

### Single-cell RNA-seq analysis of adoptively transferred $T_{reg}$ cells

Single-cell preparation was performed using GEM-X Flex (10x Genomics). On the day of transfer (day 0), cells from $Foxp3^{AID/y}ROSA26^{WT}$ and $Foxp3^{AID/y}ROSA26^{TIR1(F74G)}$ were isolated from pooled SLOs and enriched for CD4$^+$ T cells as described above. $T_{reg}$ cells (CD4$^+$TCRβ$^+$ZsGreen$^+$, mCherry$^-$ for $ROSA26^{WT}$ and mCherry$^+$ for $ROSA26^{TIR1(F74G)}$) were FACS sorted. Similarly, CD4$^+$TCRβ$^+$CD44$^-$CD62L$^+$CD25$^-$ ($T_{conv}$) cells were FACS sorted from pooled SLOs of CD45.1 mice. Cells were pooled in appropriate ratios. First, a 1:1 mixture of $ROSA26^{WT}$ and $ROSA26^{TIR1(F74G)}$ $T_{reg}$ cells was created. A portion of this mixture was further pooled with an equal number of $T_{conv}$ cells. Distribution of cells in these pools was verified by flow cytometric analysis. Cells were washed twice with sterile PBS and resuspended at an appropriate concentration. For transfer into $Foxp3^{DTR}$CD45.1 mice, $1 \times 10^6$ cells from the $T_{reg}$ cell-only pool ($0.5 \times 10^6$ $ROSA26^{WT}$, $0.5 \times 10^6$ $ROSA26^{TIR1(F74G)}$) were injected retro-orbitally in 200 µl sterile PBS (four mice total). For transfer into $Tcrbd$ KO mice, $4 \times 10^6$ cells from the $T_{reg}$ + $T_{conv}$ pool ($1 \times 10^6$ $ROSA26^{WT}$, $1 \times 10^6$ $ROSA26^{TIR1(F74G)}$, $2 \times 10^6$ $T_{conv}$) were injected retro-orbitally in 200 µl sterile PBS (four mice total). For continuous endogenous $T_{reg}$ depletion, $Foxp3^{DTR}$CD45.1 were injected i.p. with 1,000 ng DT in 200 µl sterile PBS on day −1 and day 0 followed by 500 ng on day 3 and day 6. Continuous Foxp3 degradation was maintained by daily 5-ph-IAA injections as described above. On day 7, animals were killed. Pooled SLOs were collected from each mouse and enriched for CD4$^+$ T cells as described above. Following dissociation into single-cell suspensions, cells were stained with fluorescent and TotalSeq-C anti-mouse hashtag antibodies (BioLegend) and washed extensively. Cells from each mouse were stained with a unique hashtag antibody. Cells from individual $Foxp3^{DTR}$ mice were pooled and cells from individual $Tcrbd$ KO mice were pooled. Each pool was fixed in Fixation Buffer B (10x Genomics) at 4 °C for 20 h. After quenching and washing cells as per the manufacturer's instructions (10x Genomics, demonstrated protocol CG000782 Rev A), cells were resuspended in 500 µl Quenching Buffer B. Reporter-positive live/dead stain negative $T_{reg}$ cells (TCRβ$^+$CD8α$^-$CD4$^+$ZsGreen$^+$mCherry$^+$ or mCherry$^-$) were isolated from fixed single-cell suspensions using FACS and sorted into PBS with 1% nuclease-free BSA Fraction V (Millipore Sigma, 126609) and 0.2 U µl$^{-1}$ RNase inhibitor (Millipore Sigma, PN-3335399001). mCherry$^+$ and mCherry$^-$ cells were sorted into separate tubes for each condition. All sorted mCherry$^+$ cells were pooled and all sorted mCherry$^-$ cells were pooled. Probe hybridization was performed as per the manufacturer's instructions. Cells from mCherry$^+$ and mCherry$^-$ pools were hybridized with separately barcoded probes. Single-cell encapsulation and library preparation were performed together with the cross-tissue experiment as described above.

FASTQ files were processed using Cell Ranger v.9.0 and reads were aligned to the Chromium Mouse Transcriptome Probe Set v.1.1.0 GRCm39-2024-A (10x Genomics). Cells containing fewer than 800 UMI counts or 800 unique genes were filtered out. Cells containing more than 5% mitochondria-derived transcripts were filtered out. Genes that were expressed in >10 cells were retained for further analysis. Hashtag antibody data were demultiplexed using HashSolo with the following prior probabilities (negative, 0.01; singlet, 0.91; and doublet, 0.08). Cells called doublet or negative were filtered out. The resulting counts matrix consisted of 8,449 cells × 19,085 genes and was normalized to median UMI counts. The normalized data were then ln(1 + counts)-transformed for downstream analysis.

### Cluster frequency testing

Cluster enrichment was tested by calculating the fraction of cells from each sample belonging to each cluster. These fractions were calculated for all samples and a $t$-test was undertaken between conditions. $P$ values were false discovery rate (FDR)-corrected using a Benjamini–Hochberg correction. When single-cell samples for WT and TIR1 were matched (taken from the same mouse), a paired $t$-test was used.

### Computational analysis of resting and activated $T_{reg}$ cells

Active and resting cells were identified using bulk resting and activated $T_{reg}$ cell gene signatures from Van der Veeken et al.[12] (aTreg/rT$_{reg}$, logFC > 0.5; FDR < 0.05). Scoring was performed using the sc.tl.score_genes function. Then, thresholds were used to categorize cells as resting, active or neither. For the D3/D7 SLO scRNA-seq dataset, cells with an active score 0.5 × s.d. above the mean were categorized as active ($n = 6,731$). Non-active cells with a resting gene score higher than 0.25 × s.d. below the mean were categorized as resting ($n = 24,211$). All other cells were labeled as 'neither'. For the multi-tissue scRNA-seq dataset, dividing cells were first filtered out to account for different proportions among different tissues. Then, to account for differences across tissues, tissue-specific scoring was performed. Then, a cutoff was selected for each tissue to select the upper mode of cells after plotting the distribution of active scores. Specifically, the following cutoffs were used. For active cells, active was defined as cells with an active gene score >0.075 (LILP; $n = 7,882$), 0.02 (lung; $n = 1,850$), −0.01 (SLO; $n = 677$) and −0.01 (liver; $n = 1055$). For resting cells, resting was defined as non-active cells with a resting gene score > −0.075 (LILP), 0.05 (lung), 0.05 (SLO) and 0.1 (liver), and an active gene score < −0.12 (LILP), −0.12 (lung), −0.15 (SLO) and −0.15 (liver), resulting in the following number of resting cells: 25 (LILP), 4,202 (lung), 4,150 (SLO) and 2,581 (liver).

### Differential gene expression analysis

Differential gene expression for bulk RNA-seq data was carried out using DESeq2 v.1.42.0. Genes were filtered with a mean read count of 10. DESeq2 was run with the following formula after subsetting to samples from the same cell-type or condition: ~genotype. Differential genes were taken using an FDR cutoff of 0.05, except where otherwise stated. For single-cell data, raw read counts were pseudobulked by replicate and activation status, or for LILP data, by Leiden cluster. Genes with a mean expression per cell >5 × 10$^{-2}$ were considered. Then, raw read counts were passed into DESeq2 to identify differential genes at an FDR of 0.05. For the TCRbd/DTR transfer scRNA-seq dataset, cells from WT and TIR1 were paired (taken from the same mouse). To account for this effect, DESeq2 was run with the following formula: ~genotype + mouse_id.

### Foxp3 ChIP-seq and ATAC-seq data analysis

Foxp3 peaks were analyzed from Konopacki et al.[18] and lifted over to mm39 using pyliftover v.1.1.17. Foxp3 peaks were annotated to the nearest transcription start site (TSS) within 1 Mb using mm39 GENCODE GTF annotations. Only TSSs corresponding to genes with sufficient expression to be included in bulk RNA-seq analysis (mean read count >10) were included.

ATAC-seq reads were aligned using bowtie2. A peak atlas was generated by running MACS2 v.2.2.7.1. MACS2 was run with a $P$ value cutoff of 0.05, −nomodel, −shift = −100 and −extsize of 200. Irreproducible discovery rate was run on replicates with a threshold of 0.2. Reproducible peaks were merged into a single peak atlas and peak summits overlapping these peaks ($n = 248,805$) were used for quantification. Accessibility was quantified using Rsubread FeatureCounts v.2.8.2 and differential accessibility was calculated using DESeq2 v.1.42.0 with an FDR cutoff of 0.05.

## Motif analysis

The JASPAR2020 Mouse and Human Motif Databases were used to identify motifs at Foxp3 and ATAC-seq peaks using FIMO v.5.2.2 with a default $P$ value cutoff of $1 \times 10^{-4}$. Statistical enrichment for motifs was calculated using a Fisher exact test for differential enrichment within one peak set compared to a control peak set.

## FC–FC plots

For FC–FC plots between bulk RNA-seq experiments in the context of degradation and Foxp3 KO (Fig. 2), genes were labeled as shared if they shared the same sign in the log fold change and FDR < 0.05. For FC–FC plots between scRNA-seq experiments in the context of TCRbd/DTR or different organs, a more promiscuous threshold was applied to define 'shared' DEGs to account for the differences in statistical power between experimental conditions. First, all DEGs with an FDR < 0.05 across either condition were taken. A DEG (FDR < 0.05) in condition 1 was considered 'shared' in condition 2 if it met either of the following conditions in condition 2: (1) FDR < 0.2 with the same sign as in condition 1 or (2) |log fold change| > 0.5 with the same sign as in condition 1. The same conditions were applied for condition 2 DEGs with condition 1. This defined the set of all 'shared' DEGs. DEGs that were not shared were labeled as condition specific.

## Statistics

Statistical significance was determined using tests indicated in the respective figure legends. $P$ values for $t$-tests and analyses of variance were calculated with GraphPad Prism v.7 and were corrected for multiple hypothesis testing.

## Reporting summary

Further information on research design is available in the Nature Portfolio Reporting Summary linked to this article.

## Data availability

RNA-seq and ATAC-seq data generated in this study have been deposited in the Gene Expression Omnibus and are publicly available. Accession numbers are GSE297451 (bulk RNA-seq), GSE297597 (bulk ATAC-seq), GSE297598 (tumor scRNA-seq) and GSE297622 (tissue $T_{reg}$ cell and adoptive transfer scRNA-seq). Source data are provided with this paper.

## Materials availability

The corresponding authors can provide reagents and materials generated in this study upon request.

## Code availability

The code used for computational analysis is available at https://github.com/pritykinlab/scTir1_Analysis. The corresponding authors can provide additional information required to reanalyze the data reported in this paper upon request.

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

## Acknowledgements

We acknowledge the use of the Integrated Genomics Operation Core at the Memorial Sloan Kettering Cancer Center, funded by the National Cancer Institute Cancer Center Support Grant (P30 CA08748), Cycle for Survival and the Marie-Josée and Henry R. Kravis Center for Molecular Oncology. This work was supported by the National Institutes of Health (NIH)/National Cancer Institute Cancer Center Support Grant P30 CA008748 (A.Y.R.), NIH grant R01 AI034206 (A.Y.R.), NIH/National Institute of Allergy and Infectious Diseases grant DP2AI171161 and the Ludwig Institute for Cancer Research (Y.P.). A.Y.R. is an investigator with the Howard Hughes Medical Institute. E.Y.W. was supported by a Medical Scientist Training Program grant from the National Institute of General Medical Sciences of the NIH under award no. T32GM152349 to the Weill Cornell/Rockefeller/Sloan Kettering Tri-Institutional MD–PhD Program.

## Author contributions

W.H. and A.Y.R. conceived the project, interpreted the results and wrote the paper. Y.P., G.A.D. and E.Y.W. performed computational analyses of the sequencing data and wrote the relevant sections of the paper, with input from W.H. and A.Y.R. W.H., E.Y.W., G.B., Z.-M.W. and R.B.-P. conducted wet laboratory experiments and analyzed the data. A.P.G., L.F.K.U., R.B.-P., X.H., A.J.M., B.E.H. and W.J. assisted with wet laboratory experiments.

## Competing interests

A.Y.R. is a Scientific Advisory Board member and has equity in Sonoma Biotherapeutics, RAPT Therapeutics, Coherus BioSciences, Santa Ana Bio, Odyssey Therapeutics, Nilo Therapeutics and Vedanta Biosciences; he is also a Scientific Advisory Board member of BioInvent and Amgen and a co-inventor of a CCR8⁺ $T_{reg}$ cell depletion IP licensed to Takeda, which is unrelated to the content of this publication. Z.-M.W. is an employee of Genentech, which is unrelated to the content of this publication. The other authors declare no competing interests.

## Additional information

**Extended data** is available for this paper at https://doi.org/10.1038/s41590-025-02295-4.

**Correspondence and requests for materials** should be addressed to Wei Hu, Yuri Pritykin or Alexander Y. Rudensky.

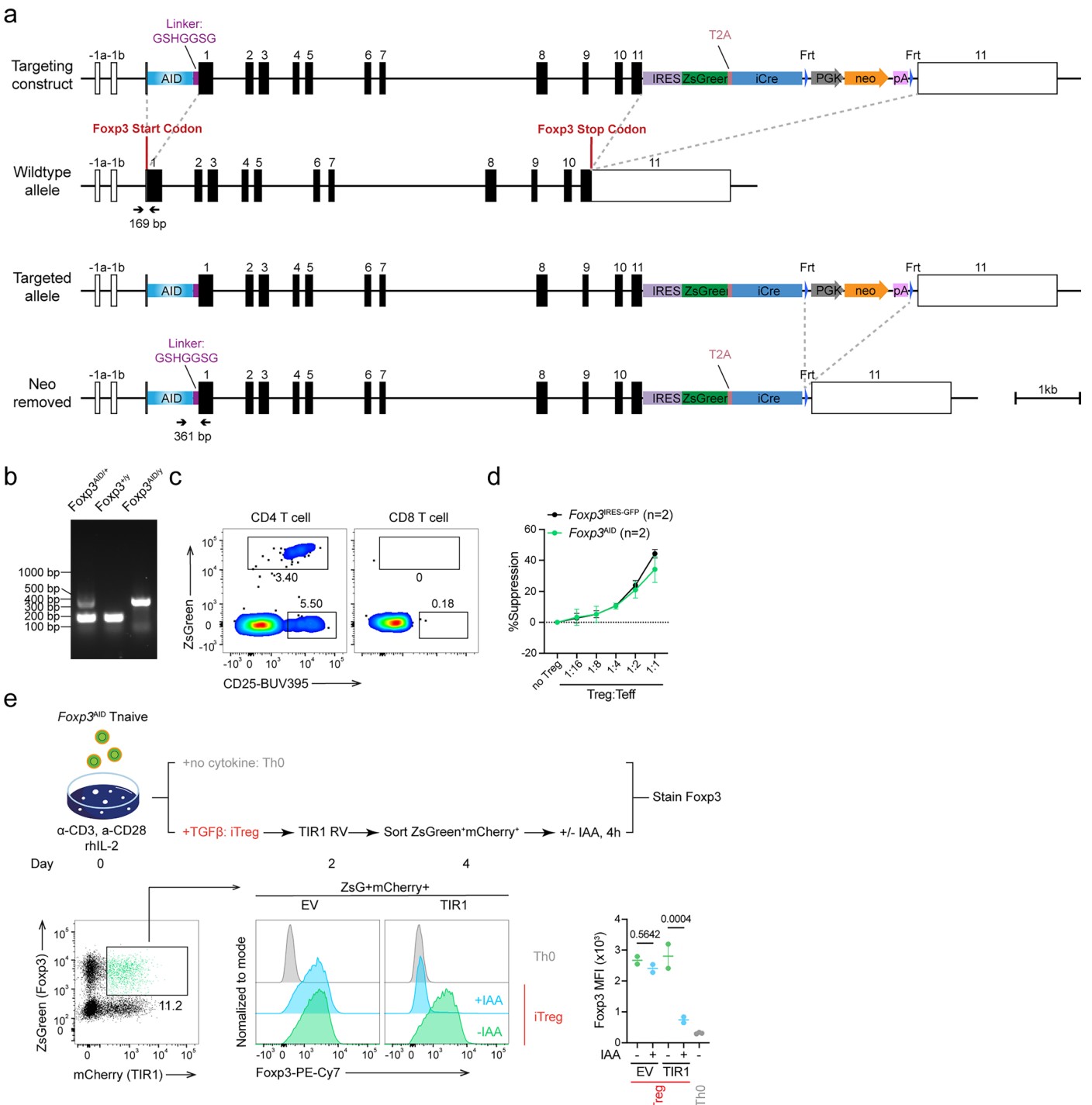

**Extended Data Fig. 1 | Generation of *Foxp3*[AID] mice. (a)** Gene targeting strategy. The auxin-inducible degron (AID) sequence was fused to the N-terminus of Foxp3 via a seven-amino-acid flexible linker. An IRES-ZsGreen-T2A-iCre-Frt-neo-Frt cassette was inserted into the 3′ UTR. Arrows indicate the locations of PCR primers used to distinguish *Foxp3*[WT] and *Foxp3*[AID] alleles. IRES: internal ribosome entry sequence. iCre: codon improved Cre recombinase. PGK: PGK promoter. Neo: neomycin-resistant gene. pA: bGH polyA sequence. **(b)** PCR validation of the knock-in allele. Data is representative of over 200 mice from each genotype. **(c)** Expression pattern of the ZsGreen reporter in *Foxp3*[AID/WT] heterozygous females. ZsGreen expression was restricted to CD25[+] CD4 T cells, consistent with Foxp3 expression. **(d)** ZsGreen[+] *Foxp3*[AID] T$_{reg}$ cells suppressed naïve CD4[+] T cell proliferation comparably to *Foxp3*[GFP] T$_{reg}$ cells *in vitro*. Line graph represents mean ± SEM of two biological replicates. **(e)** Naïve CD4[+] T cells from *Foxp3*[AID] mice were cultured under T$_{reg}$ inducing conditions and transduced with either a TIR1-encoding retrovirus or the empty vector control. AID-tagged Foxp3 protein was selectively degraded in TIR1-transduced induced T$_{reg}$ (iT$_{reg}$) cells upon indole acetic acid (IAA) treatment. Scatter plots represents mean ± SEM. Each point represents cells from a unique mouse.

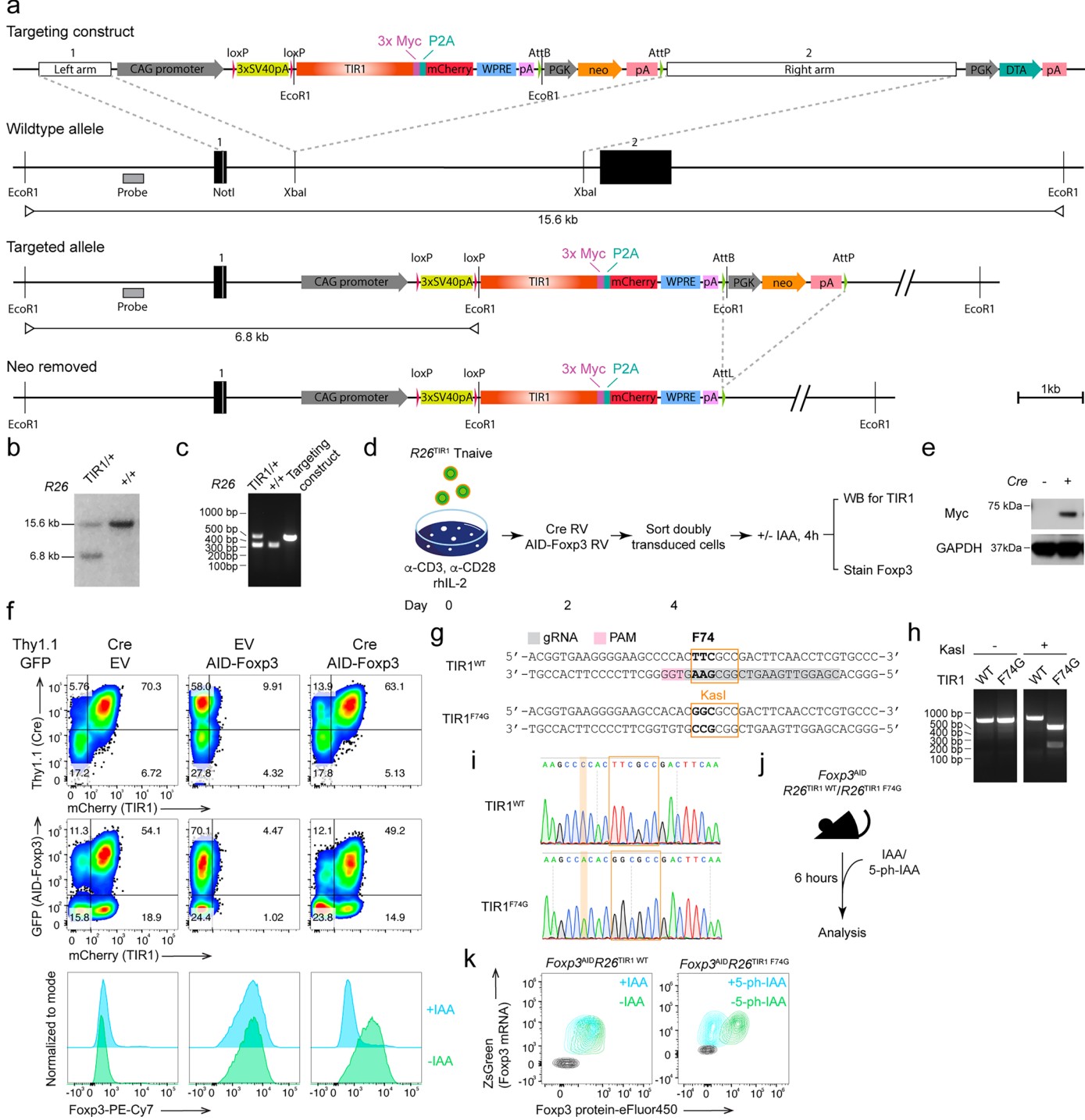

**Extended Data Fig. 2 | Generation of *ROSA26*[TIR1] and *ROSA26*[TIR1(F74G)] mice.**
(**a**) Gene targeting strategy. WPRE: Woodchuck hepatitis virus post-transcriptional regulatory element; DTA: Diphtheria toxin fragment A).
(**b**) Southern blot validation of heterozygous *ROSA26*[TIR1/+] mice using the hybridization probe shown in (**a**). Data is representative of two independent F1 mice. (**c**) PCR validation of *ROSA26*[TIR1/+] mice. Data are representative of over 100 mice from each genoype. (**d–f**) Naïve CD4+ T cells from *ROSA26*[TIR1/+] mice were co-transduced with retroviruses expressing Cre and AID-Foxp3 (**d**).

TIR1 expression was induced in a Cre-dependent manner (**e**), resulting in AID-Foxp3 degradation upon IAA treatment (**f**). Data are representative of two independent experiments. (**g**) Guide RNA (gRNA) design for CRISPR-mediated F74-to-G mutation in TIR1. The gRNA seed sequence is shown in gray; the PAM sequence is in pink. The F74G mutation creates a KasI restriction site. (**h–i**) Validation of the F74G mutation by KasI digestion (**h**) and Sanger sequencing (**i**). (**j–k**) The TIR1 F74G mutation enables *in vivo* protein degradation in response to 5-ph-IAA.

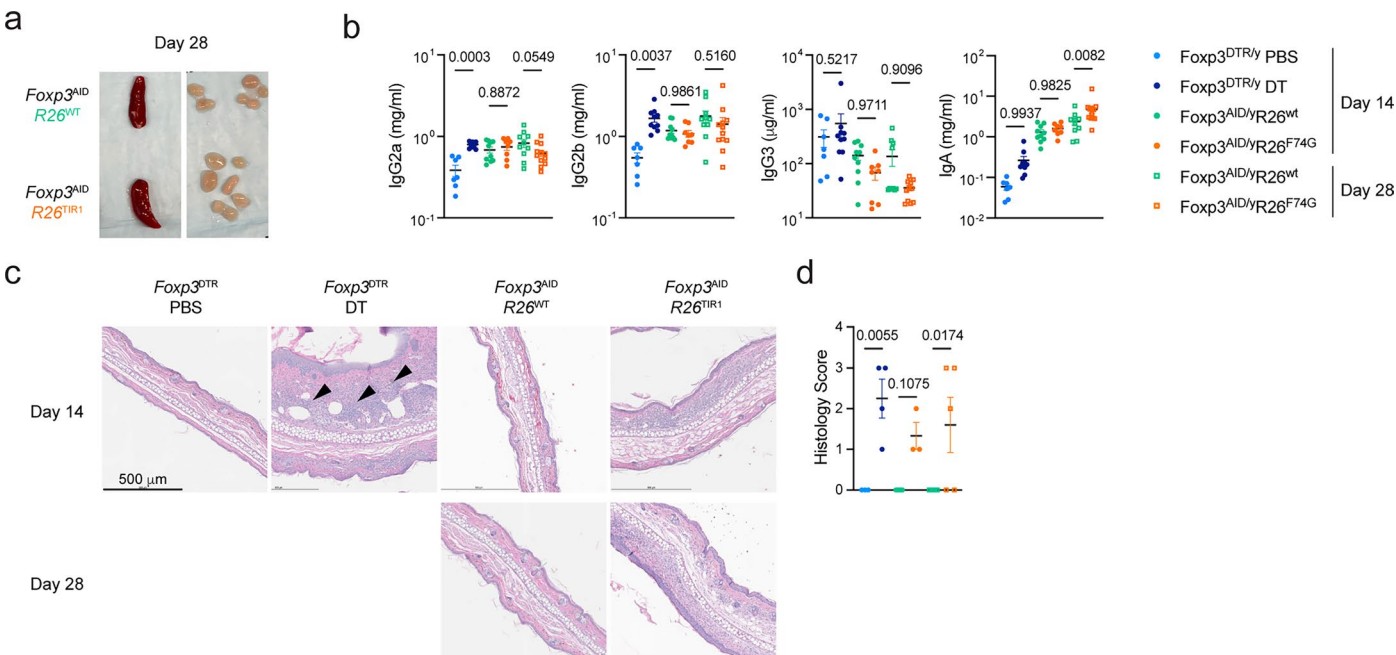

**Extended Data Fig. 3 | Foxp3 protein degradation in adult lymphoreplete mice induces minimal immune activation. (a)** Size of the spleen and lymph nodes after 28 days of Foxp3 degradation. **(b)** Serum antibody levels following T$_{reg}$ ablation or Foxp3 degradation. **(c,d)** Representative H&E stain **(c)** and histology scores **(d)** of the skin following T$_{reg}$ ablation or Foxp3 degradation. Each point represents a unique mouse. Data are pooled from two independent experiments. Scatter blots represent mean ± SEM. One-way ANOVA.

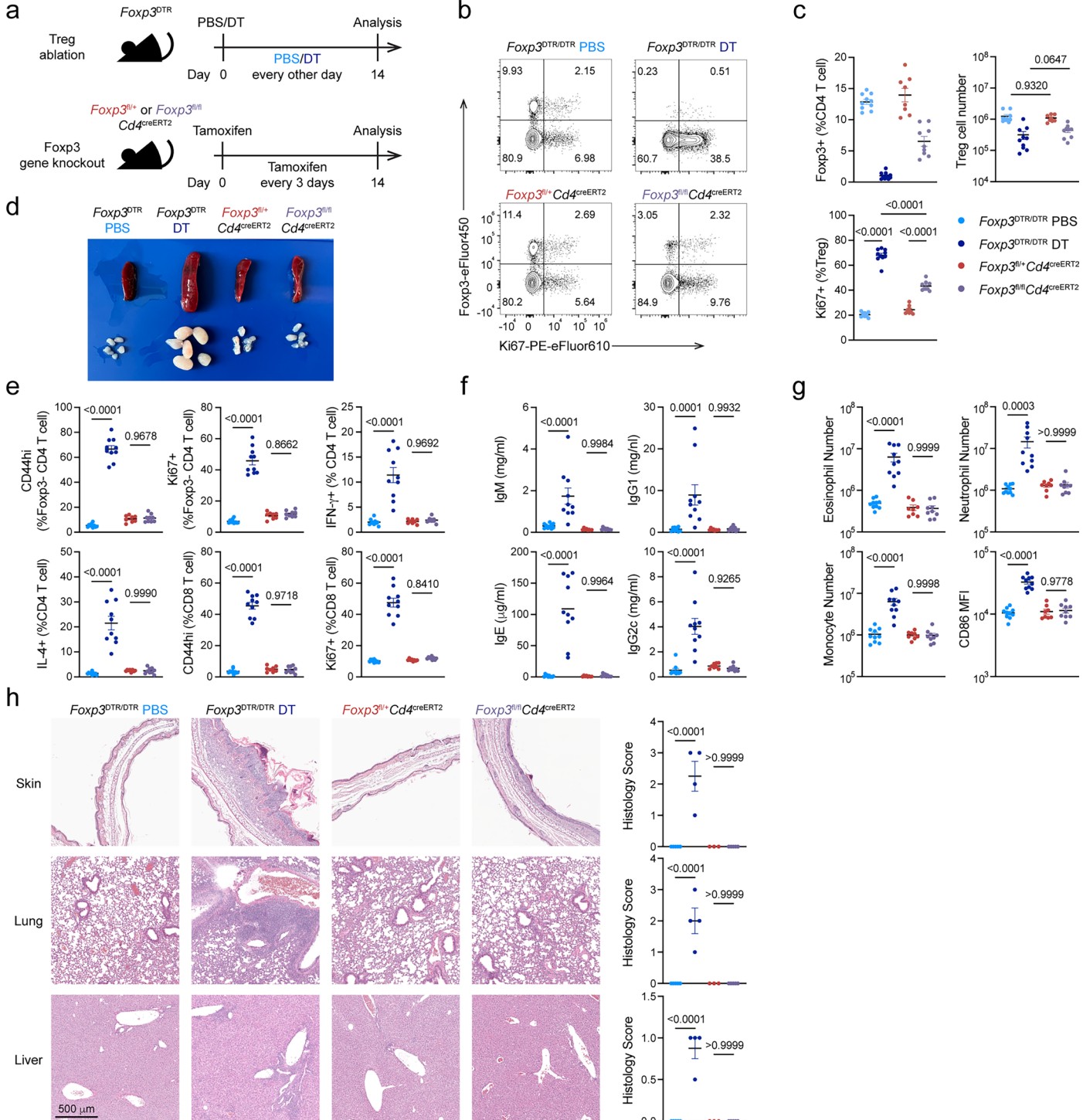

**Extended Data Fig. 4 | Inducible Foxp3 gene knockout causes minimal immune activation in adult lymphoreplete mice. (a)** Experimental design. **(b-c)** Representative plot (**b**) and combined data (**c**) showing the efficiency of Foxp3 gene knockout. **(d)** Size of the spleen and lymph nodes after 14 days of inducible Foxp3 gene knockout. **(e-g)** T cell activation (**e**), serum antibody

levels (**f**), and myeloid cell expansion (**g**) following 14 days of inducible Foxp3 gene knockout. **(h)** Representative H&E stain (left) and histology scores (right) of the skin, liver, and lung following 14 days of T_reg ablation or Foxp3 degradation. Each point represents a unique mouse. Data are pooled from two independent experiments. Scatter blots represent mean ± SEM. One-way ANOVA.

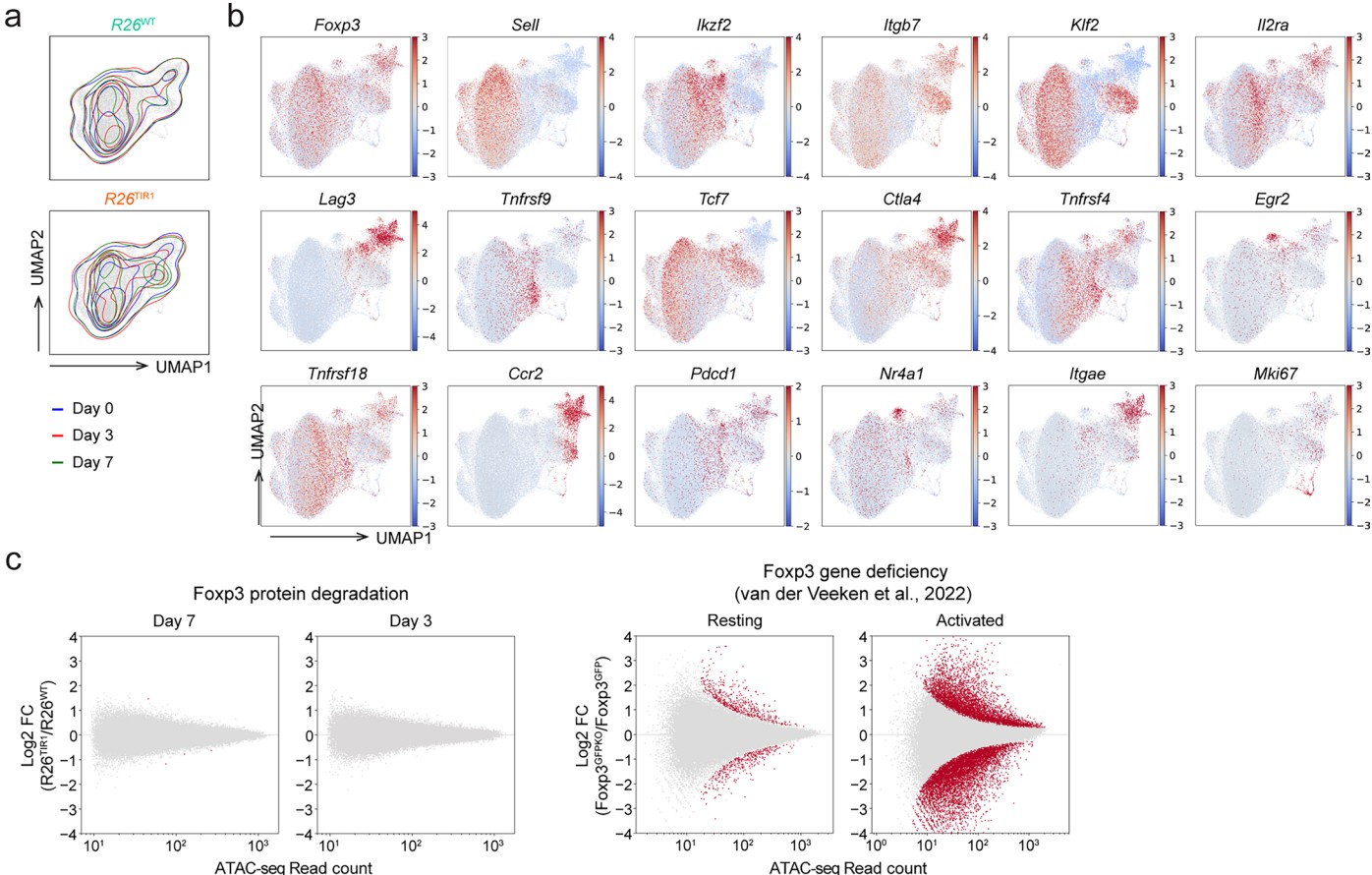

**Extended Data Fig. 5 | Foxp3 degradation in developed T$_{reg}$ cells induces minimal gene expression and chromatin accessibility changes. (a)** UMAP visualization of scRNA-seq data from *Foxp3*$^{AID}$*R26*$^{WT}$ and *Foxp3*$^{AID}$*R26*$^{TIR1(F74G)}$ T$_{reg}$ cells on days 0, 3 and 7 of 5-ph-IAA-induced Foxp3 degradation. **(b)** UMAP visualization of representative genes from the single-cell RNA-seq dataset colored by expression level. **(c)** MA plot showing differentially accessible ATAC-seq peaks induced by Foxp3 degradation (left) and Foxp3 gene deficiency (right). Red points represent differentially accessible regions.

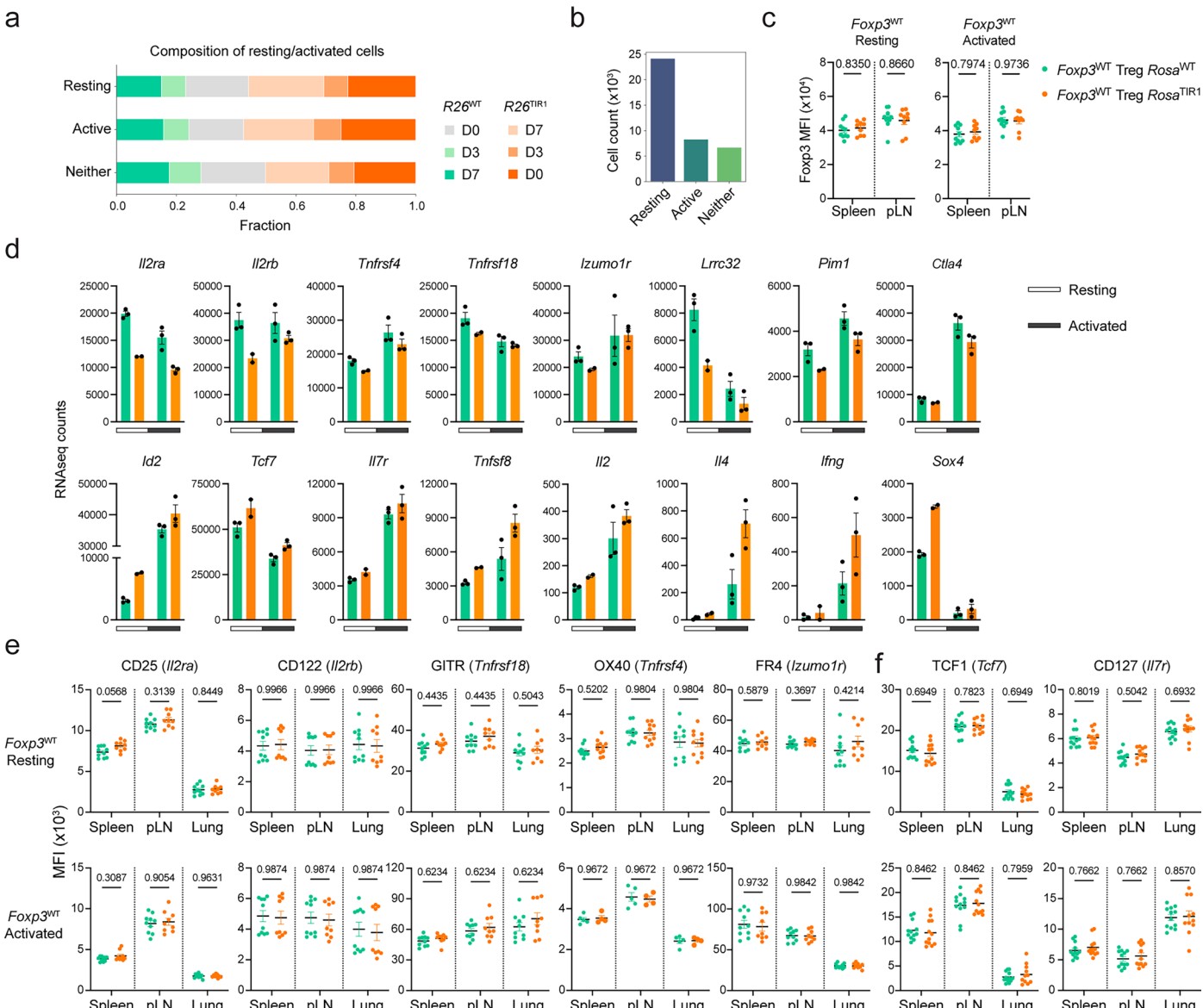

**Extended Data Fig. 6 | Foxp3 degradation in developed T$_{reg}$ cells altered the expression of a small group of genes. (a)** The composition of *Foxp3*$^{AID}$*R26*$^{WT}$ and *Foxp3*$^{AID}$*R26*$^{WT(F74G)}$ T$_{reg}$ cells within resting and activated subsets from day 0, day 3, and day 7 of Foxp3 degradation. **(b)** Number of resting and activated T$_{reg}$ cells utilized for the differential gene expression analysis in Fig. 3. **(c)** Flow cytometry analysis of Foxp3 protein and mRNA levels (reported by ZsGreen) in *Foxp3*$^{WT}$ T$_{reg}$ cells from heterozygous *Foxp3*$^{AID/WT}$*R26*$^{WT}$ and *Foxp3*$^{AID/WT}$*R26*$^{TIR1(F74G)}$ females after 7 days of Foxp3 degradation. Scatter plots represent mean ± SEM. Data are pooled from two independent experiments. Two-way ANOVA. **(d)** Bulk RNA-seq read counts of indicated genes in *Foxp3*$^{AID}$ T$_{reg}$ cells from heterozygous *Foxp3*$^{AID/WT}$*R26*$^{WT}$ and *Foxp3*$^{AID/WT}$*R26*$^{TIR1(F74G)}$ females after 7 days of Foxp3 degradation. Each point represents a unique mouse. Bar graphs represent mean ± SEM. **(e)** Flow cytometry analysis of CD25, CD122, OX40, GITR, and FR4 protein levels in *Foxp3*$^{WT}$ T$_{reg}$ cells from heterozygous *Foxp3*$^{AID/WT}$*R26*$^{WT}$ and *Foxp3*$^{AID/WT}$*R26*$^{TIR1(F74G)}$ females after 7 days of Foxp3 degradation. **(f)** Flow cytometry analysis of CD127 and TCF1 protein levels in *Foxp3*$^{AID}$ T$_{reg}$ cells from heterozygous *Foxp3*$^{AID/WT}$*R26*$^{WT}$ and *Foxp3*$^{AID/WT}$*R26*$^{TIR1(F74G)}$ females after 7 days of Foxp3 degradation. **(e-f)** Scatter plots represent mean ± SEM. Each point represents a unique mouse. Data are pooled from two independent experiments. Multiple *t*-tests.

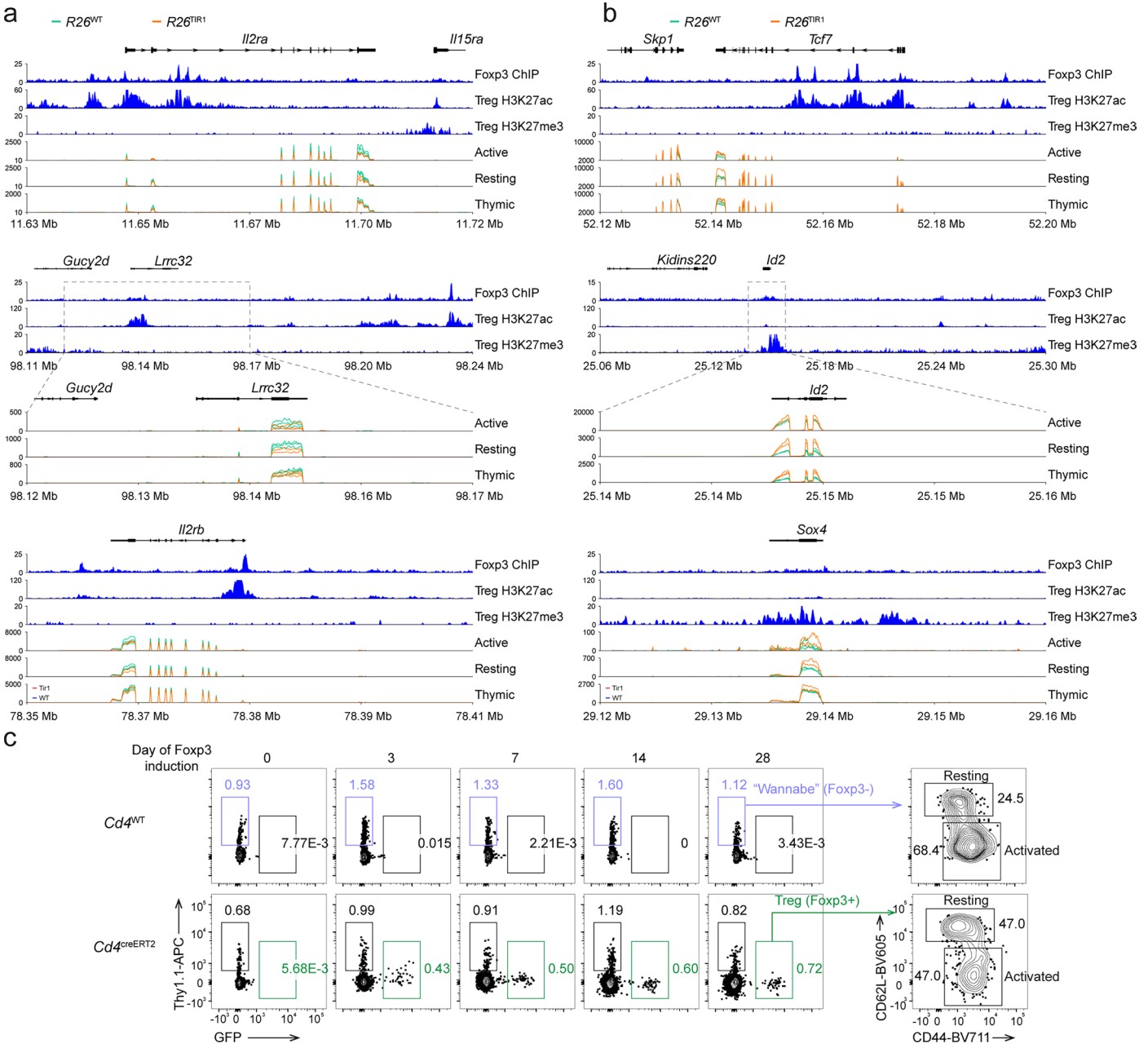

**Extended Data Fig. 7 | Genes sensitive to Foxp3 degradation in mature T_regs are potentially enriched for direct Foxp3 targets. (a–b)** Representative tracks showing Foxp3 ChIP, T_reg H3K27Ac, T_reg H3K27me3, and RNA-seq profiles in activated, resting, and nascent thymic T_reg cells for candidate Foxp3-activated **(a)** and Foxp3-repressed **(b)** genes. **(c)** Representative flow cytometry plots and gating strategy for RNA-seq analysis of T_reg "wannabe" cells with or without Foxp3 induction at the indicated time points following 4-OHT administration.

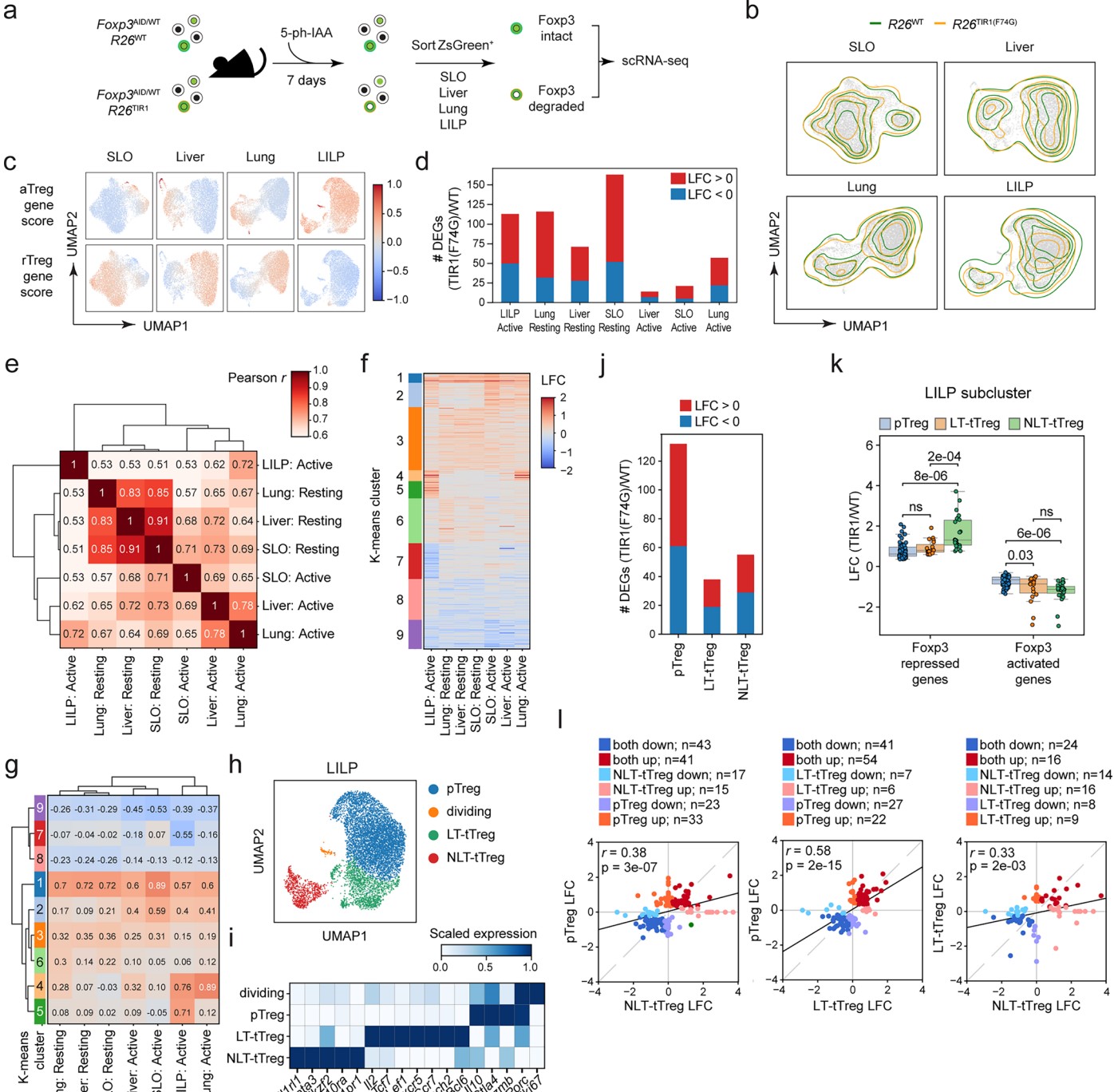

**Extended Data Fig. 8 | Conservation of Foxp3 degradation effects across different tissues. (a)** Experimental design of single-cell RNA-seq experiment. Each genotype consisted of four independent biological replicates. **(b)** Density contour plots of $Foxp3^{AID}R26^{WT}$ and $Foxp3^{AID}R26^{TIR1(F74G)}$ $T_{reg}$ cells from SLO, liver, lung and LILP overlaid on UMAP embeddings. **(c)** UMAP visualization of $T_{reg}$ cells from depicted tissues colored by "resting" and "activated" gene signature scores based on the scoring strategy from Fig. 3a. **(d)** Bar plots showing the number of DEGs between $R26^{WT}$ and $R26^{TIR1(F74G)}$ in resting and activated $T_{reg}$ cells from indicated tissues, colored by up- or downregulation. **(e)** Pairwise Pearson correlation coefficients for log₂ fold changes of DEGs between activated and resting $T_{reg}$ cells from each tissue. Rows and columns are hierarchically clustered. **(f)** Heatmap of log₂ fold changes in Foxp3 degradation DEGs separated by $T_{reg}$ cell activation status and tissue of origin. Genes are clustered by k-means clustering and cluster numbers are indicated by the color bars on the left. **(g)** Heatmap of average log₂ fold changes for each k-means cluster in $T_{reg}$ cells separated by tissue and activation status. Rows and columns are hierarchically

clustered. **(h)** Leiden clustering results visualized by UMAP for LILP $T_{reg}$ cells. Clusters are labeled as peripheral $T_{reg}$ cell (p$T_{reg}$), dividing $T_{reg}$ cell, lymphoid tissue thymic $T_{reg}$ cell (LT-t$T_{reg}$) and non-lymphoid tissue thymic $T_{reg}$ cell (NLT-t$T_{reg}$) based on differences in their gene expression profile. **(i)** Heatmap of scaled mean UMI-normalized expression values of indicated genes in each cluster. **(j)** Number of DEGs between $R26^{WT}$ and $R26^{TIR1(F74G)}$ $T_{reg}$ cells in p$T_{reg}$, LT-t$T_{reg}$ and NLT-t$T_{reg}$ cells, colored by up- or downregulation. **(k)** Log₂ fold changes of "Foxp3-repressed" and "Foxp3-activated" genes in p$T_{reg}$, LT-t$T_{reg}$ and NLT-t$T_{reg}$ cells. Each point represents a gene from the Foxp3-repressed or Foxp3-activated gene set identified in Fig. 2. Horizontal line marks the median and box marks the interquartile range. Mann-Whitney U test. **(l)** Scatter-plot of log₂ fold changes of DEGs between p$T_{reg}$ and NLT-t$T_{reg}$ cell clusters (left-top), p$T_{reg}$ and LT-t$T_{reg}$ cell clusters (left-bottom) and LT-t$T_{reg}$ and NLT-t$T_{reg}$ cell clusters (right-top). Points are colored by their direction and populations in which they are altered. Correlations were calculated with Pearson correlation over all DEGs.

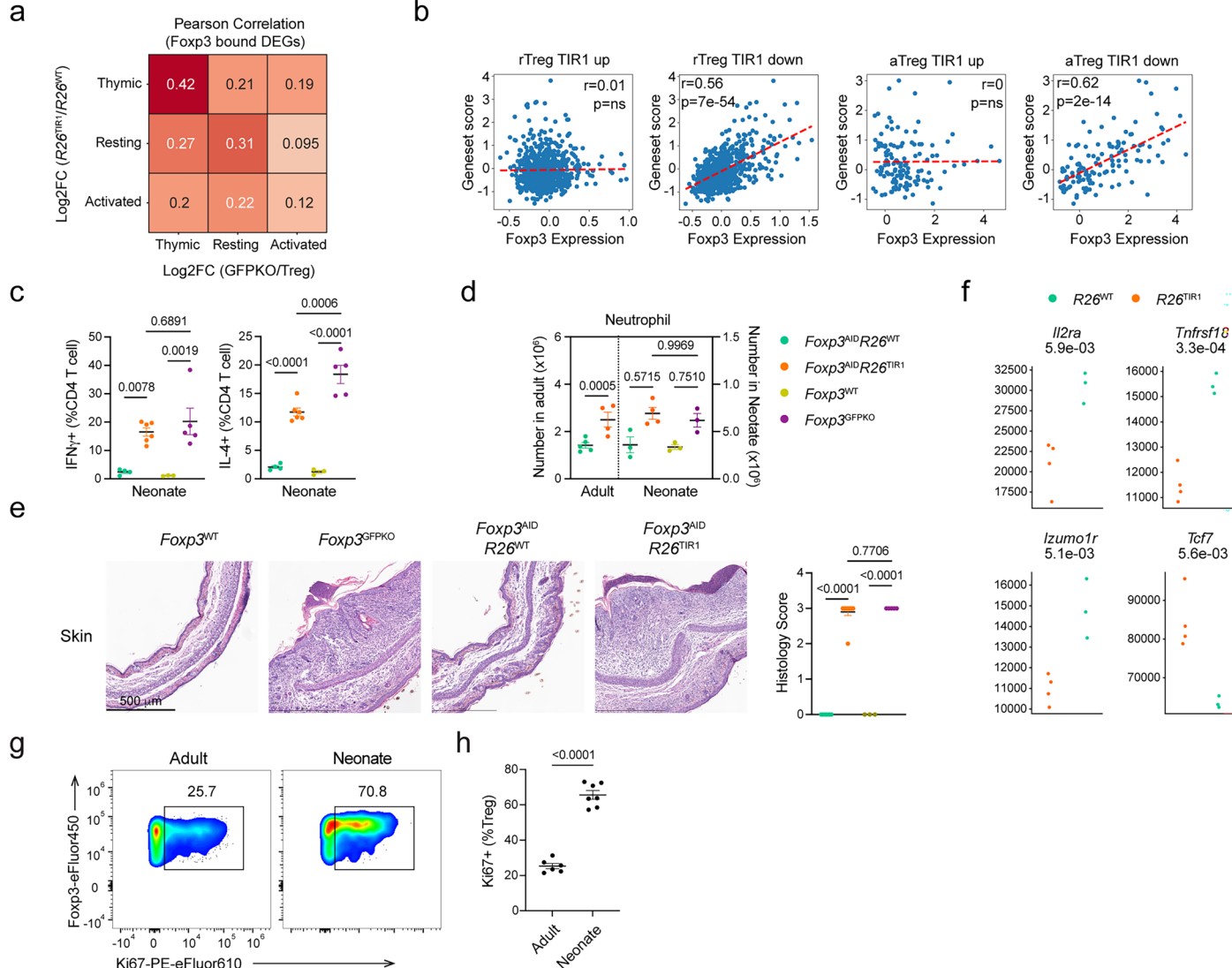

**Extended Data Fig. 9 | Foxp3 is preferentially required during T_reg cell development. (a)** Pearson correlation between Foxp3 degradation-induced and Foxp3-dependent DEGs in thymic, resting, and activated T_reg cells, limited to Foxp3-bound genes. **(b)** Meta-cell analysis of resting and activated T_reg scRNA-seq data from secondary lymphoid organs following Foxp3 degradation, correlating Foxp3 expression levels with "TIR1-up" and "TIR1-down" gene signatures identified in Fig. 3. Dashed red line depicts line of best fit. Correlations and corresponding p-values were calculated with Pearson correlation over all genes. **(c)** Cytokine production by CD4+ T cells from neonatal *Foxp3*^AID mice after 14 days of Foxp3 degradation, in comparison to age-matched *Foxp3*^WT and *Foxp3*^GFPKO mice. **(d)** Neutrophil expansion in adult and neonatal *Foxp3*^AID mice following 14 days of Foxp3 degradation. Age-matched *Foxp3*^WT and *Foxp3*^GFPKO mice serve

as controls for neonatal *Foxp3*^AID mice. **(e)** Representative H&E staining and histology scores of skin inflammation in neonatal *Foxp3*^AID mice after 14 days of Foxp3 degradation, in comparison to age-matched *Foxp3*^WT and *Foxp3*^GFPKO mice. **(c–e)** Scatter plots represent mean ± SEM. Each point represents a unique mouse. Data are pooled from two independent experiments. One-way ANOVA. **(f)** Scatter plots of DESeq2 normalized counts of select DEGs from the neonatal Foxp3 degradation RNA-seq experiment (Fig. 5m). Each point represents a unique mouse. Two-sided *t*-test. **(g-h)** Representative plots **(g)** and combined data **(h)** showing T_reg proliferation in 8-week-old adult versus 7-day-old neonatal mice measured by Ki67 positivity. Scatter-plot represents mean ± SEM. Data are combined from two independent experiments. Two-tailed t-test.

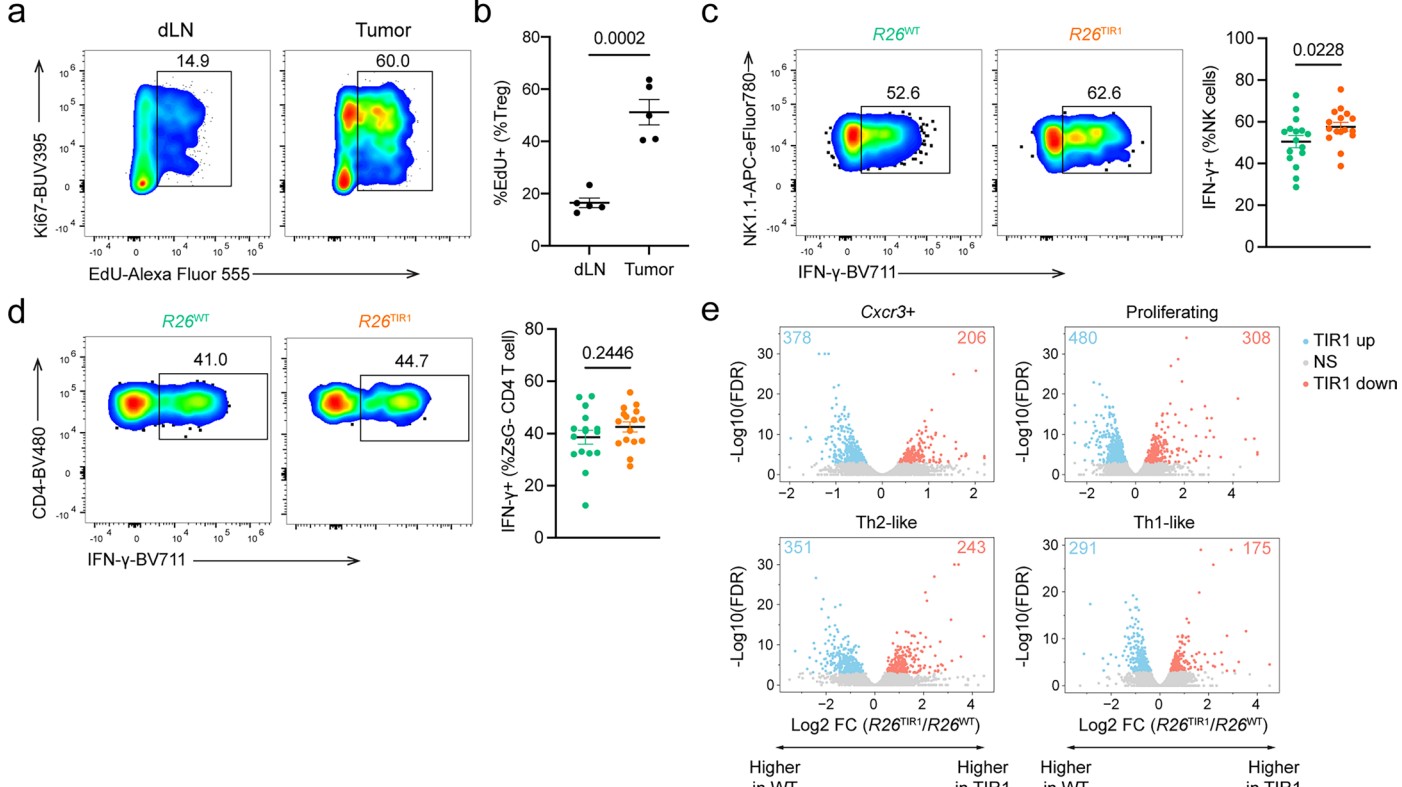

**Extended Data Fig. 10 | Foxp3 degradation leads to tumor shrinkage with minimal adverse effects. (a–b)** Representative flow cytometry plots (**a**) and combined data (**b**) of Ki67 expression and EdU incorporation in $T_{reg}$ cells from the tumor and tumor-draining lymph node (dLN). Scatter-plot shows mean ± SEM. Each point represents a unique mouse. Data are representative of two independent experiments. Two-tailed *t*-test. **(c)** Representative flow cytometry plots (left) and combined data (right) of IFN-γ production by tumor-infiltrating NK cells. Scatter-plot shows mean ± SEM. Each point represents a

unique mouse. Data are pooled from two independent experiments. Two-tailed *t*-test. **(d)** Representative flow cytometry plots (left) and quantification (right) of IFN-γ production by tumor-infiltrating ZsGreen⁻ CD4 T cells. Scatter-plot shows mean ± SEM. Each point represents a unique mouse. Data are pooled from two independent experiments. Two-tailed *t*-test. **(e)** Volcano plts showing the number of genes up- and downregulated in *Foxp3*ᴬᴵᴰ*R26*ᵀᴵᴿ¹⁽ᶠ⁷⁴ᴳ⁾ tumor $T_{reg}$ cells within each clustered defined in Fig. 7(I).

# Reporting Summary

## Statistics

For all statistical analyses, confirm that the following items are present in the figure legend, table legend, main text, or Methods section.

| n/a | Confirmed | |
|---|---|---|
| ☐ | ☒ | The exact sample size (*n*) for each experimental group/condition, given as a discrete number and unit of measurement |
| ☐ | ☒ | A statement on whether measurements were taken from distinct samples or whether the same sample was measured repeatedly |
| ☐ | ☒ | The statistical test(s) used AND whether they are one- or two-sided<br>*Only common tests should be described solely by name; describe more complex techniques in the Methods section.* |
| ☐ | ☒ | A description of all covariates tested |
| ☐ | ☒ | A description of any assumptions or corrections, such as tests of normality and adjustment for multiple comparisons |
| ☐ | ☒ | A full description of the statistical parameters including central tendency (e.g. means) or other basic estimates (e.g. regression coefficient) AND variation (e.g. standard deviation) or associated estimates of uncertainty (e.g. confidence intervals) |
| ☐ | ☒ | For null hypothesis testing, the test statistic (e.g. *F*, *t*, *r*) with confidence intervals, effect sizes, degrees of freedom and *P* value noted<br>*Give P values as exact values whenever suitable.* |
| ☒ | ☐ | For Bayesian analysis, information on the choice of priors and Markov chain Monte Carlo settings |
| ☒ | ☐ | For hierarchical and complex designs, identification of the appropriate level for tests and full reporting of outcomes |
| ☐ | ☒ | Estimates of effect sizes (e.g. Cohen's *d*, Pearson's *r*), indicating how they were calculated |

*Our web collection on statistics for biologists contains articles on many of the points above.*

## Software and code

Policy information about availability of computer code

| Data collection | Flow cytometry data were collected on an Aurora cytometer (Cytek) using SpectroFlo v3.1.2 (Cytek).<br>ELISA data were recorded with Synergy HTX plate reader with Gen5 v3.02.2(BioTek).<br>Bulk RNA-seq libraries were sequenced on NovaSeq 6000 (Illumina).<br>scRNA-seq and scATAC-seq libraries were sequenced on an NovaSeq 6000 System (Illumina). |
|---|---|
| Data analysis | Flow cytometry data were analyzed using FlowJo v10.6.1 (BD).<br>ELISA data were calculated with Gen5 3.02.2 (BioTek).<br>Statistical analyses of biological experiments were performed using Prism v10.0.<br>For RNA-seq experiment, STAR aligner and Genome Analysis Toolkit were used for alignment, R was used for measuring the raw count of reads per gene, and the DESeq2 R package was used to perform differential gene expression analysis.<br>For scRNA-seq, fastq files were processed using Cell Ranger (10x Genomics).<br>DNA sequence analysis was performed using SnapGene v8.0.1.<br>The code used for computational analysis is available at https://github.com/pritykinlab/scTir1_Analysis . The lead contacts can provide and any additional information required to reanalyze the data reported in this paper upon request. |

For manuscripts utilizing custom algorithms or software that are central to the research but not yet described in published literature, software must be made available to editors and reviewers. We strongly encourage code deposition in a community repository (e.g. GitHub). See the Nature Portfolio guidelines for submitting code & software for further information.

## Data

Policy information about <u>availability of data</u>

All manuscripts must include a <u>data availability statement</u>. This statement should provide the following information, where applicable:

- Accession codes, unique identifiers, or web links for publicly available datasets
- A description of any restrictions on data availability
- For clinical datasets or third party data, please ensure that the statement adheres to our <u>policy</u>

RNA-seq, and ATAC-seq data generated in this study have been deposited in GEO and are publicly available. Accession numbers are listed below.
Bulk RNA-seq: GSE297451
Bulk ATAC-seq: GSE297597
Tumor scRNA-seq: GSE297598
Tissue Treg and adoptive transfer scRNA-seq: GSE297622

## Research involving human participants, their data, or biological material

Policy information about studies with <u>human participants or human data</u>. See also policy information about <u>sex, gender (identity/presentation), and sexual orientation</u> and <u>race, ethnicity and racism</u>.

| | |
|---|---|
| Reporting on sex and gender | *Use the terms sex (biological attribute) and gender (shaped by social and cultural circumstances) carefully in order to avoid confusing both terms. Indicate if findings apply to only one sex or gender; describe whether sex and gender were considered in study design; whether sex and/or gender was determined based on self-reporting or assigned and methods used. Provide in the source data disaggregated sex and gender data, where this information has been collected, and if consent has been obtained for sharing of individual-level data; provide overall numbers in this Reporting Summary. Please state if this information has not been collected. Report sex- and gender-based analyses where performed, justify reasons for lack of sex- and gender-based analysis.* |
| Reporting on race, ethnicity, or other socially relevant groupings | *Please specify the socially constructed or socially relevant categorization variable(s) used in your manuscript and explain why they were used. Please note that such variables should not be used as proxies for other socially constructed/relevant variables (for example, race or ethnicity should not be used as a proxy for socioeconomic status). Provide clear definitions of the relevant terms used, how they were provided (by the participants/respondents, the researchers, or third parties), and the method(s) used to classify people into the different categories (e.g. self-report, census or administrative data, social media data, etc.) Please provide details about how you controlled for confounding variables in your analyses.* |
| Population characteristics | *Describe the covariate-relevant population characteristics of the human research participants (e.g. age, genotypic information, past and current diagnosis and treatment categories). If you filled out the behavioural & social sciences study design questions and have nothing to add here, write "See above."* |
| Recruitment | *Describe how participants were recruited. Outline any potential self-selection bias or other biases that may be present and how these are likely to impact results.* |
| Ethics oversight | *Identify the organization(s) that approved the study protocol.* |

Note that full information on the approval of the study protocol must also be provided in the manuscript.

# Field-specific reporting

Please select the one below that is the best fit for your research. If you are not sure, read the appropriate sections before making your selection.

☒ Life sciences        ☐ Behavioural & social sciences        ☐ Ecological, evolutionary & environmental sciences

For a reference copy of the document with all sections, see <u>nature.com/documents/nr-reporting-summary-flat.pdf</u>

# Life sciences study design

All studies must disclose on these points even when the disclosure is negative.

| | |
|---|---|
| Sample size | No sample size calculation was performed. Whenever possible at least 2 independent biological replicates were analyzed. In vivo experiments were performed at least twice to ensure reproducibility. |
| Data exclusions | In the bulk RNA-seq experiment in Figure 2, three biological replicates of each cell type were generated but Foxp3-AID R26-TIR1 resting Treg ended up being represented by only two because the third replicate seemed to be an outlier based on principal component analysis. |
| Replication | All in vivo experiments were repeated at least twice and all attempts at replication were successful. |
| Randomization | Mice were grouped according to genotype and all experiments were performed with sex-matched littermates. |

| Blinding | For pathology scoring, the pathologist was blinded from the genotype and treatment of the mice. |

# Reporting for specific materials, systems and methods

We require information from authors about some types of materials, experimental systems and methods used in many studies. Here, indicate whether each material, system or method listed is relevant to your study. If you are not sure if a list item applies to your research, read the appropriate section before selecting a response.

## Materials & experimental systems

| n/a | Involved in the study |
|---|---|
| ☐ | ☒ Antibodies |
| ☐ | ☒ Eukaryotic cell lines |
| ☐ | ☐ Palaeontology and archaeology |
| ☐ | ☒ Animals and other organisms |
| ☐ | ☐ Clinical data |
| ☐ | ☐ Dual use research of concern |
| ☐ | ☐ Plants |

## Methods

| n/a | Involved in the study |
|---|---|
| ☐ | ☐ ChIP-seq |
| ☐ | ☒ Flow cytometry |
| ☐ | ☐ MRI-based neuroimaging |

## Antibodies

| Antibodies used | The following antibodies and reagents were used in this study for flow cytometry, with clones, venders, catalog numbers and dilutions as indicated: anti-Siglec-F (E50-2440, BD, 562681, 1:400), anti-I-A/I-E (M5/114.15.2, Biosciences, 566086, 1:1,200), anti-NK1.1 (PK136, Thermo Fisher, 47-5941-82, 1:400), anti-CD45 (30-F11, BioLegend, 103136, 1:600), anti-CD11b (M1/70, BioLegend, 101257, 1:800), anti-CD11b (M1/70, BD Biosciences, 363-0112-82, 1:400), anti-CD3ε (17A2, BioLegend, 100237, 1:500), anti-γδTCR (GL3, BD Biosciences, 750410, 1:300), anti-Cd278 (C398.4A, BD Biosciences, 567918, 200), anti-TCR beta (H57-597, BD Biosciences, 748405, 1:300), Anti-TCR beta (H57-597, Thermo Fisher, 47-5961-82, 1:300), anti-TCR beta (H57-597, BioLegend, 109227, 1:200), anti-TCR beta ( H57-597, Thermo Fisher, 12-5961-83, 1:400), anti-CD153 (RM153, BD Bioscience, 741575, 1:400), anti-CD24 (M1/69, Thermo Fisher, 46-0242-82, 1:800), anti-CD304 (3E12, BioLegend, 145209, 1:300), anti-CD3 (IM7, BioLegend, 103049, 1:400), anti-CD44 (IM7, BD Biosciences, 563971, 1:400), anti-CD44 (IM7, BioLegend, 103026, 1:100), anti-ZsGreen (polyclonal, Frontier Institute co.,ltd, MSFR106470, 1:800), anti-KLRG1 (2F1, Thermo Fisher, 35-5893-82), anti-CD39 (24DMS1, Thermo Fisher, 25-0391-82, 1:400), anti-TCF1 (C63D9, Cell signaling, 6709, 1:200), anti-IL-10 (JES5-16E3, BioLegend, 505021, 1:200), anti-CD4 (RM4-5, BD Biosciences, 414-0042-82, 1:400), anti-CD4 (RM4-5, BioLegend, 100536, 1:400), anti-CD4 (RM4-5, Thermo Fisher, 47-0042-82, 1:400), anti-CD4 (RM4-5, BioLegend, 100548, 1:400), anti-TNFa (MP6-XT22, BioLegend, 506329, 1:400), anti-IFNg (XMG1.2, BioLegend, 505836, 1:200), anti-IL-22 (1H8PWSR, Thermo Fisher, 46-7221-80, 1:400), anti-IL13 (eBio13A, Thermo Fisher, 12-7133-82, 1:400), anti-IL-4 (11B11, Thermo Fisher, 17-7041-82, 1:300), anti CD11c (N418, Thermo Fisher, 48-0114-82, 1:200), anti-Ly6c (HK1.4, BioLegend, 128037, 1:1,200), anti-Ly6C (HK1.4, BioLegend, 128041, 1:1000), anti-CD122 (TM-β1, BD Bioscience, 564763, 1:200), anti-GARP (YGIC86, Thermo Fisher, 25-9891, 1:200), anti-CD86 (GL1, Thermo Fisher, 12-0862-85, 1:400), anti-Ly6G (1A8, BioLegend, 127618, 1:500), anti-CD64 (X54-5/7.1, BioLegend, 139306, 1:200), anti-CD127 (A7R34, Tonbo Bioscience, 20-1271-U100, 1:200), anti-CD122 (5H4, Thermo Fisher, 13-1221-82, 1:200), anti-Guinea Pig (polyclonal, Thermo Fisher, SA5-10094, 1:1,000), anti-FR4 (12A5, BD Biosciences, 744121, 1:200), anti-FR4 (12A5, BD Biosciences, 560318, 1:200), anti-OX40 (OX-86, Thermo Fisher, 46-1341-82, 1:300), anti-CD120b (TR75-89, BD Bioscience, 564088, 1:200, anti-CD103 (M290, BD Biosciences, 566118, 1:300), anti-Ly-6C (HK1.4, BioLegend, 128037, 1:1,000), anti-CD90.2 (30-H12, BioLegend, 105320, 1:800), anti-CD90.2 (53-2.1, BD Biosciences, 564365, 1:1,500), anti-Foxp3 (FJK-16s, Thermo Fisher, 48-5773-82, 1:200), anti-Foxp3 (FJK-16s, Thermo Fisher, 17-5773-82, 1:200), anti-CD19 (6D5, BioLegend, 115510, 1:600), anti-F4|80 (BM8, BioLegend, 123133, 1:200), anti-CD4 (RM4-5, EBioscience, 564667, 1:400), anti-CD4 (RM4-5, BioLegend, 100553, 1:400), anti-CD8α (53-6.7, BioLegend, 100780, 1:600), anti-CD8α (53-6.7, BioLegend, 564297, 1:400), anti-CD8α (53-6.7, BioLegend, 100752, 1:500), anti-GITR (DTA-1, Thermo Fisher, 48-5874-82, 1:500), anti-CD73 (eBioTY/11.8, Thermo Fisher, 46-0731-82, 1:400), anti-CD73 (TY/11.8, BioLegend, 127208, 1:400), anti-CD62L (MEL-14, BioLegend, 104441, 1:100), anti-CD62L (MEL-14, BD Biosciences, 565213, 1:600), anti-CD62L (MEL-14, BD Biosciences, 741230, 1:800), anti-CD62L (MEL-14, BioLegend, 104441, 1:400), anti-CD62L (MEL-14, BioLegend, 104438, 1:1,600), anti-CTLA4 (UC10-4B9, BioLegend, 106323, 1:200), anti-CTLA4 (UC10-4B9, Thermo Fisher, 12-1522-82, 1:400), anti-Helios (22F6, BioLegend, 137216, 1:400), anti-Helios (22F6, BioLegend, 137236, 1:400), anti-Eos (ESB7C2, Thermo Fisher, 12-5758-82, 1:400), anti-Ki-67 (SolA15, Thermo Fisher, 61-5698, 1:2,000), anti Ki67 (B56, BD Biosciences, 563757, 1:1,000), anti-Ki67 (SolA15, Fisher Scientific, 15-5698-82, 1:8,000), anti-CD25 (PC61, BD Biosciences, 564022, 1:300; Thermo Fisher, 17-0251-82, 1:400), anti-PD-1 (29F.1A12, BioLegend, 135225, 1:400), anti-CD45 (30-F11, BioLegend, 103157, 1:1,000), anti-IL-2 (JES6-5H4, BioLegend, 503818, 1:400), streptavidin (Thermo Fisher, 46-4317-82, 1:1,000), Picolyl-Azide (Jena Bioscience, CLK-1288-5), CTV (Thermo Fisher, C34557), Zombie NIR dye (BioLegend, 423105, 1:1,000), Sytox Blue (Thermo Fisher, S34857), anti-mouse CD16/32 (2.4G2, Tonbo, 70-0161-M001, 1:500).<br><br>The following antibodies were used for ELISA capturing in this study: Purified anti-mouse IL-13 (14-7133-68, Invitrogen, 88-7137-88), Purified anti-mouse IL-4 (14-7041-68A, Invitrogen, 88-7044-88), Purified anti-mouse IL-2 (eBioscience, 14-7022-68). Purified anti-mouse IgE (R35-72, BD Pharmingen, 553413), goat anti-mouse IgG1 (2794408, Southern Biotech, 1070-01), Goat Anti-Mouse IgG3 (2794567, Southern Biotech, 1100-01), Goat anti-mouse IgG2a (2794475, Southern Biotech, 1080-01), goat anti-mouse IgG2b (2794517, Southern Biotech, 1090-01), Goat Anti-Mouse IgG2c (2794464, Southern Biotech, 1079-01), goat anti-mouse IgA (2314669, Southern Biotech, 1040-01), goat anti-mouse IgM (2794197, Southern Biotech, 1020-01).<br><br>The following antibodies were used for ELISA detection in this study: biotin anti-mouse IL-13 (13-7135-68A, Invitrogen, 88-7137-88), anti-mouse IL-4 (13-7042-68C, Invitrogen, 88-7044-88), anti-mouse IL-2 (eBioscience, 33-7021-68), Goat Anti-Mouse Ig (2728714, Southern Biotech, 1010-05), biotin rat anti-mouse IgE (R35-118, BD Pharmingen, 553419). |

The following reagents were used to construct standard curves for ELISA in this study: mouse IL-4 lyophilized standard (39-8041-60, Invitrogen, 88-7044-88), mouse IL-13 lyophilized standard (39-7137/2EB-60, Invitrogen, 88-7137-88), mouse IL-2 (Thermo Fisher 212-12-5UG), purified Mouse IgG1, kappa, Isotype Control (15H6, Southern Biotech, 0102-01), purified mouse IgG2a, kappa, Isotype Control (UPC-10, Sigma, M5409), IgG2b Isotype Control from murine myeloma (MOPC-141, Sigma, M5534), Mouse IgG2c (6.3, AB_2794064, Southern Biotech, 0122-01), Purified Mouse IgG3, kappa, Isotype Control (A112-3, BD Pharmingen, 553486), Purified Mouse IgA, kappa, Isotype Control (M18-254, BD Pharmingen, 553476), IgM Isotype Control from murine myeloma (MOPC 104E, Sigma, M5909), Purified Mouse IgE, kappa, Isotype Control (C38-2, BD Pharmingen, 557079).

All antibodies are routinely QC'ed by the manufacturer for their specificity. The optimal dilution factor is determined in house by titration.

| | |
|---|---|
| Validation | All above antibodies are well validated commercial clones or preps routinely QC'ed by the manufacturer. |

## Eukaryotic cell lines

Policy information about cell lines and Sex and Gender in Research

| | |
|---|---|
| Cell line source(s) | 293T cells were originally purchased from ATCC (CRL-3216). B16-OVA cells were gifted by James Allison. |
| Authentication | Cells were not authenticated. However, the 293 T cells maintained their ability to produce high titer retrovirus throughout the study, and the B16-OVA cells maintained their ability to form a black tumor subcutaneously. |
| Mycoplasma contamination | Cells were routinely tested for mycoplasma contamination and remained negative throughout the study. |
| Commonly misidentified lines (See ICLAC register) | NA |

## Palaeontology and Archaeology

| | |
|---|---|
| Specimen provenance | *Provide provenance information for specimens and describe permits that were obtained for the work (including the name of the issuing authority, the date of issue, and any identifying information). Permits should encompass collection and, where applicable, export.* |
| Specimen deposition | *Indicate where the specimens have been deposited to permit free access by other researchers.* |
| Dating methods | *If new dates are provided, describe how they were obtained (e.g. collection, storage, sample pretreatment and measurement), where they were obtained (i.e. lab name), the calibration program and the protocol for quality assurance OR state that no new dates are provided.* |

☐ Tick this box to confirm that the raw and calibrated dates are available in the paper or in Supplementary Information.

| | |
|---|---|
| Ethics oversight | *Identify the organization(s) that approved or provided guidance on the study protocol, OR state that no ethical approval or guidance was required and explain why not.* |

Note that full information on the approval of the study protocol must also be provided in the manuscript.

## Animals and other research organisms

Policy information about studies involving animals; ARRIVE guidelines recommended for reporting animal research, and Sex and Gender in Research

| | |
|---|---|
| Laboratory animals | Foxp3-AID R26-WT, Foxp3-AID R26-TIR1, Foxp3-fl, Cd4-creERT2, Foxp3-DTR, TCRbd double knockout, C57BL/6, and Foxp3 knockout mice were used at 1 day or 6-10 weeks of age as described in the manuscript. All experiments were performed using sex-matched littermate controls whenever possible. |
| Wild animals | *Provide details on animals observed in or captured in the field; report species and age where possible. Describe how animals were caught and transported and what happened to captive animals after the study (if killed, explain why and describe method; if released, say where and when) OR state that the study did not involve wild animals.* |
| Reporting on sex | Both male and female mice were used in this study. Because Foxp3 is an X chromosome linked gene, Foxp3-AID/WT heterozygous female mice were used to eliminate cell-extrinsic effects upon Foxp3 degradation (Figure 2, Figure 3, Figure 4a-d, Figure 5a-f). Foxp3AID/y hemizygous males were used in other experiments to assess the immunological consequence of Foxp3 degradation in vivo (Figure 1, Figure 5g-o, Figure 6, Figure 7). |
| Field-collected samples | NA |
| Ethics oversight | All animal experiments in this study were approved by the Sloan Kettering Institute Institutional Animal Care and Use Committee under protocol 08-10-023 or Yale University Institutional Animal Care and Use Committee under protocol 2023-20503. |

Note that full information on the approval of the study protocol must also be provided in the manuscript.

# Clinical data

Policy information about clinical studies

All manuscripts should comply with the ICMJE guidelines for publication of clinical research and a completed CONSORT checklist must be included with all submissions.

| | |
|---|---|
| Clinical trial registration | *Provide the trial registration number from ClinicalTrials.gov or an equivalent agency.* |
| Study protocol | *Note where the full trial protocol can be accessed OR if not available, explain why.* |
| Data collection | *Describe the settings and locales of data collection, noting the time periods of recruitment and data collection.* |
| Outcomes | *Describe how you pre-defined primary and secondary outcome measures and how you assessed these measures.* |

# Dual use research of concern

Policy information about dual use research of concern

## Hazards

Could the accidental, deliberate or reckless misuse of agents or technologies generated in the work, or the application of information presented in the manuscript, pose a threat to:

No | Yes

☒ ☐ Public health

☒ ☐ National security

☒ ☐ Crops and/or livestock

☒ ☐ Ecosystems

☒ ☐ Any other significant area

## Experiments of concern

Does the work involve any of these experiments of concern:

No | Yes

☒ ☐ Demonstrate how to render a vaccine ineffective

☒ ☐ Confer resistance to therapeutically useful antibiotics or antiviral agents

☒ ☐ Enhance the virulence of a pathogen or render a nonpathogen virulent

☒ ☐ Increase transmissibility of a pathogen

☒ ☐ Alter the host range of a pathogen

☒ ☐ Enable evasion of diagnostic/detection modalities

☒ ☐ Enable the weaponization of a biological agent or toxin

☒ ☐ Any other potentially harmful combination of experiments and agents

# Plants

| | |
|---|---|
| Seed stocks | *Report on the source of all seed stocks or other plant material used. If applicable, state the seed stock centre and catalogue number. If plant specimens were collected from the field, describe the collection location, date and sampling procedures.* |
| Novel plant genotypes | *Describe the methods by which all novel plant genotypes were produced. This includes those generated by transgenic approaches, gene editing, chemical/radiation-based mutagenesis and hybridization. For transgenic lines, describe the transformation method, the number of independent lines analyzed and the generation upon which experiments were performed. For gene-edited lines, describe the editor used, the endogenous sequence targeted for editing, the targeting guide RNA sequence (if applicable) and how the editor was applied.* |
| Authentication | *Describe any authentication procedures for each seed stock used or novel genotype generated. Describe any experiments used to assess the effect of a mutation and, where applicable, how potential secondary effects (e.g. second site T-DNA insertions, mosiacism, off-target gene editing) were examined.* |

# ChIP-seq

## Data deposition

☐ Confirm that both raw and final processed data have been deposited in a public database such as GEO.

☐ Confirm that you have deposited or provided access to graph files (e.g. BED files) for the called peaks.

| | |
|---|---|
| Data access links<br>*May remain private before publication.* | *For "Initial submission" or "Revised version" documents, provide reviewer access links.  For your "Final submission" document, provide a link to the deposited data.* |
| Files in database submission | *Provide a list of all files available in the database submission.* |
| Genome browser session<br>(e.g. UCSC) | *Provide a link to an anonymized genome browser session for "Initial submission" and "Revised version" documents only, to enable peer review.  Write "no longer applicable" for "Final submission" documents.* |

## Methodology

| | |
|---|---|
| Replicates | *Describe the experimental replicates, specifying number, type and replicate agreement.* |
| Sequencing depth | *Describe the sequencing depth for each experiment, providing the total number of reads, uniquely mapped reads, length of reads and whether they were paired- or single-end.* |
| Antibodies | *Describe the antibodies used for the ChIP-seq experiments; as applicable, provide supplier name, catalog number, clone name, and lot number.* |
| Peak calling parameters | *Specify the command line program and parameters used for read mapping and peak calling, including the ChIP, control and index files used.* |
| Data quality | *Describe the methods used to ensure data quality in full detail, including how many peaks are at FDR 5% and above 5-fold enrichment.* |
| Software | *Describe the software used to collect and analyze the ChIP-seq data. For custom code that has been deposited into a community repository, provide accession details.* |

# Flow Cytometry

## Plots

Confirm that:

☐ The axis labels state the marker and fluorochrome used (e.g. CD4-FITC).

☒ The axis scales are clearly visible. Include numbers along axes only for bottom left plot of group (a 'group' is an analysis of identical markers).

☒ All plots are contour plots with outliers or pseudocolor plots.

☒ A numerical value for number of cells or percentage (with statistics) is provided.

## Methodology

| | |
|---|---|
| Sample preparation | Animals were euthanized and perfused with 20 ml PBS. Cells were isolated form the lymphoid organs by meshing with the end of a syringe plunger through a 100 mm cell strainer (Corning, 07-201-432). Lungs and tumors were digested in RPMI 1640 with 2% FBS,10 mM HEPES buffer, 100 U/mL penicillin-streptomycin, 2 mM L-glutamate, 0.2 U/mL collagenase A (Sigma, 11088793001) and 1U/mL DNase I (Sigma-Aldrich, 10104159001) for 45 min at 37°C with vigorous shaking at 250 r.p.m. 6.35mm ceramic beads (MP Biomedicals, 116540034) were included to help with tissue dissociation. The digested lungs and tumors were filtered through 70 mm separation filters (Miltenyi Biotec, 130-095-823), washed and centrifuged in PBS-adjusted 40% Percoll (Sigma Aldrich, 17-0891-01) to enrich for lymphocytes. Erythrocytes from spleen, lung and tumor were lysed by using of ACK lysis buffer (150 mM NH4Cl (Sigma-Aldrich, A9434), 10 mM KHCO3 (Sigma-Aldrich, P7682) and 0.1 mM Na2EDTA at pH 7.4).<br><br>For flow cytometry staining cells were stained with Zombie NIR dye in PBS for 10 min at 4°C to identify the dead cells followed by staining with anti-mouse CD16/32 in Staining Buffer (PBS with 0.2% BSA, 10 mM HEPES buffer and 2 mM EDTA) for 10 min at 4°C to block the Fc receptors. Next, cells were stained with fluorescently conjugated antibodies detecting cell surface antigens for 30 minutes at 4°C. To access the intracellular antigens, cells were fixed and permeabilized with eBioscience transcription factor staining buffer set (00-5523-00) according to the manufacturer's instructions.<br><br>To measure cytokine production following ex-vivo stimulation, single cell suspension was incubated with 5% CO2 at 37°C for 4 hours in the cell culture media (RPMI 1640 media supplied with 10% FBS, 100 U/mL penicillin–streptomycin, 2mM L-Glutamine, 10 mM HEPES buffer, 50 µM β-mercaptoethanol) supplied with 50ng/ml phorbol-12-myristate-13-acetate (Sigma-Aldrich, P8139), 500ng/ml ionomycin (Sigma-Aldrich, I0634), 2µM monensin (Sigma-Aldrich, M5273), 1µg/ml brefeldin A (Sigma-Aldrich, B6542). Cells were stained for flow cytometry as described above except for the fixation/permeabilization step and cytokine staining in which case BD Cytofix/Cytoperm Kit (BD Biosciences, 554715) was used according to the manufacturer's instructions. |

| | |
|---|---|
| Instrument | Flow cytometry data was collected on a 5-laser Aurora cytometer (Cytek) |
| Software | Flow cytometry data was recorded using SpectroFlo software v3.1.2 and analyzed using FlowJo v10.10.0. |
| Cell population abundance | Post sort purity was routinely examined by flow cytometry. scRNA-seq and scATAC-seq, as well as functional assay samples were sorted to a purity of at least 95%. Bulk RNA-seq samples were double sorted to a purity of at least 99%. |
| Gating strategy | Gating strategy of Foxp3-AID Treg cells:<br>1. Lymphocyte gating based on FSC-A and SSC-A.<br>2. Singlet gating based on FSC-W and FSC-H, followed by SSC-W and SSC-H.<br>3. Gate on CD4 T cells: CD4+TCRb+.<br>4. Gate on Treg cells: CD4+TCRb+ZsGreen+ from R26-WT mice; CD4+TCRb+ZsGreen+mCherry+ from R26-TIR1(F74G) mice.<br>5. In some experiments, resting (CD62hiCD44lo) and activated (CD62LloCD44hi) Treg cells were sorted separately.<br><br>Gating strategy for naive CD4 T cells from Foxp3-DTR mice:<br>1. Lymphocyte gating based on FSC-A and SSC-A.<br>2. Singlet gating based on FSC-W and FSC-H, followed by SSC-W and SSC-H.<br>3. Gate on CD4 T cells: CD4+TCRb+.<br>4. Gate on naive CD4 T cells: CD4+TCRb+GFP-CD62hiCD44lo.<br><br>Gating strategy for Treg wannabe cells with and without Foxp3-restoration.<br>1. Lymphocyte gating based on FSC-A and SSC-A.<br>2. Singlet gating based on FSC-W and FSC-H, followed by SSC-W and SSC-H.<br>3. Gate on CD4 T cells: CD4+CD3+TCRb+.<br>4. Treg wannabe cells from Foxp3-LSL Cd4-WT mice were gate as Thy1.1+GFP-; Foxp3 restored Treg wannabe cells from Foxp3-LSL Cd4-creERT2 mice were gated as CD4+TCRb+Thy1.1+/-GFP+.<br>5. Resting (CD62hiCD44lo) and activated (CD62LloCD44hi) Treg wannabe cells (with or without Foxp3 restoration) were sorted separately. |

☒ Tick this box to confirm that a figure exemplifying the gating strategy is provided in the Supplementary Information.

## Magnetic resonance imaging

### Experimental design

| | |
|---|---|
| Design type | *Indicate task or resting state; event-related or block design.* |
| Design specifications | *Specify the number of blocks, trials or experimental units per session and/or subject, and specify the length of each trial or block (if trials are blocked) and interval between trials.* |
| Behavioral performance measures | *State number and/or type of variables recorded (e.g. correct button press, response time) and what statistics were used to establish that the subjects were performing the task as expected (e.g. mean, range, and/or standard deviation across subjects).* |

### Acquisition

| | |
|---|---|
| Imaging type(s) | *Specify: functional, structural, diffusion, perfusion.* |
| Field strength | *Specify in Tesla* |
| Sequence & imaging parameters | *Specify the pulse sequence type (gradient echo, spin echo, etc.), imaging type (EPI, spiral, etc.), field of view, matrix size, slice thickness, orientation and TE/TR/flip angle.* |
| Area of acquisition | *State whether a whole brain scan was used OR define the area of acquisition, describing how the region was determined.* |

Diffusion MRI ☐ Used ☐ Not used

### Preprocessing

| | |
|---|---|
| Preprocessing software | *Provide detail on software version and revision number and on specific parameters (model/functions, brain extraction, segmentation, smoothing kernel size, etc.).* |
| Normalization | *If data were normalized/standardized, describe the approach(es): specify linear or non-linear and define image types used for transformation OR indicate that data were not normalized and explain rationale for lack of normalization.* |
| Normalization template | *Describe the template used for normalization/transformation, specifying subject space or group standardized space (e.g. original Talairach, MNI305, ICBM152) OR indicate that the data were not normalized.* |
| Noise and artifact removal | *Describe your procedure(s) for artifact and structured noise removal, specifying motion parameters, tissue signals and physiological signals (heart rate, respiration).* |

| Volume censoring | *Define your software and/or method and criteria for volume censoring, and state the extent of such censoring.* |

## Statistical modeling & inference

| Model type and settings | *Specify type (mass univariate, multivariate, RSA, predictive, etc.) and describe essential details of the model at the first and second levels (e.g. fixed, random or mixed effects; drift or auto-correlation).* |

| Effect(s) tested | *Define precise effect in terms of the task or stimulus conditions instead of psychological concepts and indicate whether ANOVA or factorial designs were used.* |

Specify type of analysis: ☐ Whole brain ☐ ROI-based ☐ Both

| Statistic type for inference | *Specify voxel-wise or cluster-wise and report all relevant parameters for cluster-wise methods.* |

(See Eklund et al. 2016)

| Correction | *Describe the type of correction and how it is obtained for multiple comparisons (e.g. FWE, FDR, permutation or Monte Carlo).* |

## Models & analysis

n/a | Involved in the study

☒ ☐ Functional and/or effective connectivity

☒ ☐ Graph analysis

☒ ☐ Multivariate modeling or predictive analysis

