## [Peer Review File · Nature Immunology]

Temporal and Context-Dependent Requirements for the Transcription Factor Foxp3 Expression in Regulatory T Cells

Corresponding Author: Professor Alexander Rudensky

Version 0:

Reviewer comments:

Reviewer #2

(Remarks to the Author)

In the manuscript 'Temporal and Context-Dependent Requirements for the Transcription Factor Foxp3 Expression in Regulatory T Cells', Hu and colleagues use an elegant mouse model to decipher Treg cell dependency on Foxp3 protein expression. They show that under steady state, the protein Foxp3 is largely unnecessary to maintain the Treg phenotype, while during Treg cell development or during phases of rapid Treg proliferation, a loss of Foxp3 protein leads to decreased Treg cell functionality.

This is an interesting study. While the importance of Foxp3 for Treg cells during differentiation is well documented, the relevance of Foxp3 for Treg cells under steady state was not known. This study proposes that Foxp3 is largely unnecessary to maintain the Treg phenotype under steady state. However, in their manuscript the authors only analyze how lymphoid Treg cells react to a degradation of the Foxp3 protein under steady state. What about non-lymphoid sites? As the colon is a site where the immune system is constantly low-key challenged by the local microbiome (under steady state), it would be important to determine whether the observations the authors make (Foxp3 is largely unnecessary to maintain the Treg phenotype under steady state) also hold up under these conditions. Furthermore, is there a difference in Foxp3 dependency between pTreg and tTreg cells in the gut?

In addition to proliferation, do other challenges, such as pro-inflammatory cytokine signaling, destabilize the Treg identity after Foxp3 degradation?

Most experiments regarding "Foxp3 is largely unnecessary to maintain the Treg phenotype" are done in short term experiments after Foxp3 degradation (Figure 2 and Figure 3: day 3 and day 7). It would be important to observe the phenotype of Treg cells after longer loss of Foxp3, e.g., 2 – 3 weeks.

Abstract: The authors state that 'tumor Treg cells were uniquely sensitive to Foxp3 degradation'. However, according to the hypothesis proposed in their manuscript, all proliferating Treg cells (e.g., neonatal Treg cells and, presumably, also Treg cells in other (auto)inflammatory conditions) are sensitive to Foxp3 degradation. Perhaps rephrase to better represent the author's hypothesis.

Minor

1) Figure 4a/b: unclear which y-axis applies to the bars and which one applies to the dots. Please clarify in the figure legend. Also, the two different shades of red/blue used are at times difficult to tell apart, especially in subpanel c and g.

Reviewer #3

(Remarks to the Author)

In this manuscript by Hu et al., the authors examine the role of the transcription factor Foxp3 in the function of regulatory T cells (Tregs) by generating an inducible Foxp3-degradation mouse. The authors show that the new system rapidly degrades Foxp3 and degradation of Foxp3 in neonates phenocopies the autoimmune diseases previously reported in published mouse models of Treg depletion (such as Foxp3-DTR). An interesting and surprising finding is that the degradation of Foxp3 in mature Tregs led to few changes in gene expression and did not affect suppressive capacity. However, Tregs that are

activated and proliferating were more dependent on foxp3 which was seen in in vivo in the context of tumors where the degradation of Foxp3 led to loss of suppressive capacity and increased anti-tumor immune responses and reduced tumour growth.

This is an excellent manuscript that has established a new model system to dissect the direct role of Foxp3 in Treg function. The data is clearly presented and appropriate controls included. Importantly the authors perform an extensive characterisation of the new mouse model they created to ensure it functions as advertised. The authors compare a number of existing models to their new one to provide context for their conclusions.

I only have a few suggestions comments that I think will improve the paper.

1. In reference to figure 7, for completeness I think they authors should look to see if other Foxp3 expressing cells exist in the tumor microenvironment. There have been reports of other types of Foxp3-expressing cells such as macrophages and CD8+ T cells that might arise in the tumor environment also become anti-tumorigenic upon Foxp3 degradation. Do the authors see any of these populations in the ZsGreen+ pool and are they altered in their gene expression profiles? If other Foxp3 expressing cell types in the tumor are also affected this alters the interpretation and reporting of the results (but only really warrants alterations in the text if it is the case).

2. I was confused by the CTV nomenclature used in Fig6c-e. In the text it was stated: "The highly divided (CTVlow) cells..." however in figure 6c the gate labelled "lo" is placed on cells that have lowly divided. Could the authors please check and clarify this?

3. In Figure 5, degradation of Foxp3 in neonates shows similar dramatic autoimmune phenotype to the Foxp3 deletion mice confirming that the system works well. Here would be a good opportunity to provide more detail about the specific genes that are altered in this system. For example, can the true "direct" targets of Foxp3 in developing Tregs be revealed using the degon system as compared to other genetic ablation systems?

Reviewer #4

(Remarks to the Author)

The manuscript by Hu et al. introduces a novel mouse model to deplete FOXP3 protein expression in FOXP3+Tregs. They report a limited effect of induced FOXP3 protein degradation on the gene expression of mature Tregs, which appear to be enriched for direct FOXP3 target genes. This is insufficient to induce multiorgan autoimmune inflammation. Conversely, the gene expression of thymic, proliferating or tumor-infiltrating Tregs is more strongly affected by induced FOXP3 degradation. Overall, this is a highly interesting, relevant, innovative and elegant study, but the following points need to be clarified:

Major points:

1. Impact of prolonged FOXP3 degradation: On page 4 the authors wrote: "Notably, immune cell activation and tissue inflammation following Foxp3 degradation appeared to reach a new setpoint, as no further increases were observed between two and four weeks of 5-ph-IAA treatment (Figure 1f-k, Extended Data Figure 3b-d)." However, in the cited Figures there are significant differences for neutrophil and monocyte numbers, as well as IgE levels exclusively at day 28. Later time points were not analysed. It is thus still possible that FOXP3 degradation in mature Tregs may also induce autoimmunity, but with delayed kinetics. Since IL2Ralpha/beta are among the direct FOXP3 target genes and are critical for Treg homeostasis it would be surprising if Treg function would not be affected more strongly in the long run.
2. Tregs in non-lymphoid tissues: sc-RNA seq was performed with Tregs from secondary lymphoid organs, but not from non-lymphoid tissues. Some targets were however validated by flow cytometry in the lung and unveiled sometimes dramatic differences (Ox40, TCF1). The effect of FOXP3 degradation in Tregs in non-lymphoid tissues is highly relevant and should be analysed more in detail, ideally by sc-RNA seq.
3. Proliferation versus cellular "context": The key point of this study is the relatively mild phenotype of FOXP3 degradation in mature Tregs. In Figure 5 the authors provide evidence for a key role for Treg proliferation in this surprising phenomenon. A key question that remains unanswered is if the observed "vulnerability" of Tregs to FOXP3 degradation in the thymus, in neonates and in tumors is a mere consequence of their high proliferation rates, or if these cellular contexts may require Tregs with intact FOXP3 protein expression also independently of proliferation.
4. A limitation of this -otherwise very complete- study is the lack of human data, in particular in Figure 7 that has translational potential.

Minor points:

General:

This is a very complex story and any effort should be made to simplify and explain every step to the readers.

Figure 1C. A convincing density plot overlay is shown, but the FOXP3 expression levels in several mice should be shown

Figure 1f-k: I would suggest using different colours for d14 and d28

Figure 2d: The authors should report the differences in cell abundance across samples and conditions. Otherwise the

relative representation of Foxp3AIDR26WT and Foxp3AIDR26TIR1 Treg cells cannot be accurately inferred from the bar plot.

Figure 2i: The number of down-regulated genes in resting Tregs is only slightly reduced, all others are dramatically reduced. This was not clearly stated in the text ("only a very small set of genes", this seems to be an overstatement). Perhaps these genes could be simply shown in a supplementary Table?

Figure 3d: This is a rather complex panel. Interestingly, IL-17A is upregulated in the WT control. A comment on this point in the Discussion would be appropriate.

Figures 4a/b: What do the individual data points represent? Are they outliers?

Figure 4C: This panel reports data from CHIP experiments as far as I understood. This was not mentioned anywhere, please clarify.

Figure 5d-f, n-o are very complex. The Foxp3 expression levels in Figure 5f are scaled. Unscaled values should be used for a more accurate representation.

Figure 7k/n: both orange and red points are labelled R26Tr1, which is the WT control? In the dataset these conditions were apparently mislabelled.

Extended Figure 4: it is not clear to me why this control excludes a role of residual FOXP3 protein expression in suppressive functions.

Decision Letter:

2nd Jun 2025

Dear Sasha,

Thank you for providing a point-by-point response to the referees' comments on your manuscript entitled, "Temporal and Context-Dependent Requirements for the Transcription Factor Foxp3 Expression in Regulatory T Cells". As noted previously, while they find your work of considerable potential interest, they have raised quite substantial concerns that must be addressed. In light of these comments, we cannot accept the current manuscript for publication, but would be very interested in considering a revised version that addresses these concerns along the lines proposed in your point-by-point rebuttal.

We invite you to submit a substantially revised manuscript, however please bear in mind that we will be reluctant to approach the referees again in the absence of major revisions.

Specifically, the revision should include new experiments to address:

- (1) perform single-cell RNA sequencing of Treg cells isolated from the lung, liver, colon, and secondary lymphoid organs
- (2) assess the impact of inflammatory cytokines on Treg Foxp3 dependency, we will transfer Foxp3-AID Treg cells into Treg-ablated Foxp3-DTR mice
- (3) transfer Foxp3-AID Treg cells into T cell-deficient recipients to assess the effect of Foxp3 degradation on their phenotype and function

Please include the additional textual clarifications as indicated in your response letter.

When you revise your manuscript, please take into account all reviewer and editor comments, please highlight all changes in the manuscript text file in Microsoft Word format.

* If you have not done so already please begin to revise your manuscript so that it conforms to our Article format instructions at <http://www.nature.com/ni/authors/index.html>. Refer also to any guidelines provided in this letter.

The Reporting Summary can be found here:
<https://www.nature.com/documents/nr-reporting-summary.pdf>

When submitting the revised version of your manuscript, please pay close attention to our [href="https://www.nature.com/nature-portfolio/editorial-policies/image-integrity">Digital Image Integrity Guidelines. and to the following points below:](https://www.nature.com/nature-portfolio/editorial-policies/image-integrity)

Extended Data figures and tables are online-only (appearing in the online PDF and full-text HTML version of the paper), peer-reviewed display items that provide essential background to the Article but are not included in the printed version of the paper due to space constraints or being of interest only to a few specialists. A maximum of ten Extended Data display items (figures and tables) is typically permitted. When re-submitting your manuscript, please ensure that any supplementary figures and tables that are more critical to the manuscript's conclusions are converted to Extended data to increase these data's visibility.

Link Redacted

If you wish to submit a suitably revised manuscript we would hope to receive it within 6 months. If you cannot send it within this time, please let us know. We will be happy to consider your revision so long as nothing similar has been accepted for publication at Nature Immunology or published elsewhere.

Nature Immunology is committed to improving transparency in authorship. As part of our efforts in this direction, we are now requesting that all authors identified as 'corresponding author' on published papers create and link their Open Researcher and Contributor Identifier (ORCID) with their account on the Manuscript Tracking System (MTS), prior to acceptance. ORCID helps the scientific community achieve unambiguous attribution of all scholarly contributions. You can create and link your ORCID from the home page of the MTS by clicking on 'Modify my Springer Nature account'. For more information please visit <http://www.springernature.com/orcid>.

Thank you for the opportunity to review your work.

Kind regards,

Laurie

Laurie A. Dempsey, Ph.D.
Senior Editor
Nature Immunology
l.dempsey@us.nature.com
ORCID: 0000-0002-3304-796X

Reviewers' Comments:

Reviewer #2 (Remarks to the Author):

In the manuscript 'Temporal and Context-Dependent Requirements for the Transcription Factor Foxp3 Expression in Regulatory T Cells', Hu and colleagues use an elegant mouse model to decipher Treg cell dependency on Foxp3 protein

expression. They show that under steady state, the protein Foxp3 is largely unnecessary to maintain the Treg phenotype, while during Treg cell development or during phases of rapid Treg proliferation, a loss of Foxp3 protein leads to decreased Treg cell functionality.

This is an interesting study. While the importance of Foxp3 for Treg cells during differentiation is well documented, the relevance of Foxp3 for Treg cells under steady state was not known. This study proposes that Foxp3 is largely unnecessary to maintain the Treg phenotype under steady state. However, in their manuscript the authors only analyze how lymphoid Treg cells react to a degradation of the Foxp3 protein under steady state. What about non-lymphoid sites? As the colon is a site where the immune system is constantly low-key challenged by the local microbiome (under steady state), it would be important to determine whether the observations the authors make (Foxp3 is largely unnecessary to maintain the Treg phenotype under steady state) also hold up under these conditions. Furthermore, is there a difference in Foxp3 dependency between pTreg and tTreg cells in the gut?

In addition to proliferation, do other challenges, such as pro-inflammatory cytokine signaling, destabilize the Treg identity after Foxp3 degradation?

Most experiments regarding "Foxp3 is largely unnecessary to maintain the Treg phenotype" are done in short term experiments after Foxp3 degradation (Figure 2 and Figure 3: day 3 and day 7). It would be important to observe the phenotype of Treg cells after longer loss of Foxp3, e.g., 2 – 3 weeks.

Abstract: The authors state that 'tumor Treg cells were uniquely sensitive to Foxp3 degradation'. However, according to the hypothesis proposed in their manuscript, all proliferating Treg cells (e.g., neonatal Treg cells and, presumably, also Treg cells in other (auto)inflammatory conditions) are sensitive to Foxp3 degradation. Perhaps rephrase to better represent the author's hypothesis.

Minor

1) Figure 4a/b: unclear which y-axis applies to the bars and which one applies to the dots. Please clarify in the figure legend. Also, the two different shades of red/blue used are at times difficult to tell apart, especially in subpanel c and g.

Reviewer #3 (Remarks to the Author):

In this manuscript by Hu et al., the authors examine the role of the transcription factor Foxp3 in the function of regulatory T cells (Tregs) by generating an inducible Foxp3-degradation mouse. The authors show that the new system rapidly degrades Foxp3 and degradation of Foxp3 in neonates phenocopies the autoimmune diseases previously reported in published mouse models of Treg depletion (such as Foxp3-DTR). An interesting and surprising finding is that the degradation of Foxp3 in mature Tregs led to few changes in gene expression and did not affect suppressive capacity. However, Tregs that are activated and proliferating were more dependent on foxp3 which was seen in in vivo in the context of tumors where the degradation of Foxp3 led to loss of suppressive capacity and increased anti-tumor immune responses and reduced tumour growth.

This is an excellent manuscript that has established a new model system to dissect the direct role of Foxp3 in Treg function. The data is clearly presented and appropriate controls included. Importantly the authors perform an extensive characterisation of the new mouse model they created to ensure it functions as advertised. The authors compare a number of existing models to their new one to provide context for their conclusions.

I only have a few suggestions comments that I think will improve the paper.

1. In reference to figure 7, for completeness I think they authors should look to see if other Foxp3 expressing cells exist in the tumor microenvironment. There have been reports of other types of Foxp3-expressing cells such as macrophages and CD8+ T cells that might arise in the tumor environment also become anti-tumorigenic upon Foxp3 degradation. Do the authors see any of these populations in the ZsGreen+ pool and are they altered in their gene expression profiles? If other Foxp3 expressing cell types in the tumor are also affected this alters the interpretation and reporting of the results (but only really warrants alterations in the text if it is the case).

2. I was confused by the CTV nomenclature used in Fig6c-e. In the text it was stated: "The highly divided (CTVlow) cells..." however in figure 6c the gate labelled "lo" is placed on cells that have lowly divided. Could the authors please check and clarify this?

3. In Figure 5, degradation of Foxp3 in neonates shows similar dramatic autoimmune phenotype to the Foxp3 deletion mice confirming that the system works well. Here would be a good opportunity to provide more detail about the specific genes that are altered in this system. For example, can the true "direct" targets of Foxp3 in developing Tregs be revealed using the degon system as compared to other genetic ablation systems?

Reviewer #4 (Remarks to the Author):

The manuscript by Hu et al. introduces a novel mouse model to deplete FOXP3 protein expression in FOXP3+Tregs. They report a limited effect of induced FOXP3 protein degradation on the gene expression of mature Tregs, which appear to be enriched for direct FOXP3 target genes. This is insufficient to induce multiorgan autoimmune inflammation. Conversely, the gene expression of thymic, proliferating or tumor-infiltrating Tregs is more strongly affected by induced FOXP3 degradation. Overall, this is a highly interesting, relevant, innovative and elegant study, but the following points need to be clarified:

Major points:

1. Impact of prolonged FOXP3 degradation: On page 4 the authors wrote: "Notably, immune cell activation and tissue inflammation following Foxp3 degradation appeared to reach a new setpoint, as no further increases were observed between two and four weeks of 5-ph-IAA treatment (Figure 1f-k, Extended Data Figure 3b-d)." However, in the cited Figures there are significant differences for neutrophil and monocyte numbers, as well as IgE levels exclusively at day 28. Later time points were not analysed. It is thus still possible that FOXP3 degradation in mature Tregs may also induce autoimmunity, but with delayed kinetics. Since IL2Ralpha/beta are among the direct FOXP3 target genes and are critical for Treg homeostasis it would be surprising if Treg function would not be affected more strongly in the long run.
2. Tregs in non-lymphoid tissues: sc-RNA seq was performed with Tregs from secondary lymphoid organs, but not from non-lymphoid tissues. Some targets were however validated by flow cytometry in the lung and unveiled sometimes dramatic differences (Ox40, TCF1). The effect of FOXP3 degradation in Tregs in non-lymphoid tissues is highly relevant and should be analysed more in detail, ideally by sc-RNA seq.
3. Proliferation versus cellular "context": The key point of this study is the relatively mild phenotype of FOXP3 degradation in mature Tregs. In Figure 5 the authors provide evidence for a key role for Treg proliferation in this surprising phenomenon. A key question that remains unanswered is if the observed "vulnerability" of Tregs to FOXP3 degradation in the thymus, in neonates and in tumors is a mere consequence of their high proliferation rates, or if these cellular contexts may require Tregs with intact FOXP3 protein expression also independently of proliferation.
4. A limitation of this -otherwise very complete- study is the lack of human data, in particular in Figure 7 that has translational potential.

Minor points:

General:

This is a very complex story and any effort should be made to simplify and explain every step to the readers.

Figure 1C. A convincing density plot overlay is shown, but the FOXP3 expression levels in several mice should be shown

Figure 1f-k: I would suggest using different colours for d14 and d28

Figure 2d: The authors should report the differences in cell abundance across samples and conditions. Otherwise the relative representation of Foxp3AIDR26WT and Foxp3AIDR26TIR1 Treg cells cannot be accurately inferred from the bar plot.

Figure 2i: The number of down-regulated genes in resting Tregs is only slightly reduced, all others are dramatically reduced. This was not clearly stated in the text ("only a very small set of genes", this seems to be an overstatement). Perhaps these genes could be simply shown in a supplementary Table?

Figure 3d: This is a rather complex panel. Interestingly, IL-17A is upregulated in the WT control. A comment on this point in the Discussion would be appropriate.

Figures 4a/b: What do the individual data points represent? Are they outliers?

Figure 4C: This panel reports data from CHIP experiments as far as I understood. This was not mentioned anywhere, please clarify.

Figure 5d-f, n-o are very complex. The Foxp3 expression levels in Figure 5f are scaled. Unscaled values should be used for a more accurate representation.

Figure 7k/n: both orange and red points are labelled R26Tr1, which is the WT control? In the dataset these conditions were apparently mislabelled.

Extended Figure 4: it is not clear to me why this control excludes a role of residual FOXP3 protein expression in suppressive functions.

Version 1:

Reviewer comments:

Reviewer #2

(Remarks to the Author)

Hu and colleagues have adequately addressed all of my concerns. However, I would like to point out that during the review process, a paper was published in Science Immunology by the group of Joris van der Veeke (PMID: 40478934) with almost identical content to the present manuscript. Both the mouse model used (Auxin degradable Foxp3 mice) and the conclusion drawn (that Foxp3 is not really important for resting Treg cells but vital for Treg cells in inflammatory environments) is very similar in both manuscripts. While the present paper only uses a murine tumor model, van der Veeke and colleagues study the loss of Foxp3 in tumor, viral infections and autoimmunity. Unfortunately, this greatly diminishes the novelty of the findings presented in this manuscript.

Reviewer #3

(Remarks to the Author)

The authors have addressed my all questions.

Reviewer #4

(Remarks to the Author)

The authors have addressed my concerns and the manuscript has improved.

Decision Letter:

Our ref: NI-A40307A

7th Aug 2025

Dear Dr. Rudensky,

Thank you for submitting your revised manuscript "Temporal and Context-Dependent Requirements for the Transcription Factor Foxp3 Expression in Regulatory T Cells" (NI-A40307A). It has now been seen by the original referees and their comments are below. The reviewers find that the paper has improved in revision, and therefore we'll be happy in principle to publish it in Nature Immunology, pending minor revisions to satisfy the referees' final requests and to comply with our editorial and formatting guidelines.

Since a very similar paper was published in Sci Immunol, we will fast-track your manuscript.

We will now perform detailed checks on your paper and will send you a checklist detailing our editorial and formatting requirements in about a week. Please do not upload the final materials and make any revisions until you receive this additional information from us.

If you had not uploaded a Word file for the current version of the manuscript, we will need one before beginning the editing process; please email that to immunology@us.nature.com at your earliest convenience.

Thank you again for your interest in Nature Immunology Please do not hesitate to contact me if you have any questions.

Sincerely,

Jamie D K Wilson, D.Phil
Chief Editor
For:

Laurie A. Dempsey, Ph.D.
Senior Editor
Nature Immunology
l.dempsey@us.nature.com
ORCID: 0000-0002-3304-796X

Reviewer #2 (Remarks to the Author):

Hu and colleagues have adequately addressed all of my concerns. However, I would like to point out that during the review process, a paper was published in Science Immunology by the group of Joris van der Veeke (PMID: 40478934) with almost identical content to the present manuscript. Both the mouse model used (Auxin degradable Foxp3 mice) and the conclusion drawn (that Foxp3 is not really important for resting Treg cells but vital for Treg cells in inflammatory environments) is very similar in both manuscripts. While the present paper only uses a murine tumor model, van der Veeke and colleagues study the loss of Foxp3 in tumor, viral infections and autoimmunity. Unfortunately, this greatly diminishes the novelty of the findings presented in this manuscript.

Reviewer #3 (Remarks to the Author):

The authors have addressed my all questions.

Reviewer #4 (Remarks to the Author):

The authors have addressed my concerns and the manuscript has improved.

Reviewer #2

(Remarks to the Author)

In the manuscript 'Temporal and Context-Dependent Requirements for the Transcription Factor Foxp3 Expression in Regulatory T Cells', Hu and colleagues use an elegant mouse model to decipher Treg cell dependency on Foxp3 protein expression. They show that under steady state, the protein Foxp3 is largely unnecessary to maintain the Treg phenotype, while during Treg cell development or during phases of rapid Treg proliferation, a loss of Foxp3 protein leads to decreased Treg cell functionality.

1. This is an interesting study. While the importance of Foxp3 for Treg cells during differentiation is well documented, the relevance of Foxp3 for Treg cells under steady state was not known. This study proposes that Foxp3 is largely unnecessary to maintain the Treg phenotype under steady state. However, in their manuscript the authors only analyze how lymphoid Treg cells react to a degradation of the Foxp3 protein under steady state. What about non-lymphoid sites? As the colon is a site where the immune system is constantly low-key challenged by the local microbiome (under steady state), it would be important to determine whether the observations the authors make (Foxp3 is largely unnecessary to maintain the Treg phenotype under steady state) also hold up under these conditions. Furthermore, is there a difference in Foxp3 dependency between pTreg and tTreg cells in the gut?

The absence of tissue pathology in the liver suggests that liver-resident Treg cells are largely compromised by Foxp3 degradation. Similarly, aside from direct Foxp3 target genes, our flow cytometric analysis revealed minimal phenotypic changes in lung-resident Treg cells following Foxp3 degradation (Fig. 3f,g). These findings indicate that Foxp3 degradation has limited impact on tissue-resident Treg cells, consistent with our observations in lymphoid-resident Treg cells.

To experimentally address reviewer's questions, we performed single-cell RNA-seq analysis of Treg cells isolated from the lung, liver, large intestine lamina propria (LILP), and secondary lymphoid organs (SLOs) of mosaic heterozygote female *Foxp3*^{AID/WT} *R26*^{TIR1(F74G)} mice harboring a mix of *Foxp3*^{AID} and *Foxp3*^{WT} expressing Treg cells after 7 days of continuous Foxp3 degradation under physiologic conditions. Analysis and discussion of this experiment and its results are now included in the revised manuscript (**Extended Data Figure 8**). In brief, we found that the effects of Foxp3 degradation are highly conserved across tissues in resting Treg cells, but less conserved in activated Treg cells (**Extended Data Figure 8e**). This is particularly prominent in the LILP, where almost all Treg cells are activated.

Given the prominent role of both pTreg and tTregs in the gut, we further subclustered the LILP Treg cells and identified 3 major clusters, which we classified as pTregs, lymphoid-tissue tTreg (LT-tTreg), and non-lymphoid tissue tTreg (NLT-tTreg) (**Extended Data Figure 8h-i**). This classification is consistent with published gut Treg transcriptional datasets (Miragaia et. al., *Immunity*, 2019). Foxp3 degradation led to gene expression changes in cells from all 3 clusters, with pTreg cells having the most DEGs (**Extended Data Figure 8j**). This can be potentially explained by ongoing generation of ROR γ ^t pTreg cells and likely enrichment of this subset in

recently generated ones. It is notable, however, that the Foxp3 repressed and activated genes shown in **Figure 3a**, which are likely enriched for Foxp3 direct targets, had the largest magnitude of change in tTreg cell populations (**Extended Data Figure 8k**). These findings are consistent with our previous work demonstrating that peripherally induced Treg (pTreg) cells in the gut maintain their fitness and exert some of their suppressive functions independently of Foxp3 and have (van der Veeken et al. *Immunity*, 2022). Together with the previous results, these data suggest that the Foxp3 gene regulatory network is differently wired between pTreg and tTreg cells.

2. In addition to proliferation, do other challenges, such as pro-inflammatory cytokine signaling, destabilize the Treg identity after Foxp3 degradation?

We agree with the Reviewer that in addition to proliferation, inflammatory cytokines may further increase Treg cell reliance on Foxp3. To assess the impact of inflammatory cytokines on Treg Foxp3 dependency, we adoptively transferred *Foxp3^{AID}R26^{TIR1(F74G)}* and *Foxp3^{AID}R26^{WT}* Treg cells at a 1:1 ratio into *Foxp3^{DTR}* mice subjected to continuous Treg ablation upon DT administration, a setting of systemic polytypic (type 1, 2 and 3) inflammation accompanied by Treg cell activation and proliferation. As a control, we performed similar *Foxp3^{AID}R26^{TIR1(F74G)}* and *Foxp3^{AID}R26^{WT}* Treg cell transfer into *Tcrb/Tcrd* double deficient mice, a setting that involves proliferation and activation with minimal inflammation within 7 days post transfer. After 7 days of continuous Foxp3 degradation, *R26^{TIR1(F74G)}* and *R26^{WT}* were isolated by FACS (sorting on the marker mCherry) and single-cell RNA-seq analysis was performed. The results of this experiment have been added to the manuscript (**new Figure 6i-o**). In brief, we found that Foxp3 degradation in Treg cells in the polytypic inflammatory environment (*Foxp3^{DTR}*) perturbed Treg transcriptional networks to a much greater extent when compared to those in a lymphopenic environment (*Tcrb/Tcrd* double KO) (**Figure 6m-o**). Furthermore, Foxp3 degradation in the inflammatory environment dramatically reduced Treg cell fitness (**new Figure 6k**). Overall, our findings suggest that Foxp3 regulatory networks in Treg cells are dramatically altered in settings of severe inflammation. The exact mechanisms through which this occurs and the precise signals that drive instability of the Treg transcriptional program in inflammatory conditions are intriguing future directions to pursue outside the scope of our study.

3. Most experiments regarding “Foxp3 is largely unnecessary to maintain the Treg phenotype” are done in short term experiments after Foxp3 degradation (Figure 2 and Figure 3: day 3 and day 7). It would be important to observe the phenotype of Treg cells after longer loss of Foxp3, e.g., 2 – 3 weeks.

We apologize for not being sufficiently clear. We did not mean to imply that Foxp3 is completely irrelevant to mature Treg cell function (see also response to Reviewer 4). Instead, we meant to highlight the slow tempo of acquisition of a stable Foxp3-dependent transcriptional and functional program by Treg cells and the prolonged stability of this program in mature Treg cells under physiologic conditions. In Figure 1, we extended Foxp3 degradation up to 28 days, where we observed only minor changes in immune activation compared to that which we observed at 14 days. These results suggest that Foxp3 is largely dispensable for maintaining Treg functionality

for at least 28 days. We expect that prolonged loss of Foxp3 would eventually result in autoimmune pathology. However, the slow kinetics of this response is surprising and stands in stark contrast to acute ablation of Treg cells, in which mice succumb to disease within 2-3 weeks.

In Figure 2 and Figure 3, we focused on shorter time points to identify direct Foxp3 gene targets, as longer Foxp3 degradation will reveal more Foxp3 indirect targets.

4. Abstract: The authors state that ‘tumor Treg cells were uniquely sensitive to Foxp3 degradation’. However, according to the hypothesis proposed in their manuscript, all proliferating Treg cells (e.g., neonatal Treg cells and, presumably, also Treg cells in other (auto)inflammatory conditions) are sensitive to Foxp3 degradation. Perhaps rephrase to better represent the author’s hypothesis.

We have revised the abstract to more accurately reflect the conclusions from our study. Specifically, we aim to emphasize that, post initial development in the thymus, tumor-infiltrating Treg cells exhibit a heightened sensitivity to Foxp3 degradation when compared to non-tumor-infiltrating Treg cells in the same mouse. This is likely due to some combination of the highly proliferative nature of intratumoral Treg cells and the inflammatory tumor microenvironment, but the precise mechanism driving this sensitivity remains to be elucidated. Nevertheless, we wanted to emphasize these tumor findings as they represent a disease state in which Foxp3 degradation offers a unique therapeutic approach to manipulate Treg cells in a manner that promotes anti-tumor responses while limiting systemic toxicity. Of note, when Foxp3 was degraded in the context of experimental autoimmune encephalomyelitis (EAE), we observed no change in disease severity (data not shown). Given that Treg cell ablation is known to worsen EAE, these findings suggest that Foxp3 degradation does not substantially impair Treg cell function at least in this autoimmune setting.

5. Minor

1) Figure 4a/b: unclear which y-axis applies to the bars and which one applies to the dots. Please clarify in the figure legend. Also, the two different shades of red/blue used are at times difficult to tell apart, especially in subpanel c and g.

We have removed the dots from the figures to avoid any confusion and because the information they contain (number of genes in each p-value bin) is indicated below each plot. We have also altered the colors to make them easier to distinguish.

Reviewer #3

(Remarks to the Author)

In this manuscript by Hu et al., the authors examine the role of the transcription factor Foxp3 in the function of regulatory T cells (Tregs) by generating an inducible Foxp3-degradation mouse. The authors show that the new system rapidly degrades Foxp3 and degradation of Foxp3 in neonates phenocopies the autoimmune diseases previously reported in published mouse models of Treg depletion (such as Foxp3-DTR). An interesting and surprising finding is that the

degradation of Foxp3 in mature Tregs led to few changes in gene expression and did not affect suppressive capacity. However, Tregs that are activated and proliferating were more dependent on foxp3 which was seen in vivo in the context of tumors where the degradation of Foxp3 led to loss of suppressive capacity and increased anti-tumor immune responses and reduced tumour growth.

This is an excellent manuscript that has established a new model system to dissect the direct role of Foxp3 in Treg function. The data is clearly presented and appropriate controls included. Importantly the authors perform an extensive characterisation of the new mouse model they created to ensure it functions as advertised. The authors compare a number of existing models to their new one to provide context for their conclusions.

I only have a few suggestions comments that I think will improve the paper.

1. In reference to figure 7, for completeness I think they authors should look to see if other Foxp3 expressing cells exist in the tumor microenvironment. There have been reports of other types of Foxp3-expressing cells such as macrophages and CD8+ T cells that might arise in the tumor environment also become anti-tumorigenic upon Foxp3 degradation. Do the authors see any of these populations in the ZsGreen+ pool and are they altered in their gene expression profiles? If other Foxp3 expressing cell types in the tumor are also affected this alters the interpretation and reporting of the results (but only really warrants alterations in the text if it is the case).

We observed a very minor population of Foxp3⁺ CD8⁺ T cells in the tumor. These cells on average constituted less than 0.15% of CD45⁺ cells in the tumor, whereas Foxp3⁺ CD4⁺ T cells comprised approximately 8-9% of CD45⁺ cells. Given the extremely low abundance of Foxp3⁺ CD8⁺ T cells, we do not expect them to play a significant role in anti-tumor immune responses. Additionally, we did not detect any Foxp3⁺ macrophages in the tumor. The discrepancies between our findings and previous reports describing high levels of non-Treg Foxp3-expressing cells may be due to species differences (mouse vs. human), distinct mouse genetic backgrounds, or variations in cancer type. Regardless, we believe our interpretation remains valid under our specific experimental conditions. For the Reviewer's reference, we have included a representative flow plot of gating for ZsGreen⁺ CD8⁺ T cells and their frequencies compared to CD4⁺ T cells within the tumor in 9 Foxp3^{AID} mice.

2. I was confused by the CTV nomenclature used in Fig6c-e. In the text it was stated: "The highly divided (CTVlow) cells..." however in figure 6c the gate labelled "lo" is placed on cells that have lowly divided. Could the authors please check and clarify this?

We apologize for the confusion caused by the labeling. In the revised manuscript, we have ensured consistent labeling of this panel between the figure and the text.

3. In Figure 5, degradation of Foxp3 in neonates shows similar dramatic autoimmune phenotype to the Foxp3 deletion mice confirming that the system works well. Here would be a good opportunity to provide more detail about the specific genes that are altered in this system. For example, can the true "direct" targets of Foxp3 in developing Tregs be revealed using the degron system as compared to other genetic ablation systems?

Developing Treg cells rely heavily on Foxp3 to establish the Treg-specific transcriptional network. Therefore, Foxp3 degradation in recently generated Treg cells, which represent the bulk of Treg cells in the neonates, reveals both direct and indirect Foxp3 targets, similar to what is observed in genetic Foxp3 deficiency, as shown in Figure 5n. In the revised manuscript we have included a supplemental table (**Supplemental Table 1**) that lists all DEGs from analyses in the manuscript. As expected, the expression levels of many of the direct Foxp3 target genes identified in adult mice (Figure 3) are accordingly altered upon Foxp3 degradation. A number of these genes have been included in **Extended Data Figure 9f**.

Reviewer #4

(Remarks to the Author)

The manuscript by Hu et al. introduces a novel mouse model to deplete FOXP3 protein expression in FOXP3+Tregs. They report a limited effect of induced FOXP3 protein degradation on the gene expression of mature Tregs, which appear to be enriched for direct FOXP3 target genes. This is insufficient to induce multiorgan autoimmune inflammation. Conversely, the gene expression of thymic, proliferating or tumor-infiltrating Tregs is more strongly affected by induced FOXP3 degradation. Overall, this is a highly interesting, relevant, innovative and elegant study, but the following points need to be clarified:

Major points:

1. Impact of prolonged FOXP3 degradation: On page 4 the authors wrote: "Notably, immune cell activation and tissue inflammation following Foxp3 degradation appeared to reach a new setpoint, as no further increases were observed between two and four weeks of 5-ph-IAA treatment (Figure 1f-k, Extended Data Figure 3b-d)." However, in the cited Figures there are significant differences for neutrophil and monocyte numbers, as well as IgE levels exclusively at day 28. Later time points were not analysed. It is thus still possible that FOXP3 degradation in mature Tregs may also induce autoimmunity, but with delayed kinetics. Since IL2Ralpha/beta are among the direct FOXP3 target genes and are critical for Treg homeostasis it would be surprising if Treg function would not be affected more strongly in the long run.

It is not our intention to suggest that Foxp3 plays no role in mature Treg cell function. Rather, we propose that the degree of Foxp3 dependency varies across different stages of Treg cell development. While mature Treg cells in healthy mice are only mildly dependent on Foxp3, developing and tumor-infiltrating Treg cells are significantly more reliant on Foxp3 for maintaining

their identity and suppressive function. We agree with the Reviewer that long-term Foxp3 loss—beyond the four-week window we examined—will eventually compromise Treg gene expression and function, ultimately resulting in autoimmune pathology. However, this occurs with much slower kinetics in mature Treg cells compared to developing or recently formed Tregs. For example, 14 days of Foxp3 degradation induced substantially less autoimmunity than Treg cell ablation in adult mice with fully differentiated Treg cells, whereas the same experiment in neonates with developing Tregs produced effects indistinguishable from Foxp3 gene deficiency. We will revise the manuscript to clarify this conclusion. (see also our reply to Reviewer #2)

2. Tregs in non-lymphoid tissues: sc-RNA seq was performed with Tregs from secondary lymphoid organs, but not from non-lymphoid tissues. Some targets were however validated by flow cytometry in the lung and unveiled sometimes dramatic differences (Ox40, TCF1). The effect of FOXP3 degradation in Tregs in non-lymphoid tissues is highly relevant and should be analysed more in detail, ideally by sc-RNA seq.

We agree with the Reviewer that tissue-resident Treg cells merit more detailed analysis in this study. To experimentally address reviewer's questions, we performed single-cell RNA-seq analysis of Treg cells isolated from the lung, liver, large intestine lamina propria (LILP), and secondary lymphoid organs (SLOs) of mosaic heterozygote female *Foxp3*^{AID/WT} *R26*^{TIR1(F74G)} mice harboring a mix of *Foxp3*^{AID} and *Foxp3*^{WT} expressing Treg cells after 7 days of continuous Foxp3 degradation under physiologic conditions. Analysis and discussion of this experiment and its results are now included in the revised manuscript (**Extended Data Figure 8**). Please, see our response (#1) to Reviewer 2 above for a brief summary of our findings.

3. Proliferation versus cellular “context”: The key point of this study is the relatively mild phenotype of FOXP3 degradation in mature Tregs. In Figure 5 the authors provide evidence for a key role for Treg proliferation in this surprising phenomenon. A key question that remains unanswered is if the observed “vulnerability” of Tregs to degradation in the thymus, in and in tumors is a mere consequence of their high proliferation rates, or if these cellular contexts may require Tregs with intact FOXP3 protein expression also independently of proliferation.

We agree with the Reviewer that factors beyond proliferation rate can contribute to the vulnerability of Treg cells to Foxp3 degradation. One such factor is the developmental stage. Differentiating and recently generated Treg cells rely heavily on Foxp3 to establish Treg-specific transcriptional and epigenetic programs. We believe this reliance significantly contributes to the sensitivity of thymic and neonatal Treg cells to Foxp3 degradation. Notably, neonatal Treg cells are also highly proliferative, which likely exacerbates their dependency on Foxp3.

In addition to proliferation, inflammatory cytokines—known to destabilize Treg cells—may further increase their reliance on Foxp3. The inflammatory tumor microenvironment may thus render tumor-infiltrating Treg cells particularly dependent on Foxp3 for their stability and function. To assess the impact of inflammatory cytokines on Treg Foxp3 dependency, we adoptively transferred 1:1 mixes of *Foxp3*^{AID} *R26*^{TIR1(F74G)} and *Foxp3*^{AID} *R26*^{WT} cells into a lymphopenic setting (*Tcrb/Tcrd* double knockout mice) and a systemic inflammatory setting (*Foxp3*^{3DTR} mice

undergoing continuous Treg ablation via DT administration). After 7 days of continuous Foxp3 degradation, $R26^{TIR1(F74G)}$ and $R26^{WT}$ were isolated by FACS (sorting on the marker mCherry) and single-cell RNA-seq was performed (**Figure 6i**). This approach allowed us to directly compare the effects of Foxp3 degradation in two distinct in vivo contexts: one that involves proliferation and activation with minimal inflammation (*Tcrb/Tcrd* double KO) and another that combines proliferation and activation with systemic, polytypic autoimmunity. This experiment and subsequent analysis have been added to the manuscript (**Figure 6i-o**). Please see our response to Reviewer #2 for a brief summary of our findings. Overall, our findings suggest that inflammation is an additional factor that heightens the dependence of Treg cells on Foxp3. The precise mechanism behind the increased dependence of Treg cells on Foxp3 in the settings of inflammation and proliferation as well as the specific signals that perturb Treg cell transcriptional networks in inflammatory settings are key future directions and will be followed up on in future studies. We have modified the manuscript to highlight both inflammation and proliferation as factors that increase vulnerability of Treg transcriptional and functional programs.

4. A limitation of this -otherwise very complete- study is the lack of human data, in particular in Figure 7 that has translational potential.

We appreciate the Reviewer's comment regarding the translational potential of our findings. However, generating human data is currently challenging due to the absence of effective methods to degrade human Foxp3 protein. Our results provide a strong rationale for developing small molecule Foxp3 degraders such as PROTACs or molecular glues to target human Foxp3 in clinical settings, foremost, in cancer. However, such major efforts fall beyond the scope of the current study.

Minor points:

General:

This is a very complex story and any effort should be made to simplify and explain every step to the readers.

We have modified the text to better explain key aspects of our story.

Figure 1C. A convincing density plot overlay is shown, but the FOXP3 expression levels in several mice should be shown.

We have added the requested plot to the revised manuscript.(Fig.3e).

Figure 1f-k: I would suggest using different colours for d14 and d28

We have changed the colors in the revised manuscript.

Figure 2d: The authors should report the differences in cell abundance across samples and

conditions. Otherwise the relative representation of $Foxp3^{AID}R26^{WT}$ and $Foxp3^{AID}R26^{TIR1}$ Treg cells cannot be accurately inferred from the bar plot.

We have replaced Figure 2d with the requested plots.

Figure 2i: The number of down-regulated genes in resting Tregs is only slightly reduced, all others are dramatically reduced. This was not clearly stated in the text (“only a very small set of genes”, this seems to be an overstatement). Perhaps these genes could be simply shown in a supplementary Table?

We have updated the text to more accurately reflect the data in the revised manuscript. In addition, we have included a supplementary table (**Supplementary Table 1**) listing the DEGs for all analyses performed in the manuscript.

Figure 3d: This is a rather complex panel. Interestingly, IL-17A is upregulated in the WT control. A comment on this point in the Discussion would be appropriate.

We apologize for the confusion the panel 3d has caused. This dot plot summarizes the differentially expressed genes in resting or activated cells between $R26^{TIR1}$ and $R26^{WT}$ treated with 5-ph-IAA for 3 or 7 days. The color of each dot is the log fold change between $R26^{TIR1}$ and $R26^{WT}$ and the size of the dot reflects the p-value of the comparison. The columns indicated by the white and black bars do not represent $R26^{TIR1}$ and $R26^{WT}$ cells, but rather resting and activated cells. Thus, IL-17A is not upregulated in the WT controls; it is upregulated upon Foxp3 degradation in activated cells (but not resting cells). We have modified this figure and its legend to prevent any confusion.

Figures 4a/b: What do the individual data points represent? Are they outliers?

We apologize for the confusion caused by the labeling. The dots originally represented the number of genes in each group. For the sake of clarity, we have removed these dots from the figure as the information on the number of genes in each group is shown below each plot as well.

Figure 4C: This panel reports data from CHIP experiments as far as I understood. This was not mentioned anywhere, please clarify.

This analysis is based on previously performed ChIP experiments (Konopacki et al., *Nature Immunology*, 2019). We have included the appropriate citation in the revised manuscript.

Figure 5d-f, n-o are very complex. The Foxp3 expression levels in Figure 5f are scaled. Unscaled values should be used for a more accurate representation.

We have plotted the unscaled UMI-normalized count values in Figure 5f. The conclusions are unchanged.

Figure 7k/n: both orange and red points are labelled R26Tr1, which is the WT control? In the dataset these conditions were apparently mislabelled.

We apologize for the mislabeling. We have revised the figure and correctly labelled $R26^{WT}$ and $R26^{TIR1}$ samples.

Extended Figure 4: it is not clear to me why this control excludes a role of residual FOXP3 protein expression in suppressive functions.

Our flow cytometry analysis indicates that 5-phe-IAA induced degradation achieves close to complete Foxp3 protein elimination, yet not more than approximately 5% of the protein may remain (Figure 3e). Of note, Wan and Flavell showed that a 10-fold decrease in Foxp3 transcript and protein expression, caused by insertion of a tandem IRES-driven reporter sequences, resulted in a severe impairment of Treg function and autoimmunity (*Nature*, 2007). Nevertheless, we were concerned that potential residual Foxp3 could still bind chromatin and support Treg function. To address this, we used the inducible CD4-CreERT2 system to delete the Foxp3 gene in mature Treg cells. Given Foxp3 protein's rapid turnover, this approach results in complete loss of the Foxp3 protein shortly after gene deletion. This experiment provides an independent validation of our conclusions and addresses the limitation of imperfect protein degradation. We have clarified and emphasized this point more thoroughly in the revised manuscript.